# Structural basis for ligand promiscuity and high signaling activity of Kaposi's Sarcoma-associated Herpesvirus-encoded GPCR

Jun Bae Park[1,3], Bibekananda Sahoo [2,3], Amita Rani Sahoo [2,3], Dokyun Kim[1,3], Hogyu David Seo[1,3], James Bowman[1], Mi-Jeong Kwak[1], Sophia Suh[1], Matthias Buck[2] ✉, Xinghong Dai [2] ✉ & Jae U. Jung [1] ✉

Kaposi's Sarcoma-associated Herpesvirus encodes ORF74, a viral G protein-coupled receptor homologous to CXCR2, which plays a crucial role in Kaposi's Sarcoma development through its high basal signaling activity. Our cryoEM analysis of ORF74 in ligand-free, BRIL-fused ligand-free, and CXCL1/Gi$_{trimer}$-bound forms elucidates its ligand-independent signaling activity. A widely open, static extracellular cavity facilitates ligand promiscuity by enabling dynamic access and diverse binding modes. Structural alterations in CWxP, E/DRY, and NPxxY micro-switches stabilize the active conformation, leading to constitutive signaling. Metadynamics simulations reveal a dynamic ensemble between local switch structures corresponding to the inactive and active states, supporting spontaneous activation. CXCR2-ORF74 chimeras highlight intracellular loops 2 and 3 as key modulators of basal and agonist-induced activity. This study defines the structural basis of ORF74's ligand promiscuity, spontaneous activation, and high basal signaling, providing insights into its role in viral oncogenesis.

Human Herpesvirus-8 (HHV-8), also known as Kaposi's Sarcoma-associated Herpesvirus (KSHV), is the etiological agent of Kaposi's Sarcoma (KS)[1], multicentric Castleman's disease[2], and primary effusion lymphoma[3,4]. KS is the most prevalent neoplasm in AIDS patients and is endemic in sub-Saharan Africa[5]. Considering the global prevalence of KSHV-derived cancers in 2020, 73% of cases occurred in Africa, with the majority occurring in Eastern and Southern Africa. Additionally, ~15,000 deaths are attributed to Kaposi's sarcoma annually, with 86.6% occurring in Africa. The high occurrence in Africa is likely due to the relatively higher rates of KSHV and HIV coinfection than in other regions[6,7]. Notably, recent studies have also reported a concerning increase in the incidence of KS among younger populations in the U.S.[8,9]. There are no targeted therapies available for KSHV-derived cancers.

KSHV encodes multiple oncogenes, with ORF74 being one of the most potent[10,11]. ORF74 encodes a viral G protein-coupled receptor (vGPCR) capable of endothelial cell transformation in vitro and KS-like tumorigenesis in vivo[12]. G protein coupling leads to dysregulated activation of a wide range of oncogenic signaling cascades, including the ERK1/2, JNK, RhoA, and Rac1 pathways[13–15]. Expressed during the latent infection cycle, ORF74 mediates type-I interferon immune evasion and angioproliferation through oncogenic pathways, including the VEGF, Akt, PLC-mediated MAPK, and NF-κB pathways[14,16–18].

ORF74 was identified as a homolog of the chemokine receptor CXCR2, which is a member of the class A GPCR subfamily[19]. Notably, ORF74 was shown to be activated by a wide range of host (human) chemokines, including members of the CC and CXC families (CCL1, CCL5, CXCL1–CXCL8, CXCL10, and CXCL12), as well as the KSHV-

[1]Cancer Biology Department, Infection Biology Program, and Global Center for Pathogen and Human Health Research, Lerner Research Institute, Cleveland Clinic, Cleveland, OH, USA. [2]Department of Physiology and Biophysics, School of Medicine, Case Western Reserve University, Cleveland, OH, USA. [3]These authors contributed equally: Jun Bae Park, Bibekananda Sahoo, Amita Rani Sahoo, Dokyun Kim, Hogyu David Seo. ✉e-mail: mxb150@case.edu; xxd159@case.edu; JUNGJ@ccf.org

encoded chemokine vCXCL2[16,20]. This repertoire is approximately twice as extensive as the known target ligands for CXCR2 (CXCL1, 2, 3, 5, 6, 7, 8)[21]. Investigations have revealed that CXCL1 and CXCL3 function as full agonists, CXCL2 functions as a partial agonist, CXCL4, 5, 7, and 8 function as neutral antagonists, CXCL6 functions as a partial inverse agonist and CXCL10, 12, and vCXCL2 functions as full inverse agonists[22–25].

ORF74 has been shown to have robust and high basal activity for multiple signaling pathways[19,23,26,27]. The potent transformative capacity of ORF74 is attributable to both its ligand-independent activity and ligand-induced signaling, coupled with its promiscuous interactions with the $G_{\alpha i}$, $G_{\alpha 12/13}$, and $G_{\alpha q}$ proteins[19,28]. Notably, ~80% of $G_{\alpha i}$-mediated signaling of ORF74 is attributed to its basal activity, whereas the remaining 20% is induced by agonist stimulation[29]. These properties were not observed in its human homolog CXCR2[16,17].

The high basal signaling activity of ORF74 highlights its potential as a therapeutic target for KS[12,30]. In a transgenic mouse model, sustained ORF74 expression in endothelial cells was essential for KS development[31,32]. Notably, although ORF74-expressing cells constitute only a small fraction of the tumor cell population, their targeted pharmacological elimination effectively mitigated tumor progression and induced apoptosis in cells expressing KSHV latent genes[10]. Moreover, pharmacological interventions targeting signaling pathways activated by this vGPCR during KS pathogenesis have shown promising results[10,33].

Despite its significant role in KSHV oncogenesis, the mechanisms underlying ORF74-mediated signaling initiation remain unclear. In this study, we present cryoEM structures of ORF74 in ligand-free, BRIL-fused ligand-free, and CXCL1 & $Gi_{trimer}(G_{\alpha i}/G_{\beta 1}/G_{\gamma 2})$-bound states, providing an inactive-active pair of structures of a vGPCR, allowing comparisons and insights into the likely conformational transitions. To further investigate the direction of these transitions, we employed metadynamics calculations[34,35], which have been successfully used to compute the free energy landscapes of complex conformational rearrangements in different types of receptors[36,37].

Our findings suggest that ORF74 exhibits a uniquely large, inactive to active state unaltered (i.e. essentially static) extracellular protein surface, stabilized by an outward-tilted of transmembrane helix 2 (TM2) and a rigid extracellular loop 2 (ECL2), enabling promiscuous ligand recognition with minimal conformational change. Unlike typical Class A GPCRs, ORF74 micro-switches facilitate spontaneous activation. Metadynamics simulations reveal a dynamic ensemble between local structures corresponding to the inactive and active states. In this process, TM6 displacement increases intracellular cavity accessibility, which promotes spontaneous activation. Additionally, intracellular loop 2 (ICL2) and intracellular loop 3 (ICL3) act as key regulators of signaling. These findings provide insights into the high basal signaling activity of ORF74.

## Results

### CryoEM structures of ORF74 in the CXCL1-bound (active) and apo (inactive) state

To optimize the expression level and structural stability of ORF74, we conducted systematic screening of single and combinatorial mutations, and selected a construct with L169W, L258V, and two threonine residue deletions at the C-terminus (Supplementary Fig. 1a). To capture the CXCL1-bound state, we employed a disulfide trapping system[38] by coexpressing ORF74-G30C with CXCL1-N56C (Supplementary Fig. 1b). β-arrestin recruitment assays suggested that these stabilizing and cysteine mutations do not significantly alter ORF74-mediated β-arrestin signaling (Supplementary Fig. 1c).

The CXCL1-ORF74 complex was purified from HEK293S GnTI⁻ cells. This complex was then combined with a recombinant $G_{\alpha i}/G_{\beta 1}/G_{\gamma 2}$ heterotrimer ($Gi_{trimer}$) and stabilized using an anti-G protein scFv16 (Supplementary Fig. 1d). We used single-particle cryoEM to determine the structure of the CXCL1-ORF74-$Gi_{trimer}$-scFv16 complex at ~3.0 Å

resolution. The cryoEM densities clearly showed CXCL1 and $Gi_{trimer}$ on the detergent-encapsulated ORF74 surface (Fig. 1a, Supplementary Fig. 2).

This structure exhibits the canonical features of an active GPCR, including a solvent-exposed extended N-terminus, seven TM regions embedded in a detergent micelle, and a short membrane-proximal C-terminal helix. CXCL1 binds the extracellular surface of ORF74, inserting its N-terminus into the extracellular cavity between the TMs (a clear density for the side chain of R8 acting as an anchor could be seen), while its body interacts with the extended N-terminus of ORF74 (Fig. 1b). On the intracellular side, TM6 is closer to TM5, facilitating the $G_{\alpha i}/\alpha 5$-helix insertion into the intracellular cavity formed by the TMs (Fig. 1c, Supplementary Fig. 3). Extensive electrostatic interactions between ICL2, ICL3, and the bound G protein heterotrimer is seen, which contribute to ORF74's high basal activity as discussed later.

To obtain the inactive structure of ORF74, we expressed and purified ORF74 without the agonist CXCL1. We optimized the sample for cryoEM by reconstituting the detergent-purified apo-ORF74 in nanodiscs (Supplementary Fig. 1d, first panel). CryoEM analysis revealed homodimer formation in the nanodiscs, enabling structure determination at 2.9 Å resolution despite ORF74's small size (~38 kDa) (Fig. 1d, Supplementary Fig. 4). Unexpectedly, atomic modeling showed an antiparallel orientation of the two protomers, an uncommon feature among GPCRs, likely induced during purification. Notably, the apo-ORF74 protomer displayed significant structural differences from the CXCL1 & $Gi_{trimer}$-bound active form, contrary to expectations based on its high basal signaling activity.

To validate whether the apo structure of ORF74 represents a biologically relevant conformation despite the antiparallel arrangement within the dimer, we expressed apo-ORF74 fused with a BRIL domain[39] on its ICL3 for cryoEM imaging (Supplementary Fig. 1d, second panel). We constructed the BRIL-fused ORF74 to serve as a fiducial marker, facilitating structure alignment during data processing. Purified ORF74-BRIL fusion protein formed a monomer when reconstituted into nanodiscs under the same conditions as the "wild-type" ORF74. However, the flexibility of the BRIL domain limited the resolution to 3.7 Å in this case (Supplementary Fig. 5). Nevertheless, the apo-ORF74 model from the antiparallel dimer showed a strong structural fit to the BRIL-fused ORF74 map, while the CXCL1 & $Gi_{trimer}$-bound ORF74 differed significantly (Fig. 1e). This suggests that the apo-ORF74 structure represents a bona fide inactive conformation.

When the apo and active conformations of ORF74 were superimposed to analyze their structural differences (Fig. 1f), the root-mean-square deviation (RMSD) value for this overlay was 2.7 Å based on the Cα atoms. All of the transmembrane helices except for TM6 and TM7 showed minimal positional changes between the two conformations. The most noticeable conformational changes around the TM6, TM7, and ICL3 regions, are characteristic changes associated with a typical GPCR activation mechanism (Fig. 1f). Specifically, the significant shift in the position of TM6 relative to TM7 during the transition from the inactive to active state facilitates the displacement of ICL3, creating an opening in the intracellular cavity for $Gi_{trimer}$ binding. To further investigate whether the apo conformation adopts an inactive form, we compared the apo protomer model of ORF74 with the known inactive conformation of CXCR2[40] (Supplementary Fig. 6). An RMSD of 1.7 Å from their superposition suggested similarities between these two structures including the positioning of the ECL2, 7 TMs, and ICL3. Notably, in both structures, ICL3 occupies a position that would sterically clash with the incoming $G_{\alpha i}/\alpha 5$-helix of the $Gi_{trimer}$ in the active conformation. These observations further complicate the understanding of molecular mechanisms for the activation and signaling of ORF74.

### The unique extracellular surface area of ORF74 contributes to its promiscuity in ligand-binding

One of the unique features of ORF74 is its promiscuous ligand binding ability and high degree of basal signaling[20,41]. To characterize the

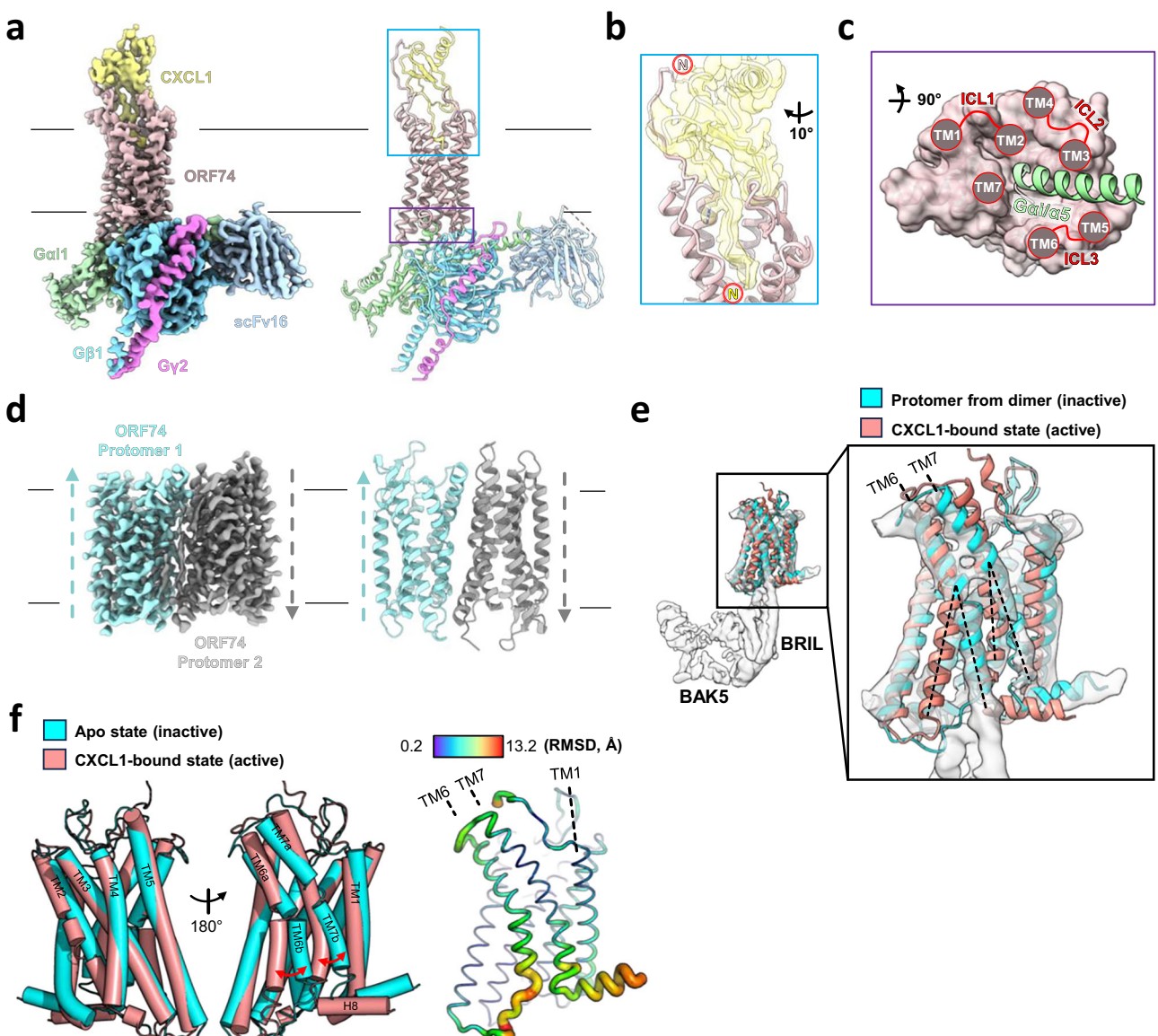

**Fig. 1 | CryoEM structures of ORF74 in the CXCL1-bound, Gi_trimer-bound active and apo-state inactive conformations. a** Color coded composite cryoEM density map and model (cartoon representation) of the CXCL1 & Gi_trimer-bound active ORF74 complex. **b** Zoomed view of the CXCL1 binding to ORF74, with their complementary N-terminal regions highlighted. **c** Binding mode of the α5-helix of the $G_{αi}$ subunit ($G_{αi}$/α5, light green) at the intracellular surface of ORF74 (pink). **d** Color coded cryoEM map and model (cartoon representation) of the ORF74 apo antiparallel dimer structure with color coded arrows indicating orientation of respective protomers. **e** CryoEM map of the ORF74-BRIL fused construct (gray surface representation), fitted with the ORF74 model derived from the antiparallel dimer structure (cyan) and the CXCL1 & Gi_trimer-bound structure (pink). Map/model fitting results with positions of the TM6 and TM7 regions from both the models highlighted as black dashed lines. **f** Superimposition of the apo state structure (cyan) and the CXCL1 & Gi_trimer-bound structure (pink) highlighting dynamic movements of TM6 and 7, illustrated with red arrows, along with their RMSD comparison.

extracellular surface (ECS) of the orthosteric binding site, which is solvent- and ligand-accessible and exposed in the extracellular region of the GPCR, we compared the ECSs of inactive and active ORF74 structures with those of CXCR2 and BILF1 of Epstein-Barr virus (EBV) (Fig. 2a). Compared with that of BILF1, the ECS of ORF74 displays an open conformation that is more similar to that of CXCR2. However, it does not demonstrate the distinctive structural movements between the active and inactive states observed in CXCR2 (Fig. 2a). To facilitate quantitative comparison, we calculated the ECS values for these receptors. Both inactive and active ORF74 conformations exhibited relatively large ECS of ~65.9 nm² and 69.1 nm², respectively (Fig. 2b). In contrast, CXCR2 displayed significantly smaller ECSs values of 47.1 nm² and 56.8 nm² for the inactive and active states, respectively (Fig. 2b). Ligand binding in CXCR2 induces substantial expansion of

ECSs through residue rearrangements, a critical factor in its activation and signaling[40]. However, despite the ECS being large, ORF74 exhibited minimal ECS differences between the apo inactive and CXCL1 & Gi_trimer-bound active states. This observation demonstrates the unique ECS dynamics of ORF74 with limited structural rearrangements upon ligand binding. This distinctive feature of ORF74 might contribute to its ligand promiscuity by facilitating the dynamic access and binding of various ligands. In contrast, the self-activated BILF1 exhibits a markedly different extracellular surface morphology with an exceptionally small ECS of 38.1 nm² mostly because of occlusion by ECL2 rearrangement and N-terminal folding, limiting solvent accessibility (Fig. 2a, b).

Comparative structural analysis of the inactive and active states of ORF74 revealed two distinctive ECS features that likely contribute to its suboptimal ligand-receptor complementarity. First, the TM2

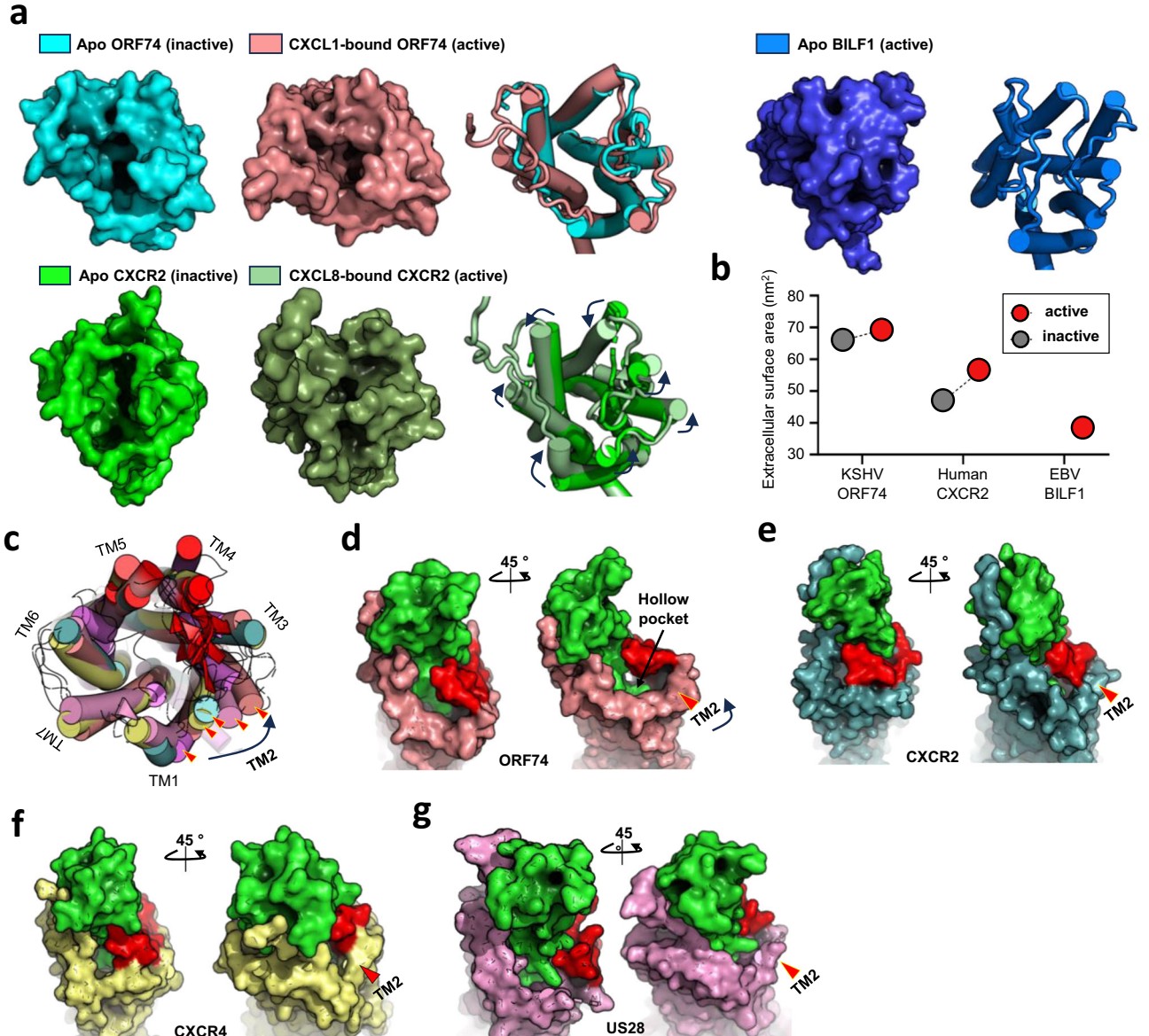

**Fig. 2 | The unique extracellular surface area of ORF74 contributes to its promiscuity in ligand-binding. a** Comparison of the ligand binding site structures in the active and inactive conformations of ORF74 and CXCR2. The movement of CXCR2 TMs is indicated by black arrow lines. The extracellular loops of BILF1 are structurally relatively clustered, resulting in a closed ligand binding pocket. Since BILF1 does not have an identified ligand-bound structure, it was excluded. **b** Graph comparing the extracellular surface areas of ORF74, CXCR2, and BILF1 in their active and inactive conformations. Red and gray circles represent the active and inactive conformations, respectively. **c** A superimposition of the ligand-binding pockets of various chemokine GPCRs, including viral GPCRs. The red arrow within the yellow frame indicates the TM2 region of each GPCR, with BILF1 (pale pink

color), vMIP-II bound CXCR4 (yellow color), CXCL8-bound CXCR2 (turquoise color), CX3CL1-bound US28 (dark pink color), and CXCL1 & Gi$_{trimer}$-bound ORF74 (brown color) arranged from left to right. The characteristic TM2 tilting of ORF74 is highlighted by a black arrow line. **d** Formation of a distinctive hollow pocket in ORF74 resulting from the unique structural conformation of TM2 upon chemokine ligand binding. Comparative analysis of this structural feature of ORF74 with **e** CXCR2, **f** CXCR4 and **g** US28. Ligands bound to each GPCR are shown in green, with TM2 helices indicated by red arrowheads and ECL2 surface-rendered in red. The Protein Data Bank codes for the structures are as follows: CXCR4 (4RWS), CXCR2 (6LFO), US28 (4XT1), and BILF1 (7JHJ).

displayed outward tilting displacement relative to the receptor's central axis in the active and inactive conformations (Fig. 2c, Supplementary Fig. 7). Consequently, ECL1 which connects TM2 and TM3, exhibited a concomitant outward shift, resulting in a consistent cavity around this region, even in the ligand-bound active state. Second, the consistent positioning of ECL2 in both ORF74 conformations also contributed to this extracellular surface cavity. Unlike chemokine receptors such as CXCR2[40], CXCR4[40,42], and human cytomegalovirus (HCMV) US28[40,42,43], where the ECL2 of these structures undergoes a downward shift upon ligand binding to form a latch-like structure, that

of ORF74 maintains a static position, creating a hollow pocket (Fig. 2d–g). These findings demonstrate that ORF74 possesses a distinct extracellular ligand-binding pocket that ultimately contributes to its unique ligand promiscuity.

## Alteration of the CWxP and E/DRY micro-switches of ORF74 for high basal signaling activity

The structural rearrangement of class A GPCRs upon agonist binding is well characterized[44,45], with key micro-switches modulating the equilibrium between inactive and active states to promote G protein

binding[44,45]. To understand ORF74's high basal signaling activity, we analyzed its micro-switches in comparison with conventional Class A GPCRs. To enable a comparison of equivalent positions across different members of the GPCR family, we employed the Ballesteros–Weinstein numbering system[46].

**CWxP motif.** The C-W$^{6.48}$-x-P motif (where x represents any residue) in TM6 is highly conserved across class A GPCRs[44,47]. Within this motif, the W$^{6.48}$ residue plays a pivotal role in the global "rotamer toggle switch" model because of its critical function in the activation mechanism of GPCRs upon agonist binding[47–49]. In the inactive state, W$^{6.48}$ forms a central sodium ion-binding allosteric pocket with D$^{2.50}$, stabilizing the inactive structure[50] (Fig. 3b, first panel). Upon ligand binding, the movement of the W$^{6.48}$ residue induces conformational changes in the central sodium ion-binding allosteric pocket, promoting the transition to the active state[50]. Notably, ORF74 harbors the F$_{263}$C$^{6.48}$FP$_{266}$ motif instead of the CWxP motif (Fig. 3a, b), preventing the typical W$^{6.48}$-mediated toggle switch mechanism upon ligand binding. Additionally, the substitution of D$^{2.50}$ with S$^{2.50}$ (Fig. 3a, b) abolishes the potential for stabilizing the inactive state through a central sodium ion-binding allosteric pocket.

Structural analysis of inactive ORF74 revealed a segmented TM6 helix, composed of distinct TM6a and TM6b regions (Fig. 3c). The outward tilt of the F261$^{6.45}$ carbonyl oxygen disrupted the continuous α-helical structure, forming a short loop between TM6a and TM6b (Fig. 3d). This unique discontinuous TM6 architecture persisted in the active structures of ORF74 and CXCR2. Interestingly, the C$^{6.48}$ residue is located within the TM6a-TM6b loop (Fig. 3c). Metadynamics simulations were employed to assess the structural implications of the C$^{6.48}$ residue in both the active and inactive states of ORF74. Given the absence of a bulky side chain in cysteine compared to tryptophan, we analyzed the flexibility of the Cα-Cβ plane (χ1-angle) of C$^{6.48}$ in ORF74 to W$^{6.48}$ in CXCR2 and BILF1 throughout the simulations. The inactive CXCR2 exhibited a broad χ1-angle distribution, although it was limited to −50° to −100° (Fig. 3f). Conversely, the active structures of CXCR2 and BILF1 displayed restricted χ1-angle mobility due to G$_{\alpha i}$ interactions[40] (Fig. 3f, g). Notably, the inactive ORF74 mirrored the inactive CXCR2, whereas active ORF74 presented increased χ1-angle distribution and flexibility (Fig. 3e). These findings suggest that the C$^{6.48}$ residue has increased mobility upon G$_{\alpha i}$ binding, unlike the W$^{6.48}$ residue in CXCR2 and BILF1. This enhanced flexibility would affect TM6 dynamics, potentially allowing it to sample both inactive and active conformations.

**E/DRY motif.** The E/D$^{3.49}$-R$^{3.50}$-Y$^{3.51}$ (E/DRY) motif, located at the interface of TM3, ICL2, and ICL3 of class A GPCRs, plays a crucial role in modulating GPCR conformational states and is directly implicated in regulating receptor conformation, G protein recognition, and coupling[51–53]. The basic R$^{3.50}$ forms stabilizing intramolecular interactions, notably with the adjacent D/E$^{3.49}$ and/or another negatively charged residue at position 6.30 on TM6, thereby constraining GPCRs in the inactive conformation[45,54]. This mechanism is partly shared by β$_2$AR and CXCR2[40,55] (Fig. 3h, second and third panels). Strikingly, ORF74 has a unique structural feature, with a valine (V142) at position 3.49 instead of an acidic residue (Fig. 3a). This substitution restricts the movement of the guanidine group of R$^{3.50}$ toward position 3.49, resulting in a distinctive conformation of R$^{3.50}$ in the inactive state (Fig. 3h, first panel). Furthermore, the R$^{3.50}$ residue of inactive β$_2$AR and CXCR2 is additionally stabilized by hydrophobic interactions with residues at positions 6.34 and 6.37 (Fig. 3h). Unlike the hydrophobic residues L$^{6.34}$ or M$^{6.34}$ which are residues of inactive β$_2$AR or CXCR2, inactive ORF74 contains an R$^{6.34}$ residue that blocks the engagement of R$^{3.50}$. Position 6.34 is situated on TM6 in β$_2$AR and CXCR2, whereas in ORF74, it is located on ICL3 (Fig. 3h). This unique

sequence rearrangement involving the V$^{3.49}$ and R$^{6.34}$ residues leads to the distinctive conformation of the R$^{3.50}$ residue in the inactive ORF74.

We then compared the structures focused on the conserved ionic lock switches of ORF74 with those of inactive and active CXCR2 and constitutively active BILF1. The ionic lock switch between TM3$^{3.50}$ and TM6$^{6.30}$ is a characteristic feature of inactive structures, but upon activation, it breaks, leading to the movement of TM6 relative to TM3. The ionic lock is less conserved in class A GPCRs, because the residues that are typically involved in the ionic lock have been replaced with other residues that have a similar positive charge. For example, in ORF74, the ionic lock consists of R143$^{3.50}$ and R246$^{6.30}$, in CXCR2, it consists of R144$^{3.50}$ and R251$^{6.30}$, and in BILF1, it consists of K122$^{3.50}$ and K211$^{6.30}$. However, the mechanism of G protein binding pocket formation through TM6 tilting remains consistent[44]. When CXCR2 is activated, the R$^{3.50}$ residue in its E/DRY motif forms hydrogen bonds with Y$^{5.58}$ and C351 of the G$_{\alpha i}$ subunit. In contrast, ORF74 only forms hydrogen bonds between R$^{3.50}$ and Y$^{5.58}$, but not with C351 of the G$_{\alpha i}$ subunit (Supplementary Fig. 8). BILF1, which possesses a lysine residue at position 3.50 instead of an arginine, lacks the characteristic TM3-TM5-G$_{\alpha i}$ interaction observed in ORF74 and CXCR2 (Supplementary Fig. 8). In the inactive conformations of ORF74 and CXCR2, the R$^{3.50}$ residue is unbound, and its side chain orientation differs from that observed in the active structures. On the basis of these structural features, we analyzed the side chain chi angles (χ1 and χ2) of R$^{3.50}$ throughout the metadynamics simulations for both active and inactive ORF74 and compared them with those of the active and inactive CXCR2, as well as with those of constitutively active BILF1 (Fig. 3i). In the inactive ORF74 simulation, the side chain rotamers overlapped with those observed in the inactive and active rotamers of ORF74 and CXCR2. Notably, it also explored conformational spaces absent in the active ORF74 and CXCR2 but present in BILF1 (Fig. 3i). These observations provide strong evidence that during inactive ORF74 simulations, the R$^{3.50}$ residue also explores transition states similar to those observed in the constitutively active BILF1. In conclusion, we confirmed that ORF74 possesses a VRY motif, which facilitates a unique R$^{3.50}$ position, contributing to its high basal signaling activity.

## Alteration of the NPxxY micro-switch of ORF74 for high basal signaling activity

**NPxxY motif.** The N-P-x-x-Y$^{7.53}$ motif (where x denotes any residue) located in TM7 has been demonstrated to play a crucial role in the ion-water network across class A GPCRs by mediating a structural rearrangement without direct interaction with the G protein upon activation[44,56,57]. This phenomenon has been elucidated in the high-resolution structure of β$_2$AR, which forms a water-GPCR interaction network[55]. CXCR2 has the same NPxxY motif as β$_2$AR[40] (Fig. 4a). N$^{7.49}$ from the NPxxY motif simultaneously coordinates two water molecules, thereby constraining the movement of TM1, TM2, and TM7. This interaction is essential for stabilizing the inactive state and is disrupted by the indirect effects of agonist binding[58,59] (Fig. 4a). Notably, in ORF74, position 7.49 is occupied by valine (V310) instead of asparagine, precluding the water-mediated inactive-state stabilizing mechanism typically associated with the NPxxY motif (Fig. 4a). The hydrophobicity of the valine residue impedes the approach of water molecules, destabilizing the inactive structure. This finding suggests that the presence of V$^{7.49}$ is one of the key structural features enabling ORF74 to destabilize its inactive state and maintain high basal signaling activity.

To elucidate the impact of the distinctive inactive conformation of ORF74 on Y$^{7.53}$ rotamer dynamics, we compared the rotation of the chi angles of Y$^{7.53}$ in the NPxxY motif from our metadynamics simulations. Notably, in both active and inactive CXCR2, as well as active BILF1, Y$^{7.53}$ remained unbound and explored specific conformational

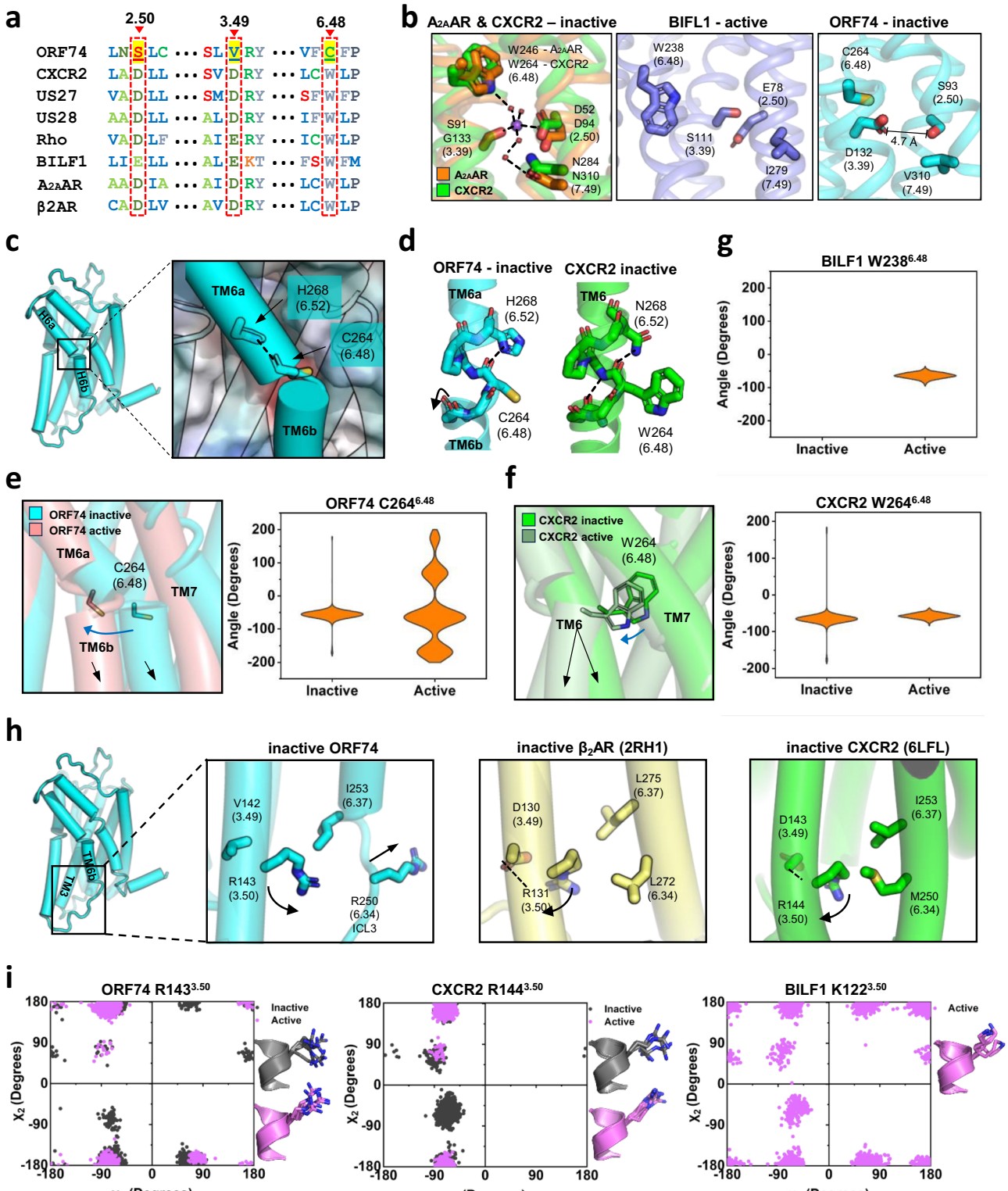

**Fig. 3 | Structure and metadynamics simulation of the CWxP and E/DRY motifs of ORF74. a** Amino acid sequence alignment focused on positions 2.50, 3.49 and 6.48. The proteins included in this analysis were bovine rhodopsin (Rho), human CXCR2, human cytomegalovirus US27 and 28, Epstein-Barr virus BILF1, human $A_{2A}AR$ and human $\beta_2AR$. The characteristic amino acid sequence of ORF74 is highlighted in yellow. **b** Structural comparison of the sodium-ion binding pocket that stabilizes the inactive conformation in inactive $A_{2A}AR$ (4EIY), inactive CXCR2 (6LFL), active BILF1 (7JHJ), and inactive ORF74. The red spheres represent water molecules, and the purple sphere represents the sodium ion. Hydrogen bonding is depicted by black dotted lines. **c** Characteristic TM6 structure and the unique positioning of Cys[6.48] in inactive ORF74. **d** Comparison of the structure of TM6 and the position 6.48 in the inactive form (6LFL) of CXCR2. **e** Structural movement of the characteristic TM6b portion of ORF74 and the χ1-angle distribution of Cys[6.48] in inactive-active conformations. **f** TM6 structural movement and χ1-angle distribution of Trp[6.48] in the inactive-active conformations of CXCR2. **g** χ1-angle distribution of Trp[6.48] in the active conformation of BILF1, which has a constitutive activation feature. **h** Structural comparison of the characteristic VRY motif and its surrounding residues of ORF74 with those of $\beta_2AR$ and CXCR2. **i** Comparative illustration of rotamer toggling (χ-torsion angle distribution) of R[3.50] observed in the simulations of inactive/ active ORF74 with that of inactive and active CXCR2, and active BILF1.

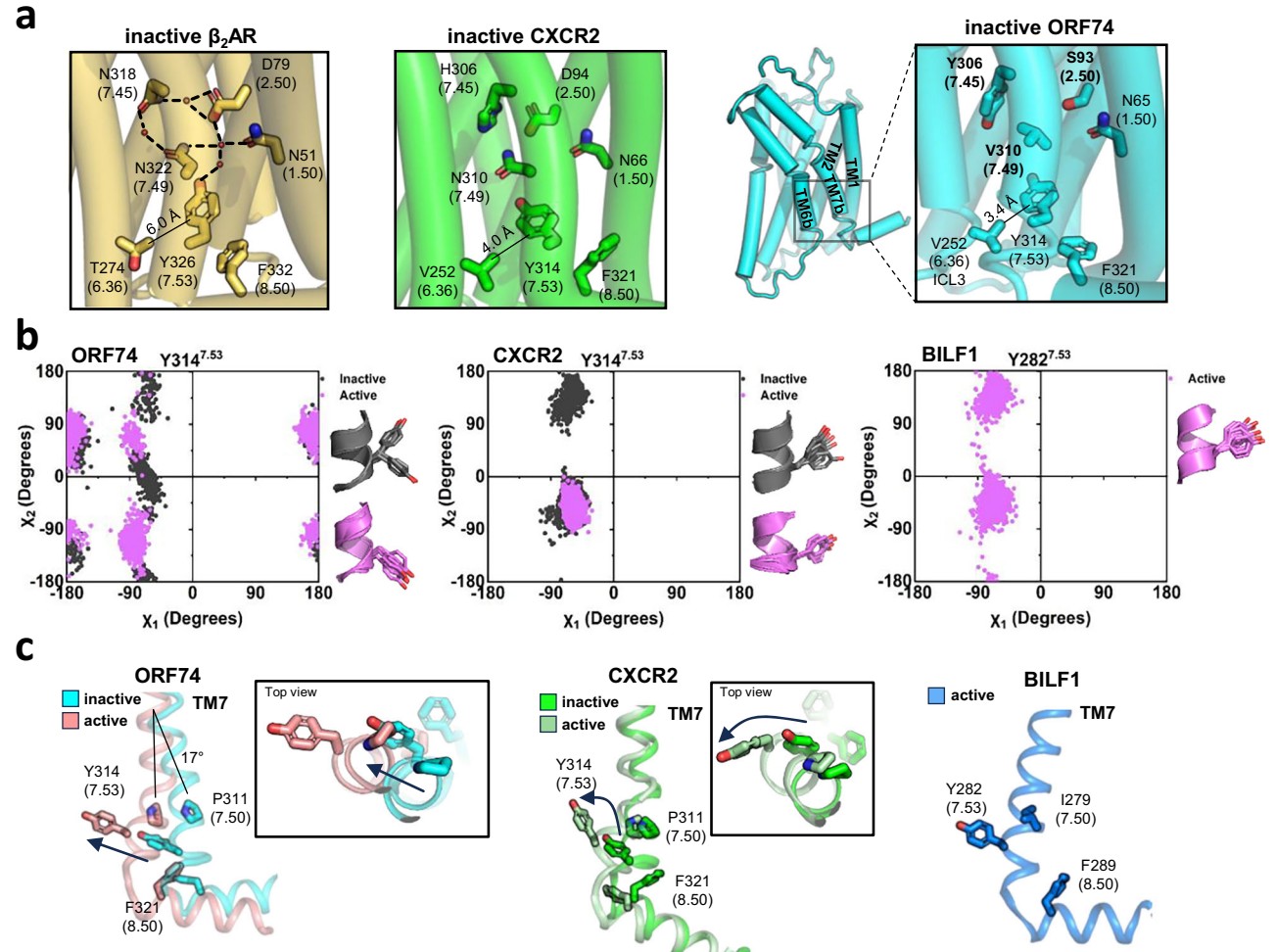

**Fig. 4 | Structure and metadynamics simulation of the NPxxY motif of ORF74.**
**a** Water-GPCR interaction network and the involvement of the NPxxY motif in $\beta_2$AR and CXCR2. Characteristic Val$^{7.49}$ substitution in ORF74 and its distinct TM7-ICL3 interaction. **b** Comparative illustration of rotamer toggling ($\chi$-torsion angle distribution) of Y$^{7.53}$ observed in the simulations of inactive/ active ORF74, alongside those of inactive and active CXCR2, and active BILF1. **c** The characteristic TM7 movement of ORF74 as revealed by the comparison of inactive and active conformations.

spaces during the simulations (Fig. 4b). However, in inactive ORF74, the sampling extends beyond both active and inactive conformations, encompassing structures in the direction of transitional intermediate state conformations, suggesting an overall structural transition from inactive ORF74 to active conformations.

Instead of conventional stabilization, Y$^{7.53}$ of ORF74 is anchored by hydrophobic interactions with V$^{7.49}$, V$^{6.36}$, and F$^{8.50}$. Specifically, the hydrophobic interaction of Y$^{7.53}$ with V$^{6.36}$ is a distinctive feature of ORF74 (3.5 Å distance) as this interaction is likely absent in $\beta_2$AR (6.0 Å distance between the two residues) and weak in CXCR2 (4.0 Å distance) (Fig. 4a). This unique characteristic can be attributed to the distinctive inactive conformation of TM6 in ORF74. While the 6.36 position in $\beta_2$AR and CXCR2 is located within the TM6 $\alpha$-helix, in ORF74 it is positioned within ICL3. These structural differences enable a unique inactive state stabilization mechanism specific to ORF74 (Fig. 4a).

To elucidate the effect of this unique Y$^{7.53}$ conformation on TM7 tilting dynamics, we superimposed the inactive and active structures of ORF74 and analyzed their structural differences. In most Class A GPCRs, including CXCR2, agonist binding causes the Y$^{7.53}$ residue to rotate. This rotation occurs alongside minimal movement of P$^{7.50}$, triggering a conformational change that initiates signaling pathways[60,61] (Fig. 4c). In contrast, ORF74 exhibited a 17° tilting movement of Y$^{7.53}$ in conjunction with TM7 towards TM6, rather than a

rotational movement, accompanied by a corresponding movement of P$^{7.50}$ (Fig. 4c). These findings further confirm that ORF74 utilizes a unique VPxxY motif to construct a unique inactive conformation. Y$^{7.53}$ in ORF74 results in a distinctive rotamer angle and significant TM7 movements, both of which deviate from those typically observed in Class A GPCRs.

## Dynamics of inactive-to-active transition of ORF74

To further analyze the conformational transition, we calculated the free-energy landscape of inactive ORF74 from the metadynamics simulations. The free-energy landscape shows four main minima corresponding to the fully inactive (1$^{st}$ population), inactive (2$^{nd}$ and 3$^{rd}$ populations) and intermediate or transiting states (4$^{th}$ population) of ORF74. The inactive state (1$^{st}$, 2$^{nd}$ and 3$^{rd}$ populations), characterized by the lowest free energies, are the most stable conformations, whereas the intermediate states have higher free energy values (Fig. 5a). The inactive state populations display a TM6 conformation closely resembling that observed in the starting inactive structure of ORF74. In these conformations, the intracellular cavities found in active receptors are absent or relatively small. In contrast, the intermediate state (4$^{th}$ population) is associated with a conformational change in TM6 (Fig. 5b), suggesting a transition from the inactive in the direction to the active conformation. We also analyzed the free-energy landscape of active ORF74 (Supplementary Fig. 9a) which follows a distinct path

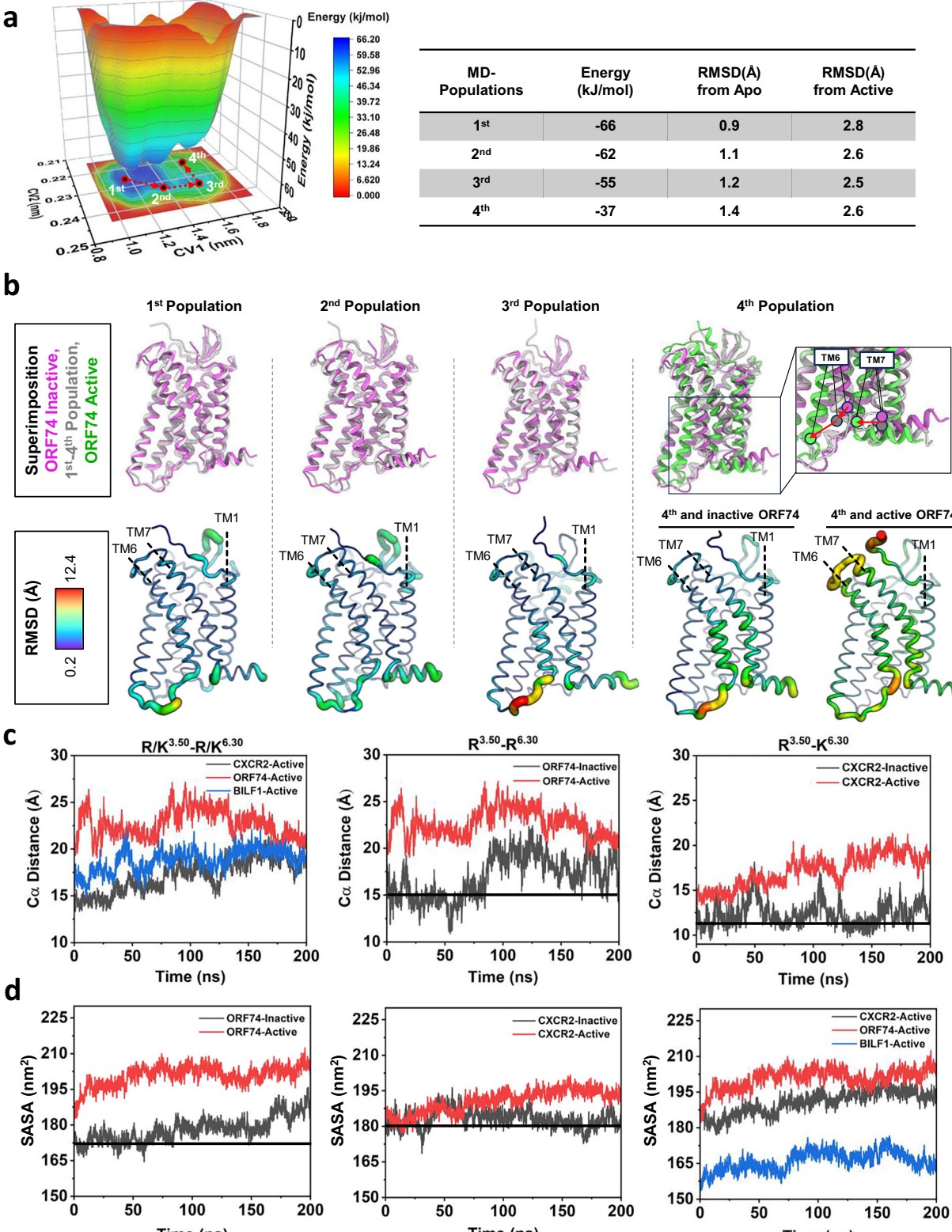

toward lower-energy (fully active) states, likely due to stabilization by both ligand and G protein binding. To ensure a fair comparison with the ligand-free inactive structure, we additionally simulated the active unbound structure of ORF74 i.e., without the ligand and G protein and computed its free energy landscape (Supplementary Fig. 9b). Together, we compared the free energy landscapes of the inactive, active-unbound, and active conformations within a composite depiction (Supplementary Fig. 9c). The active-unbound structure, like the

inactive structure, progresses toward a higher-energy intermediate referred to as transition state 2, in contrast to the progress towards transition state 1 observed from the inactive starting structure. This triad of simulations provides a clearer picture of the energy landscape and supports the presence of a wider ensemble of structures in the inactive and active-like states. As noted above the rotameric conformations of the micro-switches substantially overlap between both the inactive and active states.

**Fig. 5 | Structural transition of ORF74 from an inactive state to active state.**
**a** Free-energy landscape of inactive ORF74 from the metadynamics simulation. The marginal plot shows the projection of the free-energy surface onto the collective variables, CV1 (TM3-TM6 distance) and CV2 (backbone RMSD). Different conformational states are highlighted as 1st, 2nd, 3rd, and 4th. The corresponding energy and RMSD values from the apo/inactive and active conformations are shown.
**b** Comparison of the representative conformations associated with the states with inactive and active structures. The RMSD was visualized for each population in comparison with the inactive ORF74 conformation. The 4th population was subjected to comparative RMSD analysis against both the inactive and active ORF74 conformations, respectively. **c** Monitoring the displacement of TM6 (R/K^6.30) with respect to TM3 (R/K^3.50) by comparing the Cα distance. The first panel illustrates the change in distance over time for active simulations (ORF74, CXCR2 and BILF1), which show similar trends. The second and third panels show comparisons of the Cα distance for active and inactive simulations of ORF74 and CXCR2, further elucidating the dynamic behavior. **d** Comparison of the solvent accessible surface area (SASA) across receptors during simulations of inactive/active ORF74 with inactive/active CXCR2 and active BILF1.

To further support this interpretation, we performed a comparable analysis for CXCR2 and BILF1 (Supplementary Fig. 9d). The active structures of CXCR2 and BILF1 exhibit a trajectory similar to that of active ORF74, progressing toward fully active, lower-energy conformations. In contrast, the inactive CXCR2 transitions toward lower-energy states, which likely correspond to its fully inactive conformation, underscoring its dependence on ligand-induced activation, as previously proposed[40]. However, this contrast highlights the distinct energetic behavior of ORF74, whose inactive form appears capable of exploring higher-energy intermediates that are also accessible from the active-unbound form. These results collectively suggest an ensemble of dynamic structures unique to ORF74.

To further support the movement of TM6 relative to TM3 during activation, we used ionic lock motifs (TM3, R/K^3.50; TM6, R/K^6.30) to compare the Cα distances between TM3 and TM6 of inactive and active ORF74 with those of inactive and active CXCR2 and active BILF1 upon comparative metadynamics simulations. In the simulation of active ORF74, we observed a significant change in the Cα distance between R/K^3.50 and R/K^6.30 compared with the simulations of active BILF1 and CXCR2 (Fig. 5c, first panel). In contrast, the simulation of inactive ORF74 demonstrated a gradual displacement of TM6, leading to an increased Cα distance compared with that of CXCR2 (Fig. 5c, second and third panels). We further compared the solvent accessible surface areas (SASAs) of active and inactive ORF74, CXCR2, and active BILF1 during metadynamics simulations (Fig. 5d). In the simulations of active state ORF74 and CXCR2, we observed increased SASA due to the presence of both intra- and extracellular cavities. Conversely, in active BILF1, SASA was relatively low because only the intracellular cavity was available, with the extracellular cavity being closed (Fig. 5d, third panel). Interestingly, during dynamics, we observed an increase in SASA for inactive ORF74, which was not observed for inactive CXCR2 (Fig. 5d, first and second panels). Since the extracellular cavity or SASA is comparable between the inactive and active forms of ORF74, this increase in SASA for the inactive ORF74 is mostly due to the opening of the intracellular cavity. Comparative Cα distances and SASA across replica simulations of inactive and active ORF74 are shown in Supplementary Fig. 10. Overall, these results suggest that as the simulation progresses, ORF74 inactive state gradually transitions toward an active state, marked by an increasing distance between TM3 and TM6, which opens the intracellular cavity for potential G protein interactions. This finding also suggests that ORF74 is in relatively dynamic equilibrium between inactive and active conformations, allowing for spontaneous inactive-active conformational transitions.

## The ICL2 and ICL3 of ORF74 play a critical role in its high basal signaling activity

The recruitment of G proteins localized in the cellular membrane is critical for GPCR activation. Even when the energy barrier for the transition between active and inactive conformations is low, proper receptor signaling cannot be initiated without facilitating the access of G proteins[55].

Consequently, we analyzed the impact of ORF74 intracellular domain on its high basal signaling activity. We confirmed that the surface of the GPCR-interacting region is negatively charged (Supplementary Fig. 11a). The intracellular domains of GPCRs, including ORF74, CXCR2, US28, and BILF1, typically exhibit a positively charged surface to facilitate charge-charge interactions (Fig. 6a). Amino acid sequence analysis revealed that 58.3% of the residues in the ICL3 of ORF74 are positively charged, which is higher than that of CXCR2 or other vGPCRs known for constitutive activation (CXCR2 = 41.7%, US27 = 16.7%, US28 = 33.3%, BILF1 = 41.7%) (Supplementary Fig. 11b). Additionally, we observed that the ICL2 and ICL3 regions of active-state ORF74 have a relatively strong positive electrostatic surface charge (Fig. 6a). Although the length of ICL2 in ORF74 differs from that of CXCR2 in the active state, both contain an equal number of positively charged residues. The ICL2 of ORF74 contains a proline residue absent in CXCR2, leading to differences in loop curvature. Separately, the presence of a bulky tryptophan side chain in ORF74 also contributes to structural differences. To investigate the impact of these structural features on the GPCR-G$_{\alpha i}$ complex, we performed a structural superposition based on the G$_{\alpha i}$/α5-helix. We observed that G$_{\alpha i}$ bound to ORF74 is tilted towards ICL3 compared to other structures (Supplementary Fig. 11c), a feature similar to that of the ligand-independent constitutively active BILF1[62].

We conducted a bioluminescence resonance energy transfer (BRET2) assay to investigate the role of the intracellular loops ICL2 and ICL3 of ORF74 in G$_{\alpha i}$-mediated signaling. The BRET2 assay is a highly sensitive technique that measures receptor activation by detecting energy transfer between Renilla luciferase-conjugated G$_{\alpha}$ and GFP2-conjugated G$_{\gamma}$ in live cells. Upon GPCR activation, a conformational change induces the dissociation of the G$_{\alpha}$ and G$_{\gamma}$. This spatial separation reduces energy transfer efficiency, leading to a decrease in the BRET signal. To determine the functional significance of ICL2 and ICL3, we generated chimeric constructs where ORF74's ICL2 or ICL3 was replaced with the corresponding loops from CXCR2 (Fig. 6b, c). Both ICL2 and ICL3 alanine mutants exhibited a loss of BRET signal (Supplementary Fig. 12a), indicating that these loops are essential for G protein coupling. Furthermore, the ORF74-ICL2^CXCR2 and ORF74-ICL3^CXCR2 chimeras showed reduced baseline activity compared to wild-type ORF74, with the ICL3^CXCR2 chimera exhibiting a greater reduction than ICL2^CXCR2 (Fig. 6d). This suggests that while both intracellular loops contribute to ORF74's constitutive activity, ICL3 may play a more dominant role in G protein-mediated signaling. Taken together, these findings highlight the critical role of both ICL2 and ICL3 in ORF74's G protein-mediated signaling, with ICL3 serving as a particularly key determinant of basal activity.

We generated reciprocal chimeric constructs where the ICL2 or ICL3 of CXCR2 was replaced with the corresponding loops from ORF74 (CXCR2-ICL2^ORF74 or CXCR2-ICL3^ORF74, Fig. 6b, c). Both chimeras exhibited enhanced baseline activity compared to wild-type CXCR2, with CXCR2-ICL3^ORF74 demonstrating a greater increase than CXCR2-ICL2^ORF74 (Fig. 6d). This reinforces the notion that ICL3 is a stronger determinant of basal activity than ICL2. Interestingly, CXCR2-ICL3^ORF74 retained CXCL1 potency comparable to wild-type CXCR2 while displaying increased baseline activity, whereas CXCR2-ICL2^ORF74 exhibited altered ligand responsiveness (Fig. 6d). This suggests that while both intracellular loops modulate receptor activity, ICL3 plays a distinctive

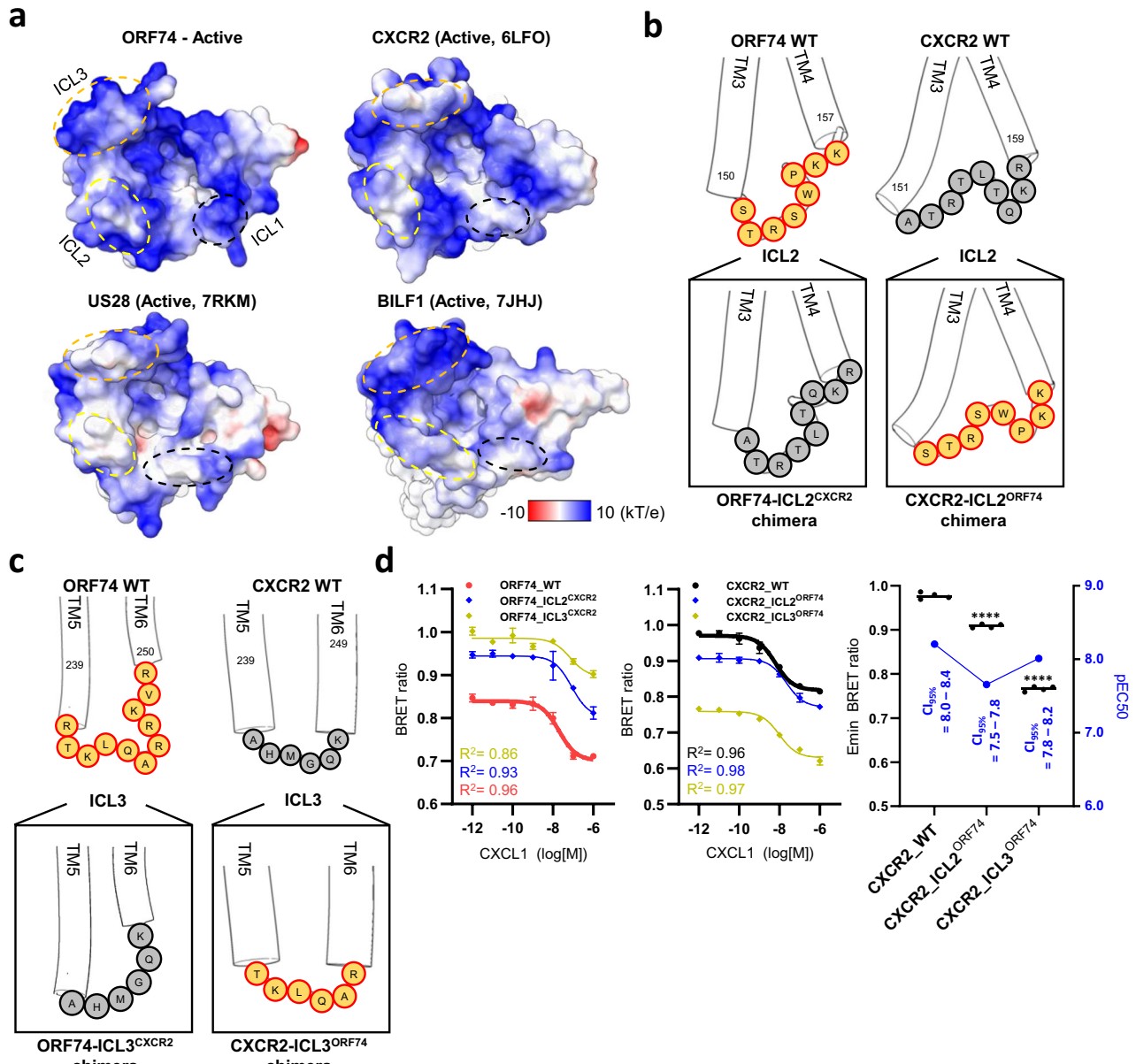

**Fig. 6 | The role of ORF74's ICL2 and ICL3 in its activity. a** Electrostatic potential surface charge analysis of the intracellular domain, including ICL1, ICL2, and ICL3, for the active states of ORF74, CXCR2, US27, and BILF1. Electrostatic potential was calculated at 298.15 K. **b, c** Schematic representation of the creation of chimeric proteins by swapping the ICL2 and 3 regions between ORF74 and CXCR2, respectively. **d** BRET changes in ORF74 and its chimeras (left panel) and CXCR2 and its chimeras (middle panel) in response to dose-dependent CXCL1 stimulation, reflecting activation of $G_{\alpha i}$ protein. The CXCR2 and CXCR2 chimera group analysis highlights the $E_{min}$ BRET ratio and pEC50 values (right panel). The term $CI_{95\%}$ stands for the 95% confidence interval, and $CI_{95\%}$ and pEC50 values were calculated using

GraphPad Prism. Statistical significance for each sample compared to the wild-type and pEC50 values are provided in Supplementary Fig. 12a. The data represents four independent experiments ($n$ = 4), with bars indicating mean ± SEM (****$p$ < 0.0001, ***$p$ < 0.001, **$p$ < 0.01, *$p$ < 0.05, ns > 0.05). The $p$ value for the comparison between CXCR2_WT and CXCR2_ICL2$^{ORF74}$ is 0.0000148, and that for the comparison between CXCR2_WT and CXCR2_ICL3$^{ORF74}$ is 0.0000000188. Significance was determined using ordinary one-way ANOVA. Detailed statistical methods are described in the Methods section and raw data are provided in the Source Data file.

role in stabilizing an active conformation, contributing more significantly to constitutive signaling.

## Discussion

To elucidate the structural determinants underlying the high basal signaling activity of KSHV-encoded ORF74, we determined the cryoEM structures of ORF74 in both ligand-free inactive and CXCL1 & Gi$_{trimer}$-bound active conformations (Fig. 1a–f). We identified an antiparallel dimer in ligand-free ORF74. The structure of each protomer in this dimer closely resembles that of monomeric ORF74 containing a BRIL

insertion in ICL3, suggesting that both represent the same conformational state. Recent single-particle cryoEM analysis has revealed that the class F GPCR Frizzled receptor 7 (Fzd7) adopts an antiparallel dimer conformation in its inactive state within a detergent micelle environment[63]. In contrast, many crystallographic studies have also reported antiparallel orientations of GPCRs[64], although these are generally considered artifacts of crystal packing and not physiologically relevant, as noted in the original review[64]. While the functional significance of such dimer orientations under physiological conditions remains uncertain, recent biophysical studies suggest that antiparallel

GPCR dimers may exhibit distinct functional properties[65,66]. Thus, our high-resolution cryoEM structures of ORF74 should serve as valuable resources for future functional studies of antiparallel-oriented GPCRs.

Comparative analysis revealed a significantly expanded extracellular surface, ECS, on ORF74 relative to that on CXCR2 and BILF1, potentially contributing to its promiscuous ligand binding capacity (Fig. 2a–g). A recent study published the structure of ORF74-Gi$_{trimer}$ (8K4P[67]), and we compared the ECS region between this structure and the CXCL1/Gi$_{trimer}$ bound ORF74. The RMSD between the two structures was 0.6 Å, indicating high similarity. Additionally, the ECS regions exhibited comparable structures, confirming that the shape of the orthosteric binding pocket remains consistent (Supplementary Fig. 7b). Notably, the characteristic structural feature of TM2 was also observed. These findings support the notion that ORF74 maintains a similar ECS conformation regardless of ligand binding and/or Gi$_{trimer}$ association.

D$^{2.50}$ is one of the most highly conserved amino acids (94%) in class A GPCRs[68]. Substituting D$^{2.50}$ with a neutral amino acid reduces the ability of sodium to compete with agonist binding and affects G protein signaling in many GPCRs. In the δ-opioid receptor, a D$^{2.50}$ variant transforms an antagonist into an arrestin-biased agonist[69]. For the human A$_{2A}$ adenosine receptor, replacing D$^{2.50}$ with alanine completely abolished G protein-dependent signaling[70]. This suggests that while ORF74 has the D$^{2.50}$ to S replacement, other structural features may compensate for its change.

ORF74 features a distinctive TM6b region and a notably long ICL3 in its inactive structure. TM6b and ICL3 are interconnected, with this region possessing a VPxxY motif instead of an NPxxY motif (Fig. 3h, i), and a VRY motif in place of the E/DRY motif (Fig. 4a–c). These alterations induce a unique inactive-active equilibrium. The divided TM6, which is involved in ligand binding, enables activation of the TM6b region without ligand interaction. Moreover, modifications of the NPxxY and E/DRY motifs destabilize the inactive structure, facilitating the transition to an active state by TM6b tilting. The extended ICL3 position residue is 6.34 within ICL3 rather than in TM6b, structurally reducing the constraints on R$^{3.50}$ dynamics. These insights into the ORF74 structure and dynamics elucidate how the inactive status of TM6 and TM7 is destabilized, promoting a shift towards an active conformation.

We characterized the unique amino acid sequence and structural features of ORF74, building upon the known micro-switches of canonical class A GPCRs. Structural analysis revealed that, in addition to the distinctive micro-switches of ORF74, the surrounding amino acid residues also display unique structural characteristics. Specifically, the water-ion network-forming CWxP and NPxxY motifs were substituted with C$^{6.48}$ and V$^{7.49}$, respectively, and were further surrounded by noncanonical residues including S$^{2.50}$, D$^{3.39}$, and Y$^{7.45}$ (Figs. 3b, 4a). Moreover, the DRY motif was substituted with V$^{3.49}$ and flanked by R$^{6.34}$ and an extended ICL3 (Fig. 3h). These structural alterations, in combination with the CWxP, NPxxY, and DRY motifs, are likely to contribute to the elevated basal activity observed in ORF74. These findings suggest that the enhanced basal activity of ORF74 is not only attributed to its unique micro-switches but also influenced by the synergistic effects of the surrounding residues.

The ICL2 and ICL3 of ORF74 exhibit a characteristic positively charged surface, which we demonstrated to be critical for G protein-mediated signaling through the CXCR2-ORF74 chimeric constructs (Fig. 6a–d). Notably, ORF74's ICL3 contains 11 amino acids, whereas CXCR2 has only 6, suggesting that loop length may contribute to functional divergence. To investigate the specific role of ORF74's extended ICL3, we generated total five CXCR2_ICL3$^{ORF74}$ variants (including CXCR2_ICL3$^{ORF74-A}$ to ICL3$^{ORF74-D}$, Supplementary Fig. 12b). Among these, the highest baseline activity was observed when only the loop region was replaced with "TKLQAR" amino acids, without modifying the TM5 and TM6 regions of CXCR2. These findings suggest that

the loop sequence itself, rather than the adjacent transmembrane domains, plays a dominant role in G protein activation. However, the mechanism underlying the increased baseline activity of the CXCR2 chimera compared to wild-type remains unclear. One possibility is that the positively charged residues in ORF74's ICL3 facilitate stronger electrostatic interactions with the G$_\alpha$ subunit, thereby promoting high basal activation. Alternatively, the loop's conformational flexibility may stabilize an active-like state even in the absence of ligand binding. Further structural studies, such as NMR or cryoEM analysis of the chimeric receptors, could provide deeper insights into this mechanism.

ORF74 is also known to recruit β-arrestin and undergo endocytic trafficking upon ligand binding[16]. To determine whether ICL2 and ICL3 contribute to this process, we performed a β-arrestin recruitment assay. Unlike the results observed in the BRET assay, none of the chimeric constructs, including CXCR2-ICL2$^{ORF74}$, CXCR2-ICL3$^{ORF74}$, and CXCR2-ICL3$^{ORF74-A}$ to ICL3$^{ORF74-D}$, exhibited enhanced β-arrestin recruitment relative to the wild-type receptor (Supplementary Fig. 13). Furthermore, these results remained consistent before and after normalization to surface expression levels. This suggests that ICL2 and ICL3 play distinct roles in G protein- versus β-arrestin-mediated signaling, and the modifications that increase G protein signaling do not necessarily enhance β-arrestin interaction. Interestingly, the ORF74-ICL3$^{CXCR2}$ chimera exhibited significantly lower β-arrestin recruitment compared to both wild-type ORF74 and the ORF74-ICL2$^{CXCR2}$ chimera (Supplementary Fig. 13a), indicating that ICL3 is a key determinant of β-arrestin interaction. Taken together, our findings highlight the functional bifurcation of ICL3 in GPCR signaling. While it plays a dominant role in G protein-mediated activation, its influence on β-arrestin recruitment appears to be context-dependent. Future studies should investigate whether specific amino acids within ICL3 differentially modulate these pathways, and whether this functional divergence could be exploited for biased ligand development targeting ORF74-mediated oncogenic signaling.

We investigated the distinct functions of ICL2 and ICL3 in ORF74 using CXCR2 chimeras, focusing on differences in activity between CXCR2 wild-type and CXCR2 chimeras, while using ORF74 wild-type and ORF74 chimeras as controls. Comparisons between these CXCR2 and ORF74 groups were not the focus of our analysis, as they fell outside the scope of our study. In the BRET assay, surface expression levels within the ORF74 and CXCR2 groups were statistically comparable (Supplementary Fig. 14b); therefore, normalization to surface expression was not performed. In contrast, in the β-arrestin recruitment assay, differences in surface expression levels were observed within both the ORF74 and CXCR2 groups (Supplementary Fig. 14c). As a result, normalization was performed separately within each group, using their respective wild-type as a reference. Consequently, direct comparisons between ORF74 and CXCR2, including their chimeras, are not appropriate based on our data.

ORF74 preferentially interacts with the G$_{\alpha i}$ subunit, distinguishing it from HCMV-derived US28[71], which prefers the G$_{\alpha q}$ subunit. To investigate this preference, we compared the CXCL1 dose-dependent activity of ORF74 with G$_{\alpha i}$ and G$_{\alpha q}$. ORF74-G$_{\alpha i}$ (pEC50 = 7.7) exhibited higher CXCL1 potency than ORF74-G$_{\alpha q}$ (pEC50 = 6.7) (Supplementary Fig. 14a). These findings support previous reports that most G protein-mediated signaling induced by ORF74 is driven by G$_{\alpha i}$[27].

Despite numerous studies demonstrating ORF74 as a potent oncogene, current clinical trials for KSHV-derived cancers lack investigations into ORF74-targeted therapies. This gap is likely due to a limited understanding of the mechanistic regulation of ORF74, particularly its inactive structure. Here, we present such a molecular insight which will be important for the development of ORF74-targeted therapeutics. The inactive conformation of ORF74 provides a valuable template for structure-based drug discovery aimed at antagonists or inverse agonists. Given the high dynamics of the TM6b and TM7

regions, targeting the extracellular orthosteric binding site as a pharmacophore may be challenging. Instead, the highly positively charged surface of the ICLs, coupled with the exceptionally long ICL3 of ORF74, offers promising avenues for drug development. To further optimize anti-KSHV drug discovery, investigating the intracellular translocation of ORF74 during different phases of the viral life cycle is essential. Additionally, exploring the physiological characteristics of ORF74 in cell type-specific contexts, particularly in virus-infected endothelial cells and B cells, would significantly enhance our understanding and facilitate the development of targeted therapies.

## Methods

### Cell lines and culture conditions

HEK293 (ATCC, CRL-1573) cells were cultured in Dulbecco's Modified Eagle Medium (DMEM, ThermoFisher, 11965) supplemented with fetal bovine serum (FBS, ThermoFisher, 26140) and penicillin/streptomycin (ThermoFisher, 15140) to final 10% and 1%, respectively. HEK293S GnTI⁻ cells (ATCC, CRL-3022) were cultured in FreeStyle 293 Expression Medium (ThermoFisher, 12338018) supplemented with FBS and penicillin/streptomycin to final 1% each and shaken at 130 rpm. The cells were maintained at 37 °C with 5% $CO_2$. ExpiSf9 cells (ThermoFisher, A35243) were cultured with ESF 921 Insect Cell Culture Medium (Expression Systems, 96-001). Hi5 cells (BTI-Tn-5B1-4, ThermoFisher, B85502) also cultured with ESF 921 medium were infected with baculoviruses produced from ExpiSf9 cells for protein expression. Both insect cells were maintained at 27 °C with shaking at 125 rpm.

### Molecular design of ORF74 and CXCL1 expression constructs

THe C-terminal residue of KSHV ORF74 (UniProt Q98146) was truncated and mutated to carry L169W and L258V stabilizing mutations to enhance protein expression. An additional mutation of G30C was introduced for disulfide trapping to solve the structure of the CXCL1-ORF74-Gi$_{trimer}$-scFv16 complex. The construct was fused with an N-terminal hemagglutinin signal peptide (HA). The construct was cloned into a baculovirus donor plasmid, pEZT-BM (Addgene, 74099), with a C-terminal Flag tag (for CXCL1-ORF74-Gi$_{trimer}$-scFv16) or 10xHis tag followed by a 3C protease cleavage site and a Flag tag (for apo ORF74 and ORF74-BRIL). ORF74-BRIL construct was generated by a fusion of BRIL (an engineered variant of apocytochrome b562a) with ICL3 of ORF74 between Lys242-Leu243 with SA linkers. Full-length human CXCL1 (UniProt P09341, 1-107) was introduced a point-mutation of N56C for disulfide trapping with ORF74 and C-terminal fusion of a 3C protease cleavage site followed by a 10xHis tag for cloning into pEZT-BM.

### Purification of ORF74

P0 baculovirus was prepared in ExpiSf9 cells using the Bac-to-Bac baculovirus expression system (ThermoFisher, 10359016), and the virus was amplified from P0 to P2 according to the manufacturer's guidelines. 10% (v/v) P2 virus was added to HEK293S GnTI⁻ cells at a density of 2-3 × 10⁶ cells/ml and culture flasks were shaken at 5% $CO_2$ and 37 °C for 24 h. After 24 h, valproic acid (Sigma-Aldrich) was added to the final 3 mM concentration and further shaken for 48 h before centrifugation. The harvested cells were lysed with a dounce homogenizer (Thomas Scientific, 20A00B426) in LS buffer (10 mM HEPES pH 7.5, 10 mM MgCl₂, 20 mM KCl) supplemented with an in-house protease inhibitor cocktail (0.2 mM AEBSF, 4.2 μM Leupeptin, 2.9 μM Pepstatin A, 1.5 μM Aprotinin) and 5 U/ml benzonase. The lysate was centrifuged at 180,000 × g for 30 min. The pellet was resuspended in LS buffer and again subjected to homogenization and centrifugation. The process was repeated three more times with HS buffer (10 mM HEPES pH 7.5, 10 mM MgCl₂, 20 mM KCl, 1 M NaCl) to isolate membrane fraction. The membrane pellet was resuspended in 100 mL LS buffer supplemented with 40% glycerol and stored at −80 °C. Thawed membranes were rotated with 2 mg/ml iodoacetamide at 4 °C for

30 min. The membrane fractions then were nutated for 3 h in 100 mL of 2× solubilization buffer (50 mM HEPES pH 7.5, 800 mM NaCl, 2% (w/v) lauryl maltose neopentyl glycol (LMNG, Anatrace, NG310), 0.2% (w/v) cholesteryl hemisuccinate (CHS, Anatrace, CH210), and the in-house protease inhibitor cocktail.

After centrifugation at 180,000 × g for 30 min, 20 mM imidazole and 10 mL Ni-NTA resin (QIAGEN, 30210) was added to the supernatant, and the mixture was rotated at 4 °C for 2 h. The resin was then collected in a column and washed with 10 column volume (CV) of W1 buffer (50 mM HEPES pH 7.5, 400 mM NaCl, 25 mM imidazole, 0.1% (w/v) LMNG, 0.01% (w/v) CHS, 10% glycerol (v/v)) followed by 5 CV of W2 buffer (50 mM HEPES pH 7.5, 150 mM NaCl, 25 mM imidazole, 0.01% (w/v) LMNG, 0.001% (w/v) CHS, 10% glycerol (v/v)) and eluted with elution buffer (50 mM HEPES pH 7.5, 150 mM NaCl, 250 mM imidazole, 0.005% (w/v) LMNG, 0.0005% (w/v) CHS, 10% glycerol (v/v)). The eluate was concentrated using 100 molecular weight cut-off (MWCO) Amicon ultra centrifugal filter (Sigma-Aldrich, UFC9100) to 0.5 mL and purified over Superdex 200 10/300 GL size-exclusion column (Cytiva, 28990944) equilibrated with 25 mM HEPES pH 7.5, 150 mM NaCl, 0.005% (w/v) LMNG, 0.0005% (w/v) CHS, 10% glycerol (v/v). Peak fractions were concentrated, aliquoted, and kept at −80 °C until further use. For purification of the CXCL1-ORF74 complex, 293S GnTI⁻ cells were co-infected with P2 viruses of ORF74 and CXCL1, with MOI of 6 and 4, respectively. The rest of the purification steps were identical as described above.

### Purification of the G$_{\alpha i}$ heterotrimer

G$_{\alpha i}$, G$_{\beta 1}$ (with an N-terminal 3C protease-cleavable His tag), and G$_{\gamma 2}$ were co-expressed in Hi5 cells using the baculovirus system[62]. After expression, cell pellets were harvested and lysed using a dounce homogenizer in lysis buffer containing 50 mM HEPES (pH 8.0), 0.1 mM MgCl₂, 5 mM 2-mercaptoethanol, and 50 μM GDP, supplemented with a protease inhibitor cocktail (25 mL per liter of cell pellet, Roche 11836170001). The lysate was centrifuged at 180,000 × g for 30 min, and the resulting pellets were resuspended in 100 mL of resuspension buffer (20 mM HEPES pH 8.0, 100 mM NaCl, 5 mM MgCl₂, 5 mM imidazole, 5 mM 2-mercaptoethanol, 10 μM GDP) using a dounce homogenizer. The resuspended mixture was then incubated with 100 mL of 2× solubilization buffer (20 mM HEPES pH 8.0, 100 mM NaCl, 5 mM MgCl₂, 5 mM imidazole, 2% sodium cholate, 0.1% n-Dodecyl-β-D-Maltoside (DDM, Anatrace, D310), 4 μL calf intestinal alkaline phosphatase (CIP, NEB M0525), 5 mM 2-mercaptoethanol, 10 μM GDP) under nutation for 1 h.

Following solubilization, the mixture was centrifuged again at 180,000 × g for 30 min, and the supernatant was incubated with Ni-NTA resin at 4 °C for 2 h with rotation. The resin was then collected in a column and subjected to sequential washing: 10 column volumes (CV) of W1 buffer (20 mM HEPES pH 8.0, 300 mM NaCl, 1 mM MgCl₂, 5 mM imidazole, 0.2% sodium cholate, 0.05% DDM, 5 mM 2-mercaptoethanol, 10 μM GDP), followed by 5 CV of W2 buffer (20 mM HEPES pH 8.0, 100 mM NaCl, 1 mM MgCl₂, 5 mM imidazole, 0.05% DDM, 0.1 mM TCEP, 10 μM GDP). Proteins were then eluted using elution buffer (20 mM HEPES pH 8.0, 100 mM NaCl, 1 mM MgCl₂, 200 mM imidazole, 0.05% DDM, 0.1 mM TCEP, 10 μM GDP).

The eluted protein was dialyzed overnight at 4 °C against dialysis buffer (20 mM HEPES pH 8.0, 100 mM NaCl, 1 mM MgCl₂, 5 mM imidazole, 1 mM MnCl₂, 0.1% LMNG, 0.01% CHS, 0.1 mM TCEP, 10 μM GDP) containing 1:500 3C protease (Thermo Fisher, 88946), 20,000 U lambda protein phosphatase (NEB, P0753), 10 U CIP, and 5 U Antarctic Phosphatase (NEB, M0289) per mL of dialysis mixture. The following day, the reaction was applied to fresh Ni-NTA resin to remove the cleaved His tag and His-tagged 3C protease. The flow-through fraction was concentrated to 20 mg/mL, aliquoted, and flash-frozen with 10% (v/v) glycerol. The purified protein aliquots were stored at −80 °C until further use.

## Purification of the BAK5 Fab

BAK5 Fab was purified[72] with slight modifications. Briefly, BL21 cells were transformed with Fab-encoding expression vectors (a gift from Dr. Anthony Kossiakoff) and grown in Terrific Broth (TB) medium supplemented with 100 µg/mL ampicillin (Sigma, A9518) and 1 mM IPTG (Goldbio, I2481C). Cultures were incubated at 37 °C for 4–5 h, after which the cells were harvested by centrifugation. The resulting cell pellets were resuspended in 20 mM HEPES (pH 7.5), 150 mM NaCl, supplemented with 1 mM PMSF and 5 U/mL benzonase, then lysed by ultrasonication. The lysate was incubated at 65 °C for 30 minutes to facilitate heat denaturation of unwanted proteins. After heat treatment, the lysate was cleared by centrifugation, filtered through a 0.22-µm filter, and loaded onto a 5 mL HiTrap Protein L column (Cytiva, 17547815) pre-equilibrated with 20 mM HEPES (pH 7.5) and 400 mM NaCl. The column was extensively washed with the equilibration buffer, and Fabs were eluted using 0.1 M acetic acid. The eluted protein was immediately loaded onto a 5 mL Resource S column (Cytiva, 17118001) pre-equilibrated with 50 mM sodium acetate (NaOAc, pH 5.0). The column was washed with the equilibration buffer, and Fabs were eluted using a linear gradient from 0 to 100% of 50 mM NaOAc (pH 5.0) containing 2 M NaCl. The purified Fabs were then dialyzed overnight against 20 mM HEPES (pH 7.5) and 150 mM NaCl to remove excess salts and buffer components.

**Purification of the BAK5 Fab-binding nanobody.** The pET-26b vector carrying the Fab-binding nanobody (a gift from Dr. Anthony Kossiakoff) was expressed in E. coli BL21 (DE3) grown in Terrific Broth (TB) medium supplemented with 1 mM MgCl$_2$ and 0.2% glucose. Cells were cultured at 37 °C until mid-log phase, then induced with 1 mM IPTG and incubated for 16 h at 30 °C. Nanobodies were purified using Ni-NTA chromatography and subsequently dialyzed overnight in a buffer containing 20 mM HEPES (pH 7.5), 150 mM NaCl, 5 mM imidazole, and TEV protease (1:500) to remove the histidine tag. The following day, the reaction mixture was applied to fresh Ni-NTA resin to separate the cleaved tag and His-tagged TEV protease. The nanobody-containing eluate was further purified using a Superdex 200 10/300 GL size-exclusion column equilibrated with 25 mM HEPES (pH 7.5) and 150 mM NaCl. Peak fractions were collected, and 25% glycerol was added before concentrating the purified nanobody. The concentrated protein was aliquoted, flash-frozen, and stored at −80 °C for future use.

**Nanodisc reconstitution of ORF74.** Purified ORF74 was mixed with a lipid mixture (POPC:POPE:POPG = 3:1:1, Anatrace, P516-P416-P616) at a 1:200 ratio and incubated at 4 °C with rotation for 30 min. NW11 (Now Scientific, CNW11-11) was then added at a 1:3 ratio (receptor:NW11) and rotated at 4 °C for another 30 min. To initiate reconstitution, Bio-Beads SM2 (100 mg per 1 ml mixture) were added, and the mixture was incubated at 4 °C for 30 min. The Bio-Beads were subsequently removed, and fresh Bio-Beads SM2 were added to the reconstitution mixture for overnight rotation. After incubation, the Bio-Beads were removed, and the reconstitution mixture was loaded onto a Superdex 200 10/300 GL size-exclusion column, pre-equilibrated with 20 mM HEPES (pH 6.9), 50 mM NaCl, 5 mM CaCl$_2$, and 0.5 mM TCEP. The fraction corresponding to ORF74 reconstituted in lipid nanodiscs was collected.

## Formation and purification of the CXCL1-ORF74-Gi$_{trimer}$-scFv16 complex

Molar excesses of 1.2 and 1.4 of purified Gi$_{trimer}$ and scFv16 (Thermo-Fisher, 703976), respectively, were added to the SEC elution containing the CXCL1-ORF74 complex. The buffer composition was then adjusted to include 1% (w/v) LMNG, 0.1% (w/v) CHS, 0.1% (w/v) GDN, and 1 mM MnCl$_2$, supplemented with 2 mg/mL iodoacetamide (ThermoFisher, 35603). The mixture was incubated at room temperature for

1 h, followed by the addition of 5 mL lambda protein phosphatase to the reaction.

After incubation on ice for an additional 1 hour, 5 µL apyrase (NEB, M0398) was added, and the reaction mixture was incubated overnight at 4 °C to hydrolyze GDP. The assembled complex was then subjected to an additional pull-down step using Ni-NTA resin to remove excess G protein, scFv16, and impurities. The reaction was diluted 10-fold with G protein coupling buffer, supplemented with 5 mM imidazole, and incubated under nutation at 4 °C for 2 h. The resin was then sequentially washed with 10 column volumes (CV) of wash buffer (10 mM HEPES pH 7.5, 150 mM NaCl, 0.01% (w/v) LMNG, 0.001% (w/v) CHS, 0.01% (w/v) GDN, 5 mM imidazole).

The CXCL1-ORF74-Gi$_{trimer}$-scFv16 complex was eluted with 5 CV of elution buffer (10 mM HEPES pH 7.5, 150 mM NaCl, 200 mM imidazole, 0.001% (w/v) LMNG, 0.0001% (w/v) CHS, 0.001% (w/v) GDN). The eluted complex was concentrated to 0.5 mL using a 100 kDa molecular weight cutoff amicon filter and further purified via Superdex 200 10/ 300 GL size-exclusion chromatography equilibrated with 10 mM HEPES pH 7.5, 150 mM NaCl, 0.001% (w/v) LMNG, 0.0001% (w/v) CHS, 0.001% (w/v) GDN. Peak fractions were collected, concentrated, and prepared for cryoEM analysis.

**Formation and purification of the ORF74-BRIL-BAK5 Fab-Nb complex.** Molar excesses of 1.2 and 1.5 of purified BAK5 Fab and anti-Fab nanobody, respectively, were added to the SEC elution containing ORF74-BRIL. The mixture was incubated on ice for 30 min and purified over Superdex 200 10/300 GL size-exclusion column equilibrated with 10 mM HEPES pH 7.5, 150 mM NaCl, 0.005% (w/v) LMNG, and 0.0005% (w/v) CHS. The peak fractions corresponding to ORF74-BRIL-BAK5 Fab-Nb were collected and concentrated for cryoEM study.

**β-Arrestin2 recruitment assay (PRESTO-Tango assay).** The HTLA cell line, HEK293 cells that stably express a tTA-dependent luciferase reporter and a β-arrestin2-TEV fusion gene, was kindly provided by the laboratory of Dr. Brian Roth. Cells were maintained in DMEM supplemented with 10% FBS, 100 U/mL penicillin, 100 µg/mL streptomycin, 2 µg/mL puromycin, and 100 µg/mL hygromycin B. The Tango plasmid (Addgene kit 1000000068) was used to clone the necessary receptor constructs, which were subsequently inserted into the Tango expression system. Cells were transfected with TransIT-2020 transfection reagent (Mirus Bio, MIR5400) in 6-well plates and incubated overnight. The following day, cells were detached using a trypsin-free dissociation reagent and plated into poly-L-lysine-coated white clear-bottom 96-well plates (Thermo Scientific, 165306) at a density of 80,000–100,000 cells per well in FreeStyle 293 expression media (Gibco, 12338018). After 24 h of incubation, cells were treated with a dose-dependent concentration of CXCL1 and incubated for an additional 24 h. The next day, Bright-Glo solution (Promega, E2610) was added to each well, and the plate was incubated at room temperature for 15 min. Luminescence was measured using a FilterMax F5 plate reader (Molecular Devices). Data were analyzed using GraphPad Prism (version 10.1.2) and expressed as relative luminescence units (RLUs).

## Surface expression analysis by ELISA

HEK293 cells were seeded in transparent poly-D-lysine-pre-coated 96-well plates and transfected as described for the Tango assay. 24 h after transfection, cells were fixed with 4% paraformaldehyde in Tris-buffered saline (TBS) for 5 min in room temperature, blocked with blocking buffer (0.1 M NaHCO$_3$ pH 8.6 in 1% (w/v) skim milk) for 4 h at 22 °C. Receptors were detected by incubation with monoclonal M2 anti-FLAG antibody (Sigma-Aldrich, F3165) at 1:3000 in 1% (w/v) BSA in TBS overnight at 4 °C. After being washed, the cells were incubated with a secondary goat anti-mouse horseradish peroxidase-conjugated IgG antibody (Sigma-Aldrich, AP308P) at 1:5000 in a blocking buffer for 2 h at 22 °C. Peroxidase activity was detected by 3,3′,5,5′-

tetramethylbenzidine substrate (TMB, Sigma-Aldrich, T0440) for 30 min and the reaction was stopped with 50 µL of 1 M $H_2SO_4$. Absorbance at 450 nm was measured using the Filtermax F5 plate reader.

### G protein coupling assay (TRUPATH assay)

*Renilla* luciferase 8 (Rluc8)-fused $G_\alpha$ subunit and green fluorescent protein 2 (GFP2)-fused $G_\gamma$ subunit constructs were obtained from Addgene (Kit #1000000163) following the TRUPATH protocol[73]. For the experimental setup, HEK293T cells (ATCC, CRL-3216) were transfected in 6-well plates (Corning, 3516) using TransIT-2020 reagent (Mirus Bio, MIR5400) in 10% FBS-containing DMEM (Gibco, 12491015). The transfection mixture included vectors for the target GPCR, as well as those encoding the Rluc8-$G_\alpha$ subunit, $G_\beta$ subunit, and GFP2-$G_\gamma$ subunit. After 24 h of transfection, cells were transferred to white, opaque 96-well plates (Corning, 3917), followed by a medium change to FreeStyle 293 expression medium (Gibco, 12338018). After an additional 24-h incubation, BRET signal measurements were performed. Recombinant CXCL1 (Biotechne, 275-GR) was used as the ligand for the assay. Coelenterazine 400a (Goldbio, C-320-1) was dissolved in ethanol and added to each well to a final concentration of 50 µM, ensuring uniform distribution. The BRET signal was measured using an EnVision 2105 plate reader (PerkinElmer, 2105-0020) equipped with a BRET2 dual mirror (Revvity, 2100-4150). Data were analyzed using GraphPad Prism (version 10.1.2).

### Surface expression analysis by FACS

N-terminus FLAG tag-containing GPCR-expressing cells were harvested and washed in FACS buffer (DPBS supplemented with 2% BSA). The cells were treated with an APC-conjugated anti-FLAG antibody (Abcam, ab72569) for 30 min at 4 °C. After staining, the cells were washed 3 times and resuspended in FACS buffer before being applied to a BD FACSCelesta (BD Biosciences). The results were analyzed with FlowJo v10 software (BD Biosciences).

For surface expression analysis by ELISA and FACS, as well as for the G protein coupling assay (TRUPATH assay) and β-arrestin2 recruitment assay (PRESTO-Tango assay), we were blinded to sample identity during both data acquisition and analysis. Group information was disclosed only after data processing had been completed.

### Statistical analysis

All data are presented as mean ± SEM. Statistical significance among multiple groups was assessed using Ordinary one-way ANOVA with GraphPad Prism's built-in multiple comparisons test (version 10.1.2). Comparisons were performed using ORF74 WT or CXCR2 WT as the reference group, depending on the experimental condition. Differences were considered statistically significant at $p < 0.05$ (****$p < 0.0001$, ***$p < 0.001$, **$p < 0.01$, *$p < 0.05$, ns >0.05). Sample sizes ranged from $n = 2$ to $n = 8$, depending on assay type and variability. Specific n values for each experiment are indicated in the respective figure legends.

**CryoEM sample preparation and data acquisition.** All three protein samples: (1) ORF74 reconstituted in nanodisc, (2) CXCL1-ORF74-$Gi_{trimer}$-scFv16 in LMNG + CHS + GDN, and 3) ORF74-BRIL-BAK5 Fab-Nb in LMNG + CHS containing HEPES pH 7.5 buffer were concentrated to ~10 mg/ml for cryoEM sample preparation. The concentrated protein samples were further mixed with 0.02% Octyl-beta-Glucoside (OG) before sample freezing on cryoEM grids. In each case, 3.5 µL of protein sample was used per a 300-mesh Quantifoil R1.2/1.3 Cu grid that had been pre-treated with glow-discharge at 35 mA for 30 s. Any surplus liquid was removed by blotting with filter paper using a Vitrobot Mark IV machine (Thermo Fisher) that was set to a blotting force of −5 and a blotting time of 2 s. Subsequently, the grid was rapidly frozen by plunging it into liquid ethane.

CryoEM data acquisition was performed using serialEM on a Titan Krios microscope (Thermo Fisher), equipped with a Gatan Bio-Quantum K3 imaging filter and camera. A 10 eV slit was utilized with the filter. Imaging was conducted at a magnification of 130,000X, yielding a pixel size of 0.33 Å/pix in the camera's super-resolution mode. The defocus range was set between −0.8 µm and −1.5 µm. Each exposure received a total dose of 50 e−/Å2, distributed across 50 frames. The movie stacks generated were saved onto a local drive along with the gain reference file for further use during data processing.

**CryoEM data processing.** The CryoEM data processing for all three protein samples was performed using CryoSPARC[74]. The overall data processing approach was similar in each case at least till the 2D classification step. Briefly, the movie stacks from the data collection step were accessed and utilized for motion correction, with the initial two frames of each stack being omitted. Following this, frame averaging was conducted with dose weighting and 2X binning, yielding micrographs with a pixel size of 0.66 Å/pix. Subsequently, CTF estimation and correction were applied to these micrographs before using them in iterative rounds of 2D and 3D classifications to isolate the highest quality particles for final 3D reconstruction. Initial 3D models from the ab-initio reconstructions were further classified and refined simultaneously through heterogeneous refinement. Best-looking classes with obvious features with respect to the protein sample of interest were chosen for further refinement and post-processing steps.

In the case of the ORF74 Apo dimer and the ORF74-BRIL-BAK5 Fab-Nb structures, further Non-Uniform refinements led to high quality 2.8 Å, and 3.8 Å resolution cryoEM density maps, respectively. For the CXCL1-ORF74-$Gi_{trimer}$-scFv16 structure, a 3D-flexible reconstruction and refinement with a static mask based on the NU-refinement 3D volume enabled a similar high-quality 3 Å resolution cryoEM density map. To further improve the resolution of the CXCL1-ORF74 interface, we selectively masked and local refined the CXCL1-ORF74 region, then integrated it with the 3DFlex map using the 'vop_maximum' command in ChimeraX, ensuring proper scaling. Separate local refinement of the $Gi_{trimer}$ region was not performed as it was already well-resolved in the 3DFlex map. Interpretability of both the 3DFlex map and CXCL1-ORF74 locally refined maps was enhanced by post processing with DeepEMhancer[75] (using half maps as input and applying default parameters) before combining them to generate the composite map, which was used for model building and visualization purposes. Comprehensive statistical information regarding all the cryoEM datasets and structures are available in the Supplementary Table 1.

**Model building and refinement for cryoEM structures.** AlphaFold2 was used for predicting the structures of ORF74 and CXCL1 utilizing their respective amino acid sequences from the constructs used in this study. The ORF74 atomic model generated was used for initial residue registration both in the ORF74 Apo dimer and the CXCL1-ORF74-$Gi_{trimer}$-scFv16 cryoEM maps, inside UCSF chimera. Similarly, the CXCL1 monomer model was utilized for the initial residue registration of CXCL1 in the CXCL1-ORF74-$Gi_{trimer}$-scFv16 structure. For initial fitting of the $Gi_{trimer}$-scFv16 part in the CXCL1-ORF74-$Gi_{trimer}$-scFv16 structure, its atomic model was retrieved from the PDB code 7JHJ. With all atomic models fitted, successive refinement iterations were carried out for each map-model combination, first involving manual fine-tuning of the models using Coot, followed by automated real-space refinement in Phenix until the validation parameters were reasonably well satisfied Supplementary Table 1.

### System setup for metadynamics simulation

For ORF74, we first modeled two distinct systems to conduct metadynamics simulations. The first system commenced with the inactive

conformation of ORF74, while the second system initiated with the active conformation of ORF74 complexed with the CXCL1 ligand. In the latter system, similar to other studies[76,77], we included only a portion of the Gαi subunit (specifically, the C-terminus helix spanning from K330 to F354) at the intracellular side of ORF74 to mimic the interaction of the Gα subunit with the receptor. We also simulated a third system starting from the active-unbound conformation of ORF74 for direct comparison with the inactive structure. Data from the active-unbound ORF74 simulations were used solely for comparison of the overall free-energy landscape. We selected CXCR2 for a more robust comparison, given its 30% sequence identity with ORF74 and the availability of both active and inactive structures in the PDB database[40]. The missing intracellular and extracellular loops of the inactive CXCR2 (PDB ID: 6LFL) were modeled using MODELLER[78]. Additionally, we extended a few residues at the N-terminus (A36-E45) to model the conserved disulfide linkage between the N-terminus and ECL3 using PyMOL. For the simulation of the active form of CXCR2, we selected the conformation with bound ligand CXCL8 at the extracellular site and at the intracellular site only a portion of the Gαi subunit (K330 to F354) from the crystal structure (PDB ID: 6LFO), mirroring our approach with ORF74. In addition, we opted to investigate another viral GPCR, BILF1, which shares a sequence identity of 24% with ORF74. Notably, BILF1 is recognized for its capacity for self-activation without a ligand, and the cryoEM structure of this self-activated receptor is accessible[62]. To initiate our analysis, we constructed the starting structure based on the cryoEM structure (PDB ID: 7JHJ), retaining only a portion of the Gαi subunit (K330 to F354) to similarly intact with the receptor, as modeled above for ORF74.

## Metadynamics simulation
In all the five systems, the GPCR was embedded within a POPC bilayer composed of 200 POPC molecules and solvated in a solvent environment modeled using the TIP3P water model. We added Na/Cl ions to achieve charge neutrality as well as a 150 mM NaCl buffer using the CHARMM-GUI interface[79]. The inactive systems of ORF74 and CXCR2 comprised ~86,000 atoms, while the active systems of ORF74 and CXCR2 consisted of ~108,000 atoms. Additionally, the active system of BILF1 comprised 90,055 atoms and the active-unbound ORF74 system has ~94,000 atoms. Molecular dynamics simulations were performed using GROMACS 2021.5 with the CHARMM36m force field. Additionally, PLUMED 2.8.0 was employed to enable Well-Tempered Metadynamics simulations for both systems[80–82]. The long-range electrostatic interactions were managed using the particle mesh Ewald (PME) method, while hydrogen-containing covalent bonds were constrained using the SHAKE algorithm. The cut-off distance for non-bonded interactions was set to 1.2 nm, and the neighbor list was updated every 20 steps. The system temperature and pressure were controlled by Nosé–Hoover thermostat and Parrinello-Rahman type barostat. Each system underwent energy minimization through the steepest descent algorithm, consisting of 5000 steps. System equilibration followed a sequential process of six steps, wherein the harmonic restraints on proteins and lipids were gradually reduced. The initial two equilibration steps were conducted within the NVT ensemble, while the subsequent four steps occurred under the NPT ensemble to maintain an isobaric-isothermal condition. For both systems, a pair of collective variables (CVs) were employed: CV1 represented the distance between the center of mass of the TM3b and TM6b regions, while CV2 denoted the backbone RMSD of the receptor. During simulations, hills were deposited every 500 integration steps with an initial height of 0.2 kcal/mol and a bias factor of 5. The metadynamics simulations were concluded upon thorough exploration of the relevant CV space. Each system was simulated for 200 ns. Quadruplicate simulations were conducted for all the systems including active/inactive pair for ORF74,

CXCR2 and only the active form for BILF1, resulting in a total of 800 ns of aggregate sampling for each system, giving a total simulation time of 4 μs.

## Simulation data analysis
Trajectory analysis was performed using the built-in modules of GROMACS[83]. Subsequently, the obtained data were visualized and plotted using Origin2022. We used MetadynView[84] to visualize a free energy plot for inactive ORF74[83].

## Reporting summary
Further information on research design is available in the Nature Portfolio Reporting Summary linked to this article.

## Data availability
The cryoEM maps have been deposited in the Electron Microscopy Data Bank (EMDB) under accession codes EMD-43717 (KSHV ORF74 apo dimer); EMD-43720 (KSHV ORF74-BRIL-BAK5-Nb complex); EMD-48100 (CXCL1-KSHV ORF74-Gi_trimer-scFv16 complex); EMD-43718 (NU-refined consensus map of CXCL1-KSHV ORF74-Gi_trimer-scFv16 complex); EMD-48095 (3DFlex refined map of CXCL1-KSHV ORF74-Gi_trimer-scFv16 Complex); EMD-48097 (Local refined cryoEM map of CXCL1-KSHV ORF74 region). The atomic coordinates have been deposited in the Protein Data Bank (PDB) under accession codes 8W1A (KSHV ORF74 apo dimer); 9EJC (CXCL1-KSHV ORF74-Gi_trimer-scFv16 complex). The source data underlying Fig. 6d and Supplementary Figs. 1b–d, 12a, b, 13a, b, 14a–c are provided as a Source Data file. Starting and final conformations from MD simulations are provided in the Source Data file. Source data are provided with this paper.

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

## Acknowledgements

We thank Dr. Brian Roth for providing the HTLA cell line for the Tango assay and Dr. Anthony Kossiakoff for the anti-BRIL antibody and anti-Fab nanobody plasmids. We are also grateful to Dr. Brian Kobilka and Dr. Raymond Stevens for their guidance on the biochemical strategies used in this study, including construct design, overexpression, and purification of ORF74. Additionally, we thank Dr. Tracy M. Handel for insightful discussions on cysteine disulfide trapping engineering approaches. The authors acknowledge the use of the CryoEM Core Facility and the Core Facility for Advanced Research Computing at Case Western Reserve University (CWRU), as well as funding support from the National Institutes of Health (CA251275 to J.J.; GM151043 to X.D.; AG084065, EY029169, and AG089561 to M.B.).

## Author contributions

J.U.J. conceived and conceptualized the project. J.U.J., X.D. and M.B. managed the resources and supervised the research. J.B.P., B.S., A.R.S., D.K., H.D.S., J.B., M.J.K., X.D., M.B. and J.U.J. designed the experiments. J.B.P., D.K., H.D.S., J.B., S.S. and B.S. optimized and performed protein expression, purifications as well as all the functional assays. B.S. and X.D. prepared cryoEM samples, conducted screening, collected and processed the cryoEM data, determined the structures and built atomic models. A.R.S. and M.B. planned, executed and analyzed all the MD simulation-related data. J.B.P., B.S., A.R.S., D.K., H.D.S. and S.S. analyzed, visualized and organized the overall data. J.B.P., B.S., A.R.S., D.K., H.D.S. prepared the original draft. J.U.J., X.D. and M.B. reviewed, edited and finalized the manuscript. All the authors reviewed and approved the paper.

## Competing interests

The authors declare no competing interests.

## Additional information

**Supplementary information** The online version contains
supplementary material available at

Matthias Buck, Xinghong Dai or Jae U. Jung.

**Peer review information** *Nature Communications* thanks Xiaowu Dong,
Naotaka Tsutsumi who co-reviewed with Shota Suzukiand the other
anonymous reviewer(s) for their contribution to the peer review of this
work. A peer review file is available.

**Publisher's note** Springer Nature remains neutral with regard to
jurisdictional claims in published maps and institutional affiliations.

