## [Transparent Peer review file · Nature Communications]

Structural basis for ligand promiscuity and high signaling activity of Kaposi's Sarcoma-associated Herpesvirus-encoded GPCR

Corresponding Author: Professor Jae Jung

Version 0:

Reviewer comments:

Reviewer #1

(Remarks to the Author)

In this article, Park et al. present the structures of the KSHV GPCR, ORF74, in its active and inactive states. The structures describe ORF74's unique high basal activity and recognition of its full agonist CXCL1. Although the CXCL1-ORF74-Gi and ORF74-Gi structures have been posted to bioRxiv recently (Liu et al. 2023), it is not yet formally published, and the addition of the inactive, apo ORF74 structures is remarkable as it may facilitate the design of molecules that shutdown ORF74's pathological activity. There is still an outstanding question if a significant ORF74 population is inactive in complicated, physiological conditions on cells, but the authors demonstrated ORF74 can be in an inactive conformation in multiple experimental settings here, in lipidic nanodisc (anti-parallel dimer) and detergent (BRIL-fused monomer) with several thermostabilized mutations in cryo-EM study, and in lipidic environment in MD simulations. The data suggested a mechanism of ORF74's high basal activity, where the accumulated 7TM mutations lower the energy barrier between inactive and active conformations. Note that we are not able to provide high-quality comments on the technical details of the MD simulations in the manuscript, which appear solid, as non-expert.

The authors further attempted to gain mechanistic insights into ORF74 activation based on the signaling assays for structure-based chimeras. The study would be an important addition to the community, but several major concerns should be addressed.

Major points

1. Map reconstruction and modeling

CXCL1-ORF74-Gi

The gold-standard FSC curve for the CXCL1-ORF74-Gi map does not fall to zero at high resolution, indicating that the final dataset for reconstruction includes substantial duplicated particles, and the map resolution is overestimated. For this map, it is also unusual that the peripheral region has the highest resolution (~2.5 Å), and the core region has a lower resolution (~4 Å). The authors need to remove the duplicated particles to re-refine and re-analyze the map. Because of the requirement for map re-refinement, we will request the new maps and models for assessment in the following review round.

2. Signaling

The data is interesting, but it is confusing as to why the authors tested β -arrestin signaling after resolving the G protein complex structure. The authors need to carry out a G protein signaling assay, preferably via Gi (structure solved) and Gq. They are ORF74's major coupling partners and, thus, possible KS drivers. We are also concerned that the ligand-stimulation was performed at a single concentration, limiting the information available. Minimally, the authors need to show CXCL1 dose-response for WT ORF74 and justify the choice of the concentration tested (5 nM) in their system. To further strengthen the authors' conclusion, it is ideal to include more chimeric constructs. For example, ORF74-based chimeras with CXCR2-ICL2/ICL3 should be tested: Does this lower ORF74's high basal activities?

It appears that the signaling is normalized to receptor expression as the authors did for cryo-EM constructs (Fig. S1c). The same set of data (expression levels and signaling without normalization) should be presented in the figure, as it is not recommended to perform simple normalization (signaling amplitude divided by expression level).

Fig. 6d

Clarify Y-axis labeling: what is "Normalized"?

3. Modeling of ORF74-BRIL-Fab-Nb

The authors did not model the BRIL-fused ORF74 due to the limited map quality, which is fine. However, RMSD is calculated between the protomer of the anti-parallel ORF74 dimer and BRIL-fused ORF74 (Fig. 1f). This needs to be consistent.

4. Statistical test

There is no description of statistical analysis applied in the method section and figure legends.

5. Language editing

A round of language editing and extensive proofreading is required by the next round of review.

As one example,

Line 149: "CXCR2 exhibits structural movements in the TMs that form the ESA region when transitioning between active and inactive conformations."

This sentence needs to be corrected, and the "ESA region" (and "ESA structure" in the following sentence) does not make sense. Extracellular region and extracellular structure?

Minor points

1. Constitutive active complex and inverse agonist-bound complex

We are curious why the authors did not attempt biochemical analysis or structure determination of the ORF74-Gi complex without chemokine to observe the constitutive active complex directly. Moreover, the inverse agonist-bound complexes (e.g. CXCL12-ORF74) is important to resolve to confirm the inactive conformation here is compatible with binding to the endogenous inverse agonists.

2. Map reconstruction and modeling

CXCL1-ORF74-Gi

3D variability analysis of the CXCL1-ORF74-Gi complex might give more insights into the docking mode between ORF74 and Gi. Even at ~3 Å resolution, the map has substantial conformational heterogeneity.

ORF74BRIL-Fab-Nb

The map quality and interpretability might be improved by a local refinement using a mask covering the 7TM region, with the use of "pose/shift gaussian prior during alignment." As this is easy to test using the particle set in hand, the author might consider re-refining the map with various masking.

Refinement statistics

The number of rotamer outliers needs to be lowered, considering the high quality of the maps. We will request the maps and models for the next round of review.

3. Signaling

Based on metadynamics analysis, the authors suggested alterations in conserved class A motifs lead to ORF74's high basal activity. In signaling assay, it is worth reverting these "mutated" amino acids to canonical motifs in Class A GPCR to see if it lowers ORF74's basal activity. This data significantly strengthens the authors' conclusion.

4. Metadynamics simulation

Line 210: "This increased flexibility could potentially influence TM6 dynamics and contribute to ligand-independent signaling and activation."

The authors claim ligand-independent signaling by metadynamics analysis, but the initial structure is derived from ORF74 bound to CXCL1 and Gi. This point needs to be clearly mentioned in the manuscript.

5. Extracellular surface area?

It is helpful to show the surface model of the structure and highlight measured "Extracellular surface area" with a different color. "ESA" is not a standardized term.

6. Other minor points

Line 118:

By introducing BRIL in ORF74-ICL3 and forming a complex with anti-BRIL Fab, dimer formation was certainly inhibited, but it is unclear from the figures if it is caused by steric hindrance because the dimers are anti-parallel in the nanodisc structure. The authors need to clarify this point by superimposing two ORF74BRIL-Fab-Nb maps on the anti-parallel dimer. We anticipated that the authors utilized the BRIL fusion as a fiducial marker to facilitate structure alignment of monomeric ORF74 during data processing rather than to induce steric hindrance.

Line 136:

"apo conformation of CXCR2" is misleading, as the antagonist is bound at the intracellular side of CXCR2, while the extracellular pocket is without any ligand.

Line 146:

"Unlike other chemokine receptors" is misleading as other chemokine receptors, such as US28, have the same properties.

Figure 3a:

β2AR and A2aAR are used for structural comparison, but they don't appear in the sequence alignment focusing on D2.50, D3.49, and W6.48. It would be easier to follow if the authors could add the sequences of β2AR and A2aAR there.

Figure S1a:

It appears that the G30C mutation is not included for the apo ORF74 sample but CXCL1-ORF74. Please clarify this point in Fig. S1a (+/- G30C).

Reviewer #2

(Remarks to the Author)

Reviewer #3

(Remarks to the Author)

Jun Bae Park and his colleagues investigated the structural basis for the ligand promiscuity and signalling activity of the Kaposi's sarcoma-associated herpesvirus encoded GPCR, ORF74. They used cryo-EM and metadynamics simulations to explore the receptor's ligand-independent signalling, highlighting unique structural features such as micro-switches and extracellular surfaces that facilitate its high basal activity and oncogenic potential. I think this is useful research, but there are still some problems in this manuscript, which need to be modified before acceptance.

1) The authors frequently referred to "promiscuity in ligand-binding". What form does this promiscuity take? Do all 13 ligands bind to ORF74 simultaneously, or do some ligands bind and dissociate at different times? Is this promiscuity always advantageous, or have any unfavorable cases been reported? If so, could the authors provide detailed explanations?

2) The paper highlighted ORF74-therapeutic potential. It would be beneficial for the author to provide some background on the clinical drug development process for relevant indications, as well as a brief description of the urgency and necessity of developing drugs targeting ORF74.

3) Some formatting and grammar errors need to be carefully checked and corrected. Including but not limited to: Page 6 Line 127, "represent" should be "represents".

Version 1:

Reviewer comments:

Reviewer #1

(Remarks to the Author)

Park et al. revised their manuscript to strengthen their conclusion and add deeper insights into the ORF74 activation. In our opinion, no additional experiment might be strictly required, but we highly recommend re-assessing their cryo-EM maps/models and signaling data and suggest expanding the discussion.

Map reconstruction

The authors attempted to improve the CXCL1-ORF74-Gi map, but several issues remain. Firstly, the authors created the composite map for visualization and model building, but it is unclear which software was used and even which maps were combined. Local refinements are usually performed on the receptor and G protein parts in the case of GPCR-G protein complexes, but Supplementary Table 1 indicates only the CXCL1-ORF74 part is locally refined, and there is no description if this is one of the constituent maps and if the G protein part is also refined locally. Please clarify this point and provide this information in the table, methods, and figure legends. In addition, a sharpening factor and/or method should be provided for each map.

Modeling

Inactive ORF74 and CXCL1-ORF74-Gi:

The authors need to consider removing atoms without adequate cryo-EM density, such as ones within the flexible loop.

CXCL1-ORF74-Gi:

The C-terminus of G α 1, F(-1), is not modeled but terminated at L(-2). Phenix or the PDB deposition system must have identified this critical issue; the authors need to carefully check their model using these tools for validation. Furthermore, the N-terminal loops of the CXCL1 and ORF74 do not best describe the cryo-EM density. Although this paper does not focus on the ligand-receptor interaction, this must be improved before the coordinates are publicly available, as it can significantly impact future studies. Lastly, it is controversial to create a model only for a composite map. The authors may consider if they want to keep this modeling strategy instead of creating models for each constituent map.

For these reasons, we request the below files for the revision of the manuscript.

- 1) unsharpen maps
- 2) sharpen maps used for modeling and refinement
- 3) models
- 4) local maps used to create the composite map
- 5) PDB validation reports

Signaling

Outline:

The authors collected ligand dose response for G protein signaling, which is nice, but the data analysis strategy is

confusing, giving the impression that the results are arbitrarily selected. The authors might use EC50 and Emax for a more robust data interpretation.

We do not think the signaling strength is proportional to the receptor expression in both the assay methods used in the manuscript, while normalization to the WT value is not a problem. The authors need to compare the signaling activities without normalization to receptor expression and then refer to expression between mutants. If the authors consider surface expression levels comparable, state so; if not, indicate the difference. For this purpose, statistical analysis of receptor expression is also necessary.

Also, the description is complicated and confusing due to the mixed claims about basal and ligand-induced activities. The authors need to consider reorganizing the main text to help readers follow the claims.

Supplementary Figure S10a:

It is somewhat concerning that not all the responses are completely saturated. We understand it might be challenging to increase ligand concentration further, but the authors may consider adjusting the concentration range for more reliable fitting. When comparing the ligand effect, it's better to adjust the origin of the curves.

In addition, the fitting for CXCR2_WT does not appear to be high quality. The authors must indicate fitting errors of the EC50 values.

Supplementary Figure S10b:

Please include mock MFI in the graphs and perform statistical analysis to check whether these expression levels differ significantly.

Supplementary Figure S10b:

We think both graphs are sensitive to experimental errors.

In the left bar graph, the authors picked the net BRET value around the approximate midpoint of the curves. EC50 should be a more robust indicator of potency.

In the right bar graph, the dose-response curves indicate the authors picked the net BRET value without the ligand effect, which is confusing. If the authors want to discuss ORF74's constitutive activity, which is important, a receptor gene dose is necessary without ligand stimulation to check the relationship between receptor expression and signaling for each variant and to obtain unambiguous conclusions. To discuss the ligand effect, use EC50 and/or Emax, or calculate intrinsic relative activity if the authors want.

Figure S6d:

The situation is like the comment for the FigS10b left, and it is unclear if this is a robust analysis. We do not know the reason for making three bar graphs instead of showing and analyzing the ligand dose-response curves.

Figure S6d:

We do not think the authors can decide the ligand concentration for the β -arrestin signaling assays solely based on the G protein signaling data, but as the authors might draw a main conclusion from the G protein signaling assays, we do not argue reconsideration, although the data needs to be re-analyzed as outlined.

Chimera selection:

The authors generated a series of CXCR2-ICL3ORF74 chimeras (A-E) for the β -arrestin recruitment assay. Did the authors perform the G protein signaling assay using the same constructs? Is there a specific reason to pick CXCR2_ICL3ORF74_C for the G protein assay?

Discussion

The structures of CXCL1-ORF74-Gi and ORF74-Gi complexes are now published in PNAS (Liu et al. bioRxiv 2023/PNAS 2024). We do not think the authors need to prepare a figure for comparison. Still, a brief mention of their study in the discussion, especially the ORF74-Gi structure, would make this manuscript more comprehensive.

Other issue

Proofreading:

The authors need to proofread the manuscript more carefully without rushing to submit the manuscript. We don't point out one by one at this stage, but for example, the name of Dr. Brian Kobilka is misspelled in the acknowledgment.

Labeling:

Figure labels and captions should also be improved; for example, it is unclear which map is displayed in each figure without proper labeling and description. In Figure S2c, the GSFSC is for the 3DFlex map based on Table S1, so Figure S2d probably shows the 3DFlex map, but the caption says only "cryoEM density map." Figure S2e is the overlay of the cryo-EM map on the model, so it is probably the composite map or 7TM local refined map, but the figure legend again says "cryoEM density map." Please go through each figure for visibility and interpretability.

Reviewer #2

(Remarks to the Author)

Reviewer #3

(Remarks to the Author)

The authors have addressed my concerns, and this research is recommended to be accepted.

Version 2:

Reviewer comments:

Reviewer #1

(Remarks to the Author)

We received the revision on April 3, 2025. The manuscript is substantially improved by refining the dose-response signaling, data reanalysis, and thorough editing. We acknowledge the authors' effort to enhance the quality of the work. There are mandatory changes and corrections, and we still suggest adjustments to controversial aspects of the data analysis, further manuscript editing, and corrections to minor errors. However, another round of peer review is not necessarily required, and an editorial assessment is sufficient for acceptance to prevent unintended delay in publication. Consider the comments below when finalizing the manuscript.

There is no page/line number in the manuscript. Thus, we can only specify either the page number of the merged file or the Figure/Table number.

<Mandatory changes>

-Statistical analysis

"ANOVA" is not enough information to disclose. Please indicate what post hoc methods are used to determine the significance between the values.

-Figure S1c

Remove the right panel.

We do not think this way of normalization is helpful, and even if it is allowed to normalize signaling to surface expression, the normalized mock signaling (~40% activity to WT) doesn't make sense. The mock values should be subtracted from the test values before normalization.

-Table S1

Please add sharpening B factors below "CryoSPARC" (or replace CryoSPARC with the B factors) as it is uniformly sharpened.

Use FSC = 0.5 to determine the model resolution (typo?). The authors can display the value at FSC = 0.143 together as a reference, but FSC = 0.5 must be used to report the model resolution. Another option is to show the map-model FSC curve in Figure S2/4.

-Page 3

The authors state ORF74 is activated by "CCL1, 2, and CXCL1-12," but no publication describes the activity at CXCL9 and CXCL11. Consult the most recent review papers or cite each original research.

-Page 12

"Among human CXCRs, CXCR2 has the highest ligand promiscuity, with 7 distinct known ligands, whereas ORF74 has approximately twice as many known ligands, with 13 known ligands." The numbers of ligands identified don't match the ones shown in the reference.

<Suggested adjustments>

-Maps and models

In addition to the previous discussion about the use of composite maps for modeling and refinement, it is even more controversial to combine globally and locally refined maps for modeling building and refinement, as well as to refine the structure in the deepEMhancer sharpened map. It is recommended to use these kinds of maps for simple visualization or a modeling guide.

However, the methods and descriptions are now detailed, and the peer review discussion will be publicly available for readers. Thus, we do not repeatedly request the change. Please make sure to deposit the half-maps, full maps, and masks used to refine the final maps in EMDB.

To avoid confusion, we do not recommend including hydrogen atoms in the final CXCL1-ORF74-Gi model at a resolution of around 3 Å, although it might be helpful to have these when refining the model.

-Signaling

We still think there is room to better support the claims, but we acknowledge these adjustments would not change the main conclusion of the study.

For example, normalization of signaling to expression does not sound, although they might behave as if they are proportional at a certain range of expression levels. However, we also acknowledge that the authors state normalization didn't change the significance of differences. Gene-dose signaling would be more reliable for assessing basal activity than Emin when comparing multiple variants. However, the improved dose-response curves are at least more convincing.

-Discussion (around page 14-16)

The Discussion is a little too lengthy because of the redundancy with the Result section and between paragraphs in the Discussion. It is helpful for readers to streamline the point already raised in the Result section to focus on more important aspects in Discussion.

-Page 22

The cryo-EM data analysis heavily relied on cryoSPARC, but the reference is not cited. As the authors used deepEMhancer maps for model refinement, more detailed information about deepEMhancer sharpening (i.e., half maps-based or full map-based, model used, and normalization mode) would be required.

Minor points

-Maybe describe the Ballesteros-Weinstein numbering and cite the paper.

-We recommend CXCL1-ORF74-Gi-scFv16, not ORF74-CXCL1-Gi-scFv16, to represent the complex as CXCL1 doesn't interact with Gi. These two are currently mixed, but it is better to use one of them to be consistent throughout the manuscript.

-Figure 1 legend

In one instance, "trimer" in "Gitrimer" is not subscripted.

"Gitrimer" should be "Gitrimer" in the Fig1f caption.

-Figure 5d

The interpretation and discussion would be more straightforward if the SASAs at the extracellular and intracellular sides were quantified separately.

-Figure 6 title

"ICL2 and ICL3 of ORF74 roles in its activity" should be something like "The role of ORF74's ICL2 and ICL3 in its activity" or insert "play" before "roles".

-Figure 6a

Add unit next to the red-white-blue bar or describe the unit in the figure legend. kT/e? at what temperature?

The structure of "ORF74-Active" is silhouetted in green. It is probably captured in a selected state in ChimeraX.

-Figure 6d legend

It is probably nice for broad readers to indicate in the caption that the graphs demonstrate Gi activation.

-Table S1

Ramachandran plot: The authors do not need to add % in each column as the unit is indicated in the left most column.

Please be consistent on whether the authors use the thousand separator or not.

-Figure S1d

The positions of the asterisks are confusing. They should be on top of the peaks, not on the side.

-Figure S2/4/5b

Add a scale bar on a 2D class.

-Figure S2/4/5d

Add "Å" next to the number indicating local resolutions.

-Figure S2e/s4e legend

Add the map contour levels.

-Figure S9b

In the MSA, the yellow highlighting makes the figure less visible. It would be easier to read by coloring only positively charged amino acids.

-Figure S12

We recommend removing this figure, as Figures S11 and S13 should be sufficient. However, as we indicated earlier, we respect the author's decision to use this normalization for ease of discussion. Just reconsider the treatment of mock values as commented for Figure S1c.

-Page 3

NF-κB, not NF-kB

-Page 5

"To capture the active conformation" is misleading. This should be "To capture the CXCL1-bound state" or similar.

"β-arrestin recruitment assays confirmed that these stabilizing and cysteine mutations did not affect ORF74 signaling" is misleading, as only β-arrestin recruitment is assessed. Either tone down this statement, or specifically state "did not affect ORF74-mediated β-arrestin signaling".

"HEK293S GnTi- cells" should be "HEK293S GnTi-(superscript) cells"

"3.0 Å resolution" and similar: be consistent with the numbers (significant digits) between the main text and Table S1. It is probably better to keep the ones in the main text.

-Page 6

"BILF1 GPCR of Epstein-Barr virus (EBV)": " of Epstein-Barr virus (EBV)" should be used when the word BILF1 appears in the manuscript for the first time.

"38.06" should be "38.1" to be consistent with other numbers.

-Page 9

HCMV, not "HCMV-1"

The abbreviation HCMV is not defined.

-Page 10

"a novel inactive conformation" should be "a unique inactive conformation" or similar.

" TM3(3.50) and TM6(6.30)" should be " R/K(3.50) and R/K(6.30)"

-Page 11

"electrostatic surface charge" should be "surface" to avoid redundancy

"curvature" should be "conformation"

The sentence "However, the ICL2 of ORF74 includes a proline residue, which is absent in CXCR2, resulting in differences in loop curvature and the presence of a bulky tryptophan side chain." is misleading. "the presence of a bulky tryptophan side chain" is not because of the proline residue.

-Page 12

"We identified an antiparallel dimer structure in the ligand-free ORF74, which is consistent with the structure of monomeric ORF74 containing a BRIL insertion in the ICL3, suggesting that both structures likely represent the same conformational state." This sentence is confusing, as it can be read as if the BRIL fusion ORF74 also forms the anti-parallel dimer. Please rephrase.

"In crystallographic approaches for elucidating GPCR structures, antiparallel orientations are frequently observed⁶². Recent

cryoEM studies have shown that Frizzled receptor 7, a class F GPCR, exists as an inactive antiparallel dimer⁶³. However, the functional significance of these orientations under physiological conditions remains unclear. Recent biophysical assays have revealed that GPCRs in antiparallel dimer configurations may possess distinct functional characteristics^{64,65}. It is probably better to introduce the Fzd7 (or single-particle cryo-EM) structure first and emphasize it is in a detergent micelle, then introduce the review paper regarding crystal structures but emphasize they are crystal packing generally considered non-functional, as reference #62 concludes, to be fair. Reference #64 (Rhodopsin) might not be appropriate here, but it might be discussed with #63.

-Page 14

The BW numberings are not superscripted.

-Page 16

"such a molecular insight" should probably be "molecular insights"

-Page 17

"P2 virus was added to cells": "cells" should be HEK293S GnTI-(superscript) cells?

April 9, 2025

Reviewer #2

(Remarks to the Author)

Reviewer #4

(Remarks to the Author)

In this work, the authors elucidate structural and functional details of ORF74, a viral GPCR that promotes cancer development. The authors perform a cryoEM analysis of ORF74 in various bound and apo forms in order to explain ORF74's high basal and agonist-induced activity. A detailed structural analysis is performed, as well as some metadynamics simulations, and the authors use the results to assert mechanisms involved in activation, including explanation for ligand promiscuity and a "highly dynamic equilibrium" between the active and inactive states.

This work is highly significant and interesting for the explanation of high basal and agonist-induced activity of this vGPCR. Plus, any novel structural explanation for activation/inactivation of membrane receptors represents interesting insight in my opinion. However, I have some critiques, and believe that the assertion by the authors that they have demonstrated that the "metadynamics simulations reveal a highly dynamic equilibrium between inactive and active states" is an overstatement given the amount of simulation evidence that they provide. Additional work will need to be performed before the authors will have adequate support for this.

On the topic of the metadynamics simulations, a review of the methods employed seem reasonable and valid, although the authors should specify what type of metadynamics they employed, and cite any relevant literature on the type, as well as, how they chose the settings, and what citations these came from, if any. From the use of a bias factor, I assume that they are using Well-Tempered Metadynamics. Also, the authors should not use statements like "standard cut-offs", because the standard or optimal cutoffs are debatable. They should report the exact values.

However, the authors' interpretations of the metadynamics simulations are where the problems start. For one, they only display the free energy landscape starting from the inactive structure (Figure 5a), and going to the transition state. They do not report or display an equivalent landscape for the active structure(s), which is a key missing piece for asserting a dynamic equilibrium. For instance, what if the active structure is even higher in energy than the transition state, making a dynamic equilibrium impossible? Also, according to that 2D energy landscape, with a transition height of approximately 30 kJ/mol, that is not a low barrier in relation to kT, so it feels like an overstatement to say "highly dynamic". Furthermore, an examination of the rest of figure 5 does not necessarily lend enough support to the idea of an equilibrium that is significantly more dynamic than CXCR or BILF1. For one, the methods state that the authors ran simulations in four separate replicas. If so, then which of these are plotted in figures 5c and 5d? And just because a metadynamics includes a change more quickly in one system simulation is no proof that the equilibrium between the active and inactive forms are more dynamic. Unfortunately, since the existence of a "highly dynamic equilibrium" is one of the main concluding points of this manuscript, the authors have not produced nearly enough convincing evidence to make this assertion.

Minor Issues:

In page 6 of the PDF provided to reviewers, the authors assert that the app and active conformations of ORF74 were superimposed and the RMSD was computed at 2.7 Angstroms. First of all, which subset of atoms were overlaid? Alpha carbons? Backbone atoms? Secondly, they say that this represents "significant structural divergence". I beg to differ - an RMSD of only 2.7 Angstroms does not seem to me to represent a significant structural divergence, in fact, if two structures were this close, I would consider them to be very similar. By what standard are these authors asserting that the divergence is "significant"? The authors should consider using a different, more compelling, metric to make this assertion.

In the section "System setup for metadynamics simulation", the authors mention that they modeled two distinct systems. They say "In the later system...", but they probably meant to say "In the latter system..."

Version 3:

Reviewer comments:

Reviewer #4

(Remarks to the Author)

The authors have adequately addressed my concerns and I believe that this manuscript is ready for publication.

Dear Reviewers,

We sincerely thank reviewers for interest in our work and for the constructive and helpful comments, which have greatly improved our manuscript. In the revised manuscript, we have included additional experimental data and carefully addressed all the comments. Furthermore, we have made efforts to enhance the clarity and accessibility of the manuscript to readers.

Please see the summary of responses to the general comments and the main comments of each reviewer as follows:

1. We performed a thorough language edit of the manuscript to correct grammatical errors and enhance clarity, revising specific sentences as suggested. (Reviewers #1, 3)
2. **Supplementary Data Figure 2.** We improved map quality, resolution consistency (2.98 Å), and local resolution accuracy by re-evaluating the particle set and refining the map with a dilated mask and increased L-BFGS iterations in the 3DFlex algorithm. (Reviewer #1)
3. **Figure 4d, e and Supplementary Figure 10, 11.** BRET assays revealed that ORF74 favors G α i over G α q in CXCL1 signaling, with ICL3 playing a more critical role than ICL2 in G protein dissociation and β -arrestin2 recruitment. (Reviewer #1)
4. **Figure 6e and Supplementary Figure 1c.** We updated Supplementary Figure 1c to include unnormalized surface expression and activity levels, calculated relative activity by normalizing to ORF74 WT, and provided detailed explanations in the figure legend. (Reviewer #1)
5. **Figure 1f.** We combined Figures 1f and 1g into a single Figure 1f for clarity and updated the figure legend accordingly. (Reviewer #1)
6. We added details about the statistical tests used to the Methods and Figure legend sections. (Reviewer #1)
7. We clarified the concept of "promiscuity" in the context of ORF74's interaction with multiple ligands, discussed its implications for signaling and immune evasion, and expanded on these points in the Discussion section of the manuscript. (Reviewer #3)
8. We added background on the clinical drug development process, the urgency, and the necessity of developing drugs targeting ORF74 to the Introduction and Discussion sections. (Reviewer #3)

Please see our detailed point-by-point responses to the reviewer's comments below. We believe that these revisions have substantially improved the manuscript, and we hope it will meet the standards for publication in Nature Communications.

Reviewer #1 (Remarks to the Author):

In this article, Park et al. present the structures of the KSHV GPCR, ORF74, in its active and inactive states. The structures describe ORF74's unique high basal activity and recognition of its full agonist CXCL1. Although the CXCL1-ORF74-Gi and ORF74-Gi structures have been posted to bioRxiv recently (Liu et al. 2023), it is not yet formally published, and the addition of the inactive, apo ORF74 structures is remarkable as it may facilitate the design of molecules that shutdown ORF74's pathological activity. There is still an outstanding question if a significant ORF74 population is inactive in complicated, physiological conditions on cells, but the authors demonstrated ORF74 can be in an inactive conformation in multiple experimental settings here, in lipidic nanodisc (anti-parallel dimer) and detergent (BRIL-fused monomer) with several thermostabilized mutations in cryo-EM study, and in lipidic environment in MD simulations. The data suggested a mechanism of ORF74's high basal activity, where the accumulated 7TM mutations lower the energy barrier between inactive and active conformations. Note that we are not able to provide high-quality comments on the technical details of the MD simulations in the manuscript, which appear solid, as non-expert.

The authors further attempted to gain mechanistic insights into ORF74 activation based on the signaling assays for structure-based chimeras. The study would be an important addition to the community, but several major concerns should be addressed.

Major points

RI-Q1. Map reconstruction and modeling. CXCL1-ORF74-Gi. The gold-standard FSC curve for the CXCL1-ORF74-Gi map does not fall to zero at high resolution, indicating that the final dataset for reconstruction includes substantial duplicated particles, and the map resolution is overestimated. For this map, it is also unusual that the peripheral region has the highest resolution (~2.5 Å), and the core region has a lower resolution (~4 Å). The authors need to remove the duplicated particles to re-refine and re-analyze the map. Because of the requirement for map re-refinement, we will request the new maps and models for assessment in the following review round.

RI-Q1-R: We thank the reviewer #1 for this valuable feedback. We thoroughly re-evaluated our particle set to ensure the absence of duplicate particles. Additionally, following the recommendation of the CryoSPARC team, we applied a slightly more dilated mask and increased the L-BFGS iterations to 60 during map re-refinement, using the same set of particles in 3DFlex reconstruction algorithm. These adjustments facilitated better convergence of the high-resolution portion of the FSC (**Supplementary Figure 2**). As a result, the GSFSC curve now falls to zero, and there is improved consistency between the map quality and the estimated resolution (2.98 Å). The accuracy of the local resolution estimation has also been improved. Notably, the 3D Flex algorithm's iterative optimization of particle position and orientation around flexible regions allowed the final refined map to capture higher local resolution in such areas. In our case, it's the CXCL1 density relative to ORF74. The updated map and model files, along with the improved statistics are included for your review. These revised results are now presented in **Supplementary Figure 2** and its corresponding figure legend as follows.

“Figure legend

~**Supplementary Figure 2.** CryoEM structural characterization of ORF74-CXCL1-G_{trimer}-scFv16. a, Representative cryo-micrograph and b, 2D class averages of the ORF74-CXCL1-G_{trimer}-scFv16 sample. c, Top panel - Gold standard Fourier shell correlation (GSFSC) plot and bottom panel - orientation distribution plot for particles (n = 190,755) used in the consensus 3D reconstruction. d, Local resolution of the cryoEM density map. e, High resolution information of the TM regions from the ORF74-CXCL1-G_{trimer}-scFv16 cryoEM density map.~

[Supplementary Data Figure 2]

RI-Q2. Signaling. The data is interesting, but it is confusing as to why the authors tested β -arrestin signaling after resolving the G protein complex structure. The authors need to carry out a G protein signaling assay, preferably via Gi (structure solved) and Gq. They are ORF74's major coupling partners and, thus, possible KS drivers. We are also concerned that the ligand-stimulation was performed at a single concentration, limiting the information available. Minimally, the authors need to show CXCL1 dose-response for WT ORF74 and justify the choice of the concentration tested (5 nM) in their system. To further strengthen the authors' conclusion, it is ideal to include more chimeric constructs. For example, ORF74-based chimeras

with CXCR2-ICL2/ICL3 should be tested: Does this lower ORF74's high basal activities?

RI-Q2-R: We thank reviewer #1 for the valuable suggestion. As suggested by reviewer, we performed additional G-protein dissociation experiments using a BRET2 assay (TRUPATH assay) including $G_{\alpha i}$ and $G_{\alpha q}$ subunit comparison.

The figure above (**Fig. RI-1**) compares the EC₅₀ values of CXCL1-mediated G-protein dissociation, as observed through changes in BRET signals, when pairing ORF74 with G_{αi} and G_{αq}. These data indicate that ORF74 has over 7-fold higher EC₅₀ value to the agonist CXCL1 with G_{αi} compared to G_{αq}, suggesting that ORF74 preferentially utilizes G_{αi} over G_{αq} in G-protein-mediated signaling. These results have been included in **Supplementary Figure 10**.

Using a BRET assay, we determined the EC₅₀ value of CXCL1 for both ORF74 and CXCR2 to be 14.2 nM (**Fig. RI-2**). Previously reported EC₅₀ value of CXCR2-CXCL1 and ORF74-CXCL1 range from 3-7 nM to 1-8 nM, respectively (doi.org/10.1074/jbc.271.34.20545, doi.org/10.1046/j.1432-1327.1998.2550067.x, doi.org/10.1371/journal.pone.0124486). These results indicate that our BRET data are consistent with previous findings.

[Figure. RI-3]

- **ORF74_ICL2_AAA** and **ORF74_ICL3_AAA** refer to constructs where the ICL2 and ICL3 regions of ORF74 have been replaced with alanine residues, respectively.
- For example, **ORF74_ICL2^{CXCR2}** represents a construct where the ICL2 of ORF74 has been replaced with the ICL2 of CXCR2.
- The **CXCR2_ICL3^{ORF74}_C** construct refers to the CXCR2 chimera construct shown in **Figure 6**.
- The Net BRET values were divided by the surface expression levels, and the resulting values were normalized to 1 based on the respective ORF74 WT or CXCR2 WT values (*Net BRET=Measured BRET-Background BRET).

To compare the ligand (CXCL1) potency of GPCRs, we measured EC50 values to evaluate the activity of the WT and chimeric constructs derived from the WT. Based on previous studies and our BRET-based assay, we identified a CXCL1 concentration range of 10^{-8} – 10^{-9} M as optimal for differentiating activities. Accordingly, this concentration range was used for G protein dissociation comparisons, and we selected 5×10^{-9} M CXCL1 (the midpoint of 10^{-8} – 10^{-9} M) for the β -arrestin2 recruitment assay. A similar study has previously been conducted to compare GPCR activity using K_i values in a BRET-based assay (doi.org/10.1038/s41467-024-54157-6).

We generated ORF74-based chimeras with a CXCR2-ICL2/ICL3 construct including the ORF74_ICL2/3_AAA construct and compared their activity to the WT (**Fig. RI-3**). A decrease in the CXCL1-stimulated activities was observed in ORF74-based chimeras in both 10^{-8} and 10^{-9} M CXCL1 concentration conditions, highlighting the critical role of ICL2 and ICL3 in activity of ORF74.

Conversely, the CXCR2 chimeras produced intriguing results. While G protein dissociation levels were comparable between CXCR2 WT and CXCR2_ICL2^{ORF74}, the CXCR2_ICL3^{ORF74} showed higher activity under the 10^{-9} M CXCL1 condition (**Figure RI-3**). Under the 10^{-8} M CXCL1 condition, however, the activity was similar to that of the WT (**Figure RI-3**), likely because the 10-fold higher CXCL1 concentration masked significant differences.

Notably, the extent of G protein dissociation differences relative to WT was lower in CXCR2_ICL2/3^{ORF74} than in ORF74_ICL2/3^{CXCR2}. Under 10^{-9} M and 10^{-10} M CXCL1 conditions, CXCR2_ICL3^{ORF74} had higher activity compared to CXCR2 WT (**Figure RI-3**). These findings confirm that ORF74 ICL2 and ICL3 are critical for G protein dissociation, with ICL3 playing a

more significant role than ICL2.

[Figure RI-4]

We have added β -arrestin2 recruitment assay data for the ORF74_ICL2/3^{CXCR2} construct to the existing dataset (Figure RI-4). The ORF74_ICL2/3^{CXCR2} construct exhibited reduced basal activity as well as lower CXCL1-mediated activity levels for both G protein dissociation and β -arrestin2 recruitment. These results indicate that ORF74 ICL2 and ICL3 are critical not only for G protein dissociation but also for β -arrestin2 recruitment.

Additionally, when comparing the G protein dissociation and β -arrestin2 recruitment levels of the CXCR2_ICL2^{ORF74} construct, we observed a decrease in G protein dissociation but an increase in β -arrestin2 recruitment. In contrast, the CXCR2_ICL3^{ORF74} construct showed an increase in both levels. These results suggest that ORF74 ICL3 is critical for both G protein dissociation and β -arrestin2 recruitment, whereas ORF74 ICL2 plays a more prominent role in β -arrestin2 recruitment than in G protein dissociation.

We have revised or added Figure 6b-e and Supplementary Figure 10 a-c, and updated the Results, Discussion, and Figure legend section of the manuscript, as follows.

[Figure 4b-e]

Supplementary Figure 10

[Supplementary Figure 10]

“Results

~To assess the G protein-mediated signaling activity of ORF74, we performed a BRET assay using Renilla luciferase 8 (Rluc8)-conjugated G α protein and green fluorescent protein 2 (GFP2)-conjugated G γ protein to measure the level of G protein dissociation. The degree of BRET signal reduction was used as an indicator of GPCR activation (**Fig. 6d**). To investigate the impact of ICL2 and ICL3 of ORF74 on GPCR activity, we created chimeric constructs by exchanging these loops between ORF74 and CXCR2. In active structures, the ICL2 of ORF74 comprises 8 amino acids (S150-K157), whereas the ICL2 of CXCR2 contains 9 amino acids (A151-R159). Similarly, The ICL3 of ORF74 spans 10 amino acids (T241-R250), whereas that of CXCR2 spans 6 amino acids (A241-K246). We carefully designed these chimeric constructs using solved structural data to minimize potential structural disruptions (**Fig. 6b,c**).

To compare sensitivity to CXCL1, we measured and compared the EC₅₀ values of ORF74 and CXCR2. Both ORF74 and CXCR2 exhibited similar CXCL1 EC₅₀ values at 14.2 nM, consistent with previous studies. Based on previous studies and our BRET-based assay, the optimal CXCL1 concentration range for differentiating activities was determined to be of 10⁻⁸–10⁻⁹ M. Therefore, this range was used for G protein dissociation comparisons, and 5x10⁻⁹ M (5 nM) CXCL1, the midpoint of this range, was selected for β -arrestin2 recruitment comparisons.

When ORF74 ICL2 and ICL3 were individually replaced with alanines, both constructs failed to show CXCL1-dose dependent G protein dissociation signals (**Supplementary Fig. 10a**). In addition, compared with the wild-type, the ORF74_ICL2^{CXCR2} and ORF74_ICL3^{CXCR2} chimeras demonstrated reduced G protein dissociation and β -arrestin2 recruitment. In particular, β -arrestin2 recruitment showed an approximately twofold reduction in both basal and 5 nM CXCL1-mediated activities (**Fig. 6d,e, Supplementary Fig. 10a,b, Supplementary Fig. 11**). Together, these findings highlight the critical role of ICL2 and ICL3 of ORF74 in G protein coupling and β -arrestin2 recruitment.

Conversely, the CXCR2 chimeras produced intriguing results. While G protein dissociation levels were similar between CXCR2 WT and CXCR2_ICL2^{ORF74}, the CXCR2_ICL3^{ORF74} showed higher levels under the 10⁻⁹ M CXCL1 condition (**Fig. 6d**). However, at the 10⁻⁸ M CXCL1 condition, the values were similar to those of the WT (**Supplementary Fig. 6c**), likely because the 10-fold higher CXCL1 concentration masked significant differences. Notably, the extent of G protein dissociation differences relative to WT was lower in CXCR2_ICL2/3^{ORF74} than in ORF74_ICL2/3^{CXCR2}. Under the 10⁻⁹ and 10⁻¹⁰ M CXCL1 conditions, CXCR2_ICL3^{ORF74} had higher G protein dissociation activity compared to CXCR2 WT. These findings highlight the critical roles of ORF74 ICL2 and ICL3 in G protein dissociation, with ICL3 playing a more significant role than ICL2.

In the β -arrestin2 recruitment assay, the CXCR2_ICL2^{ORF74} construct exhibited a 1.4-fold increase in basal activity compared with the WT (**Fig. 6e**). Notably, the CXCR2_ICL3^{ORF74} construct demonstrated basal signaling activity comparable to that of the ORF74 WT construct and significantly greater than that of the CXCL1-stimulated ORF74 construct (**Fig. 6e**). Additionally, when comparing the G protein dissociation and β -arrestin2 recruitment levels of the

CXCR2_ICL2^{ORF74} construct, we observed a decrease in G protein dissociation but an increase in β -arrestin2 recruitment. In contrast, the CXCR2_ICL3^{ORF74} construct showed an increase in both measures. These results suggest that ORF74 ICL3 is critical for both G protein dissociation and β -arrestin2 recruitment, whereas ORF74 ICL2 plays a more prominent role in β -arrestin2 recruitment than in G protein dissociation.~”

“Discussion

~ ORF74 exhibits a preference for the Gai subunit, which is a feature that sets ORF74 apart from HCMV-derived US28 with a preference for the Gaq subunit, which prefers the Gaq subunit. In the BRET assay measuring CXCL1 potency, ORF74 demonstrated a 7.3-fold higher activity under Gai subunit conditions compared to Gaq subunit conditions (**Supplementary Fig. 10a**). CXCR2 also shows a preference for the Gai subunit. However, ORF74 possesses a unique structure for Gai protein binding, with a particular focus on the ICL2 and ICL3 regions. We observed an enrichment of positively charged residues in the ICL2 and ICL3 of ORF74 (**Fig. 6**), a feature associated with increased G-protein coupling and β -arrestin2 recruitment efficiency. Given the structural importance of ICL2 and ICL3 in GPCR function, we investigated their roles in ORF74 activity. Interestingly, chimeric CXCR2 receptors with either ICL2 or ICL3 substituted with ORF74 sequences showed increased G protein dissociation and β -arrestin2 recruitment activity (**Fig. 6d,e**). Notably, the ICL3-substituted CXCR2 chimera exhibited greater activity than the wild-type CXCR2 chimera in both G protein coupling and β -arrestin2 recruitment. In the case of β -arrestin2 recruitment, CXCR2-ICL3^{ORF74} activity is comparable to that of wild-type ORF74, with even higher activation levels observed upon CXCL1 stimulation (**Fig. 6e**). These findings demonstrate that compared with CXCR2, ORF74 exhibits higher activation levels not only in G protein coupling but also in β -arrestin2 recruitment. Previous studies have revealed that β -arrestin2 can influence tumorigenesis by modulating inflammation and angiogenesis through CXCR2 and NF- κ B activation in a murine lung cancer model (doi: 10.1172/JCI15849). The relationship between ORF74 and β -arrestin in cancer has not been previously investigated. In parallel with existing research on CXCR2, we hypothesize that β -arrestin might play a role in ORF74-mediated cancer progression. This potential involvement could be particularly driven by ORF74's unique ICL2 and ICL3, with its high basal activity potentially amplifying these effects.~”

“Figure legend

~**Figure 6.**~ **d.** The results of the G protein dissociation assay using chimeric proteins with swapped ICL2 and ICL3 regions from ORF74 and CXCR2. ORF74-Gai or CXCR2-Gai activation responses in wild-type (WT) and ICL2 or ICL3 mutant upon stimulation with 10⁻⁹ [M] CXCL1. Net BRET was calculated as the raw BRET ratio minus the same ratio measured from cells expressing without the GFP2-G γ which is a BRET acceptor. Net BRET values (Y axis) which are obtained from CXCL1 concentration-response curves (**Supplementary Fig. 10a**) were corrected by dividing by the surface expression levels (**Supplementary Fig. 10b**) and then normalized to each ORF74 WT and CXCR2 WT, respectively. The data shown represent at least four independent experiments (n \geq 4), and bars denote mean \pm SEM (****p < 0.0001, ***p < 0.001, **p < 0.01, *p < 0.05, ns > 0.05). Significance was measured by ordinary one-way ANOVA with reference to each ORF74 WT and CXCR2 WT, respectively. **e.** Comparison of the basal activity and CXCL1-induced activity of the chimeric CXCR2 and ORF74 proteins, as

measured by β -arrestin2 recruitment. The relative β -arrestin2 recruitment value was calculated by dividing the activity values (luminescence RLU) by the surface expression levels (absorbance), followed by normalization to 1 using the CXCR2 basal activity value. The unnormalized activity and surface expression values are presented in **Supplementary Figure 11**. The data shown represent at least four independent experiments ($n \geq 4$), and bars denote mean \pm SEM (**** $p < 0.0001$, *** $p < 0.001$, ** $p < 0.001$, * $p < 0.05$, ns > 0.05). Significance was measured by ordinary two-way ANOVA with reference to CXCR2 WT.~”

“Figure legend

~**Supplementary Figure 10**. CXCL1-mediated G protein dissociation assay by BRET. **a**. G protein dissociation was assessed for each construct using a BRET assay. Net BRET was calculated as the raw BRET ratio minus the ratio measured in cells lacking GFP2-G γ , the BRET acceptor. EC₅₀ values were determined using GraphPad Prism. Except for the ORF74_WT-Gaq sample, all other constructs utilized the G α i subunit. ORF74_ICL2_AAA and ORF74_ICL3_AAA denote constructs where each respective ICL was replaced with alanines. The CXCR2_ICL3^{ORF74}_C construct was labeled following the method detailed in **Fig. 6c**. **b**. Surface expression levels of individual constructs. **c**. Comparison of changes in G protein dissociation among ORF74 and CXCR2 ICL2 and ICL3 chimeras in response to CXCL1 concentrations. The data represent results from at least four independent experiments ($n \geq 4$), with bars indicating mean \pm SEM. Statistical significance was assessed using ordinary one-way ANOVA, with comparisons made relative to each ORF74 WT and CXCR2 WT, respectively (**** $p < 0.0001$, *** $p < 0.001$, ** $p < 0.01$, * $p < 0.05$, ns > 0.05).~”

It appears that the signaling is normalized to receptor expression as the authors did for cryo-EM constructs (Fig. S1c). The same set of data (expression levels and signaling without normalization) should be presented in the figure, as it is not recommended to perform simple normalization (signaling amplitude divided by expression level). Fig. 6d, Clarify Y-axis labeling: what is "Normalized"?

As recommended, we have updated **Supplementary Figure 1c** to include graphs showing the unnormalized surface expression levels and activity levels. The relative activity was calculated by dividing the activity values by the surface expression levels, followed by normalization to 1 using the ORF74 WT value. This information has been detailed in the Figure legend, as follows.

“Figure legend

~**c**, *Functional comparison between wild-type ORF74 and the mutant form engineered for structural studies. The relative activity was calculated by dividing the activity values by the surface expression levels, followed by normalization to 1 using the ORF74 WT value.~”*

Additionally, we have renamed the Y-axis labeling of **Figure 6e** (previously **Figure 6d**, now updated to **Figure 6e**) to " β -arrestin2 recruitment (normalized to CXCR2 WT)" and provided an explanation of the normalization method in the Figure legend section, as follows.

“Figure legend

~The relative β -arrestin2 recruitment value was calculated by dividing the activity values (luminescence RLU) by the surface expression levels (absorbance), followed by normalization to 1 using the CXCR2 basal activity value. The unnormalized activity and surface expression values are presented in **Supplementary Figure 11.**~”

[Figure 6e and Supplementary Figure 1c]

Additionally, consistent with the presentation in **Supplementary Figure 1c**, we have included the unnormalized activity and surface expression values corresponding to **Figure 6e** in **Supplementary Figure 11**, as follows.

[Supplementary Figure 11]

“Figure legend

~**Supplementary Figure 11.** β -Arrestin2 recruitment was measured via the PRESTO-Tango method. The left panel shows the luminescence measurements reflecting β -arrestin2 recruitment, whereas the right panel presents the ELISA results for the surface expression level of each GPCR.~”

RI-Q3. Modeling of ORF74-BRIL-Fab-Nb. The authors did not model the BRIL-fused ORF74 due to the limited map quality, which is fine. However, RMSD is calculated between the protomer of the anti-parallel ORF74 dimer and BRIL-fused ORF74 (Fig. 1f). This needs to be consistent.

RI-Q3-R: We thank reviewer #1 for this comment. We have combined **Figures 1f** and **Figure 1g** into a **Figure 1f** to avoid confusion. Please refer to the updated figure legend and figure below.

“Figure legend

~ **Figure 1.** ~ **f**, Superimposition of the apo state structure (cyan) and the CXCL1/Gtrimer-bound structure (pink) highlights the dynamic movements of transmembrane helices 6 and 7, illustrated with red arrows, along with their RMSD comparison.~”

[Figure 1f]

RI-Q4. Statistical test. There is no description of statistical analysis applied in the method section and figure legends.

RI-Q4-R: As suggested, we have added information about the statistical tests to each Methods and Figure legend section as follows.

“Methods

~Statistical analysis

All the data are presented as mean \pm SEM and were assessed using the ANOVA for multiple comparisons. Statistical analysis was performed using GraphPad Prism software (version 10.1.2). Differences were considered statistically significant at $p < 0.05$.~”

“Figure legend

~**Figure 6.** ~**d.** G protein dissociation assay using chimeric proteins with swapped ICL2 and ICL3 regions from ORF74 and CXCR2. ORF74-G α i or CXCR2-G α i activation responses in wild-type (WT) and ICL2 or ICL3 mutant upon stimulation with 10^{-9} [M] CXCL1. Net BRET was calculated as the raw BRET ratio minus the same ratio measured from cells expressing without the GFP2-G γ which is a BRET acceptor. Net BRET values (Y axis) which are obtained from CXCL1 concentration-response curves (**Supplementary Fig. 10a**) were corrected by dividing

by the surface expression levels (**Supplementary Fig. 10b**) and then normalized to each ORF74 WT and CXCR2 WT, respectively. The data shown represent at least four independent experiments ($n \geq 4$), and bars denote mean \pm SEM ($****p < 0.0001$, $***p < 0.001$, $**p < 0.001$, $*p < 0.05$, $ns > 0.05$). Significance was measured by ordinary one-way ANOVA with reference to each ORF74 WT and CXCR2 WT, respectively. **e.** Comparison of the basal activity and CXCL1-induced activity of the chimeric CXCR2 and ORF74 proteins, as measured by β -arrestin2 recruitment. The relative β -arrestin2 recruitment value was calculated by dividing the activity values (luminescence RLU) by the surface expression levels (absorbance), followed by normalization to 1 using the CXCR2 basal activity value. The unnormalized activity and surface expression values are presented in **Supplementary Figure 11**. The data shown represent at least four independent experiments ($n \geq 4$), and bars denote mean \pm SEM ($****p < 0.0001$, $***p < 0.001$, $**p < 0.001$, $*p < 0.05$, $ns > 0.05$). Significance was measured by ordinary two-way ANOVA with reference to CXCR2 WT.~”

RI-Q5. Language editing. A round of language editing and extensive proofreading is required by the next round of review. As one example, Line149: "CXCR2 exhibits structural movements in the TMs that form the ESA region when transitioning between active and inactive conformations." This sentence needs to be corrected, and the "ESA region" (and "ESA structure" in the following sentence) does not make sense. Extracellular region and extracellular structure?

RI-Q5-R: As suggested, we have carefully revised the manuscript to improve its clarity and grammatical accuracy. In addition, along with the RI-MQ7 question, we revised the expressions related to ESA by changing them to ECS (Extracellular Surface) and modified the sentences as follows.

“Results

~One of the unique features of ORF74 is its promiscuous ligand binding ability and high degree of basal signaling. To characterize the extracellular surface (ECS) of the orthosteric binding site, which is solvent- and ligand-accessible and exposed in the extracellular region of the GPCR, we compared the ECSs of inactive and active ORF74 structures with those of CXCR2 and BILF1 (Fig. 2a). Compared with that of BILF1, the ECS of ORF74 displays an open conformation that is more similar to that of CXCR2; however, it does not demonstrate the distinctive structural movements between the active and inactive states observed in CXCR2 (Fig. 2a).~”

Minor points

RI-MQ1. Constitutive active complex and inverse agonist-bound complex. We are curious why the authors did not attempt biochemical analysis or structure determination of the ORF74-Gi complex without chemokine to observe the constitutive active complex directly. Moreover, the inverse agonist-bound complexes (e.g. CXCL12-ORF74) is important to resolve to confirm the inactive conformation here is compatible with binding to the endogenous inverse agonists.

RI-MQ1-R: We agree with the reviewer #1 that the ORF74-Gai complex plays a crucial role in explaining ORF74's high basal activity. While we did not directly determine the structure of the

ORF74-Gai complex in the absence of a chemokine in this study, our identification of the simultaneous binding of CXCL1 and Gai provides a sufficient structural basis to explain this phenomenon. Additionally, structural analysis of active and inactive ORF74 revealed that, unlike other Class A GPCRs, the shape and area of the extracellular pocket undergo minimal changes (**Figure 2a, b**). These findings suggest that the extracellular pocket undergoes negligible structural differences between the Gai-bound form and the CXCL1/Gai complex.

We also acknowledge the importance of studying an inverse agonist-bound complex, such as CXCL12-ORF74, to verify the compatibility of the inactive conformation with endogenous inverse agonists. While we think this to be beyond the scope of the current study, we plan to address this question in future research.

RI-MQ2. Map reconstruction and modeling. CXCL1-ORF74-Gi. 3D variability analysis of the CXCL1-ORF74-Gi complex might give more insights into the docking mode between ORF74 and Gi. Even at ~ 3 Å resolution, the map has substantial conformational heterogeneity.

RI-MQ2-R: We appreciate the reviewer's suggestion. As recommended, we performed 3D variability (3DV) analysis with the principal modes set to 5 and a filter resolution of 10 Å. Continuous variability was observed mostly among components 0, 1, and 2. Follow up analysis using the cluster mode indicated that "the helix5 of Gai1 remains stably docked within the intracellular pocket of ORF74's 7TMs, regardless of the strength or level of CXCL1 association at the extracellular surface (ECS) of ORF74". We have added the 3DV-related data in **Supplementary Figure 3** and its figure legend as shown below.

"Figure legend

*~ **Supplementary Figure 3.** 3D variability analysis of the ORF74-CXCL1-Gtrimer-scFv16 cryoEM data. a, Clustered scatter plot of the 3D variability (3DV) analysis, generated using a 3DV display job. b, Top panel – Consensus maps color-coded for each of the 5 clusters are shown. The percentage of particles distributed in each class is also indicated. The red dashed line marks the boundary of the overall map density. Bottom panel – Zoomed-in view of the Gai1/ $\alpha 5$ helix docking into the intracellular pocket of the ORF74/7TM regions.~*

Using the new re-refined Cryo-EM map and the corresponding re-refined model, we observed slight differences in the interaction between helix 5 of Ga11 and the intracellular domain of ORF74 compared to the previous model. However, we confirmed that these differences do not impact the conclusions described in the original manuscript. Based on the newly refined model, we have modified **Figure 1a-d** and **Supplementary Figure 8**, as shown below.

RI-MQ3. ORF74BRIL-Fab-Nb. The map quality and interpretability might be improved by a local refinement using a mask covering the 7TM region, with the use of "pose/shift gaussian prior during alignment." As this is easy to test using the particle set in hand, the author might consider re-refining the map with various masking.

RI-MQ3-R: We thank the reviewer #1 for this suggestion. During both the initial analysis and this revision, we conducted multiple local re-refinements using the existing particle set. These efforts included various focused masks and parameters, such as the "pose/shift Gaussian prior" option. However, these adjustments did not yield significant improvements over our current map. Given that we already have a high-resolution structure of the ORF74 apo conformation, we believe further refinement may not be necessary at this stage.

RI-MQ4. Refinement statistics. The number of rotamer outliers needs to be lowered, considering the high quality of the maps. We will request the maps and models for the next round of review.

RI-MQ4-R: As suggested, we have carefully re-evaluated and adjusted our models to minimize the number of rotamer outliers. The updated models, now with improved refinement statistics (**Supplementary Table 1**), are provided for your review.

Supplementary Table 1. Cryo-EM Data Collection, Refinement and Validation statistics

	(ORF74 Apo Dimer) (EMD-43717) (PDB 8W1A)	(ORF74-BRIL-BAG2-Nb Apo Monomer) (EMD-43720)	KSHV ORF74-CXCL1-Gi-scFv16 Complex			
			Composite map (EMD-48100) (PDB 9EJC)	NU-refined consensus map (EMD-43718)	3DFlex refined map (EMD-48095)	ORF74-CXCL1 local refined map (EMD-48097)
Data collection and processing						
Magnification	130,000×	130,000×	130,000×			
Voltage (kV)	300	300	300			
Electron exposure (e-/Å ²)	50	50	50			
Defocus range (μm)	-0.8 to -1.5	-0.8 to -1.5	-0.8 to -1.5			
Pixel size (Å)	0.66	0.66	0.66			
Symmetry imposed	C1	C1	C1			
Initial particle images (no.)	4,932,212	4,346,475	3,975,174			
Final particle images (no.)	697,456	111,016	190,755			
Map resolution (Å)	2.89	3.73	N/A	3.06	2.98	3.61
FSC threshold	0.143	0.143		0.143	0.143	0.143
Refinement						
Initial model used (PDB code)	AlphaFold2 predicted ORF74	N/A	AlphaFold2 predicted ORF74 and CXCL1, Gi-scFv16 (7JHJ)	N/A		
Model resolution (Å)	2.8		2.9			
FSC threshold	0.143		0.143			
Model composition						
Non-hydrogen atoms	4664		9485			
Protein residues	584		1214			
Ligands	0		0			
R.M.S. deviations						
Bond lengths (Å)	0.003		0.002			
Bond angles (°)	0.573		0.531			
Validation						
MolProbity score	1.28	1.45				
Clash score	4.08	3.69				
Rotamer outliers (%)	0	0				
Ramachandran plot						
Favored (%)	97.59%	95.82 %				
Allowed (%)	2.41 %	4.18 %				
Disallowed (%)	0.00 %	0.00 %				

[Supplementary Table 1]

RI-MQ5. Signaling. Based on metadynamics analysis, the authors suggested alterations in conserved class A motifs lead to ORF74's high basal activity. In signaling assay, it is worth reverting these "mutated" amino acids to canonical motifs in Class A GPCR to see if it lowers ORF74's basal activity. This data significantly strengthens the authors' conclusion.

RI-MQ5-R: We thank reviewer #1 for this suggestion. We have demonstrated that the micro-switches of ORF74 differ from those of canonical Class A GPCRs based on amino acid sequence and 3D structure analyses. Metadynamics simulations further revealed distinctive dynamics of the substituted amino acids in ORF74. Due to the complex nature of these mutations, we did not attempt back-mutations of individual amino acids. For example, in the case of W6.48C, we observed that not only this residue but also surrounding amino acids were simultaneously mutated (**Figure 3b**). This is a common feature observed in CWxP, D/ERY (D/E3.49V in **Figure 3h**), and NYxxY (N7.49V in **Figure 4a**) motifs, suggesting that single back-mutations may not fully capture the functional changes of ORF74.

However, we acknowledge that reverting the substituted residues to canonical motifs in Class A GPCRs could provide additional insights. In future studies, we plan to investigate this through systematic mutagenesis and functional assays to explore how these conserved motifs influence ORF74's high basal activity.

To enhance the manuscript, we have added the following relevant content to the Discussion section.

“Discussion

~We characterized the unique amino acid sequence and structural features of ORF74, building upon the known micro-switches of canonical class A GPCRs. Structural analysis revealed that, in addition to the distinctive micro-switches of ORF74, the surrounding amino acid residues also display unique structural characteristics. Specifically, the water-ion network-forming CWxP and NPxxY motifs were substituted with C6.48 and V7.49, respectively, and were further surrounded by non-canonical residues including S2.50, D3.39, and Y7.45 (Figure 3b, Figure 4a). Moreover, the DRY motif was substituted with V3.49 and flanked by R6.34 and an extended ICL3 (Figure 3h). These structural alterations, in combination with the CWxP, NPxxY, and DRY motifs, are likely to contribute to the elevated basal activity observed in ORF74. These findings suggest that the enhanced basal activity of ORF74 is not only attributed to its unique micro-switches but also influenced by the synergistic effects of the surrounding residues.~”

RI-MQ6. Metadynamics simulation. Line 210: "This increased flexibility could potentially influence TM6 dynamics and contribute to ligand-independent signaling and activation." The authors claim ligand-independent signaling by metadynamics analysis, but the initial structure is derived from ORF74 bound to CXCL1 and Gi. This point needs to be clearly mentioned in the manuscript.

RI-MQ6-R: We agree with the reviewer #1 that here we are explaining the flexibility of Cys compared to Trp in case of active (bound to CXCL1 and Gi) ORF74 simulation. We observed more fluctuation of Cys possibly because of G-protein binding which is potentially influencing TM6 dynamics more compared to the active CXCR2 simulation. Therefore, we have modified the sentence as follows.

“Results

~These findings suggest that the C6.48 residue has increased mobility upon Gai binding, unlike the W6.48 residue in CXCR2 and BILF1. The enhanced flexibility may affect TM6 dynamics, potentially allowing it to sample both inactive and active conformations.~”

RI-MQ7. Extracellular surface area? It is helpful to show the surface model of the structure and highlight measured "Extracellular surface area" with a different color. "ESA" is not a standardized term.¹

RI-MQ7-R: We thank reviewer #1 for this suggestion. We have added a surface model to **Figure 2a** to illustrate the differences in the "extracellular surface area" among the various structures as follows. As you suggested, this figure helps to better understand the graph of the "extracellular surface area" shown in **Figure 2b**. Additionally, we have replaced the term "ESA" (Extracellular Surface Area) with the standard term "ECS" (Extracellular Surface) throughout the manuscript, and we have included an explanation of ECS for clarity as follows.

“Results

~One of the unique features of ORF74 is its promiscuous ligand binding ability and high degree of basal signaling. To characterize the extracellular surface (ECS) of the orthosteric binding site, which is solvent- and ligand-accessible and exposed in the extracellular region of the GPCR, we compared the ECSs of inactive and active ORF74 structures with those of CXCR2 and BILF1 (Fig. 2a).~”

“Figure legend”

~ **Figure 2.** The unique extracellular surface area of ORF74 contributes to its promiscuity in ligand-binding. a, Comparison of the extracellular surfaces in the active and inactive conformations of ORF74, CXCR2 and BILF1.~”

RI-MQ8. Line 118: By introducing BRIL in ORF74-ICL3 and forming a complex with anti-BRIL Fab, dimer formation was certainly inhibited, but it is unclear from the figures if it is caused by steric hindrance because the dimers are anti-parallel in the nanodisc structure. The authors need to clarify this point by superimposing two ORF74BRIL-Fab-Nb maps on the anti-parallel dimer. We anticipated that the authors utilized the BRIL fusion as a fiducial marker to facilitate structure alignment of monomeric ORF74 during data processing rather than to induce steric hindrance.

RI-MQ8-R: We thank reviewer #1 for the valuable comment. The primary reason we introduced BRIL was to serve as a fiducial marker for cryo-EM particle analysis. In addition to this, our initial hypothesis was that the BRIL-Fab complex would disrupt the antiparallel dimer formation observed in the apo form of BRIL-fused ORF74. However, due to a lack of definitive evidence, as the reviewer pointed out, we have removed this specific claim from the manuscript and have revised as follows.

“Results

~We constructed the BRIL-fused ORF74 to serve as a fiducial marker, facilitating structure alignment during data processing.~”

RI-MQ9. Line 136: "apo conformation of CXCR2" is misleading, as the antagonist is bound at the intracellular side of CXCR2, while the extracellular pocket is without any ligand.

RI-MQ9-R: We thank reviewer #1 for pointing out the inaccuracy in our previous terminology. We have updated the text to replace 'apo conformation of CXCR2' with the more precise term 'inactive conformation of CXCR2.'

“Results

*~To further investigate whether the apo conformation adopts an inactive form, we compared the apo protomer model of ORF74 with the known inactive conformation of CXCR2 (**Supplementary Fig. 6**)~”*

RI-MQ10. Line 146: "Unlike other chemokine receptors" is misleading as other chemokine receptors, such as US28, have the same properties.

RI-MQ10-R: We thank reviewer #1 for pointing out the error in our previous statement. We have revised the sentence as follows.

“Results

~One of the unique features of ORF74 is its promiscuous ligand binding ability and high degree of basal signaling.~”

RI-MQ11. Figure 3a: β 2AR and A2aAR are used for structural comparison, but they don't appear in the sequence alignment focusing on D2.50, D3.49, and W6.48. It would be easier to

follow if the authors could add the sequences of β 2AR and A2aAR there.

RI-MQ11-R: We thank reviewer #1 for the suggestion. We have included β 2AR and A2aAR in **Figure 3a** to address this point as follows.

[Figure 3a]

RI-MQ12. Figure S1a: It appears that the G30C mutation is not included for the apo ORF74 sample but CXCL1-ORF74. Please clarify this point in Fig. S1a (+/- G30C).

RI-MQ12-R: We appreciate reviewer #1 for this insightful comment. We have revised **Supplementary Figure 1a**, as follows.

[Supplementary Figure 1a]

Reviewer #2 (Remarks to the Author):

Reviewer #3 (Remarks to the Author):

Jun Bae Park and his colleagues investigated the structural basis for the ligand promiscuity and signalling activity of the Kaposi's sarcoma-associated herpesvirus encoded GPCR, ORF74. They used cryo-EM and metadynamics simulations to explore the receptor's ligand-independent signalling, highlighting unique structural features such as micro-switches and extracellular surfaces that facilitate its high basal activity and oncogenic potential. I think this is useful research, but there are still some problems in this manuscript, which need to be modified before acceptance.

RIII-Q1. The authors frequently referred to “promiscuity in ligand-binding”. What form does this promiscuity take? Do all 13 ligands bind to ORF74 simultaneously, or do some ligands bind and dissociate at different times? Is this promiscuity always advantageous, or have any unfavorable cases been reported? If so, could the authors provide detailed explanations?

RIII-Q1-R: We thank reviewer #3 for the valuable comment. The term "promiscuity" when applied to ORF74 and its ligands refers to the protein's ability to interact with an unusually large number of distinct ligands – 13 in total – thereby modulating its signaling. This promiscuity is remarkable, as it is approximately twice the number of ligands reported for its functional homolog, human CXCR2, and represents the highest ligand count among known viral GPCRs. While individual ligands have been shown to exert specific effects on ORF74 function, the combinatorial effects of multiple ligands on ORF74 signaling remain largely unexplored.

The precise molecular mechanisms underlying ORF74's extensive ligand promiscuity and the selective advantages conferred by this property remain to be fully elucidated. Previous studies have categorized ORF74 ligands into five functional classes: full agonists, full inverse agonists, partial inverse agonists, neutral antagonists, and partial agonists. This diversity indicates that ORF74 can elicit diverse cellular responses depending on the specific ligand. Given the established link between ORF74 activation by full agonists such as CXCL1 and CXCL3 and oncogenesis, we selected CXCL1 as a primary ligand for structural studies. A comprehensive understanding of the relationship between viral life cycle phases, host chemokine activation, and ORF74 ligand promiscuity is anticipated to provide valuable insights into the viral pathogenesis.

The potential impact of ligand promiscuity on ORF74, or on KSHV mediated by ORF74, is as follows. Similar to canonical GPCRs, ORF74 undergoes β -arrestin-mediated internalization, a process facilitated by ligand binding. Through this mechanism, ORF74 downregulates cell surface TLR4, a key component of the innate immune response against KSHV, thereby promoting immune evasion. These findings suggest that ORF74-ligand interactions not only

contribute to oncogenic signaling but also play a critical role in host immune evasion. Moreover, the ligand promiscuity allows ORF74 to function as a chemokine scavenger. Overexpression of ORF74 in KSHV-associated cancers enables the internalization and subsequent proteasomal degradation of extracellular chemokines, potentially impairing antiviral immune responses from the host. However, this remains a hypothesis that requires further experimental validation.

To elaborate our manuscript, we have added more information to the Discussion section as follows.

“Discussion

~While individual ligands have been shown to exert specific effects on ORF74 function, the combinatorial effects of multiple ligands on ORF74 signaling remain largely unexplored. Previous studies have shown that the binding of CXCL1 and CXCL8 promotes the intracellular translocation of ORF74¹⁶. Intracellularly localized ORF74 results in the downregulation of host cell surface TLR4, which plays a crucial role in anti-KSHV activity, facilitating immune evasion⁶⁹. In contrast, cell surface-localized ORF74 is unable to mediate immune evasion⁶⁹. These findings, coupled with the observed ligand promiscuity of ORF74, suggest that the primary function of ORF74 may not be to propagate ligand-mediated signaling from the cell surface, but rather to execute other functions following internalization. Understanding the cellular localization of ORF74 is crucial in the anti-ORF74 drug discovery process, particularly in considering the necessity for cell penetration of potential therapeutic agents. The ligand promiscuity of ORF74 confers several potential advantages to the virus, including the ability to scavenge extracellular chemokines from the host cell microenvironment to evade host antiviral immune response and facilitate intracellular translocation to avoid neutralization by host antibodies. However, the full extent to which these mechanisms operate in KSHV-infected cells remains to be elucidated.~”

RIII-Q2. The paper highlighted ORF74-therapeutic potential. It would be beneficial for the author to provide some background on the clinical drug development process for relevant indications, as well as a brief description of the urgency and necessity of developing drugs targeting ORF74.

RIII-Q2-R: We thank reviewer #3 for this insightful suggestion. We have added more information to the Introduction and Discussion as follows.

“Introduction

~Human Herpesvirus-8 (HHV-8), also known as Kaposi's Sarcoma-associated Herpesvirus (KSHV), is the etiological agent of Kaposi's Sarcoma (KS), multicentric Castleman's disease, and primary effusion lymphoma. KS is the most prevalent neoplasm in AIDS patients and is endemic in sub-Saharan Africa. KSHV-derived pathogenesis has high global prevalence, with 73% of the cases occurring in Africa in 2020 focused in Eastern and Southern Africa . Additionally, approximately 15,000 deaths are attributed to KS annually, with 86.6% occurring in Africa. The higher occurrence in Africa is likely due to the relatively higher rates of KSHV and HIV coinfection compared to other regions. Notably, recent studies have also reported a concerning increase in the incidence of KS among younger populations in the U.S. Despite the public health impact of KSHV, there are no targeted therapies for KSHV-derived cancers.~”

“Discussion

~Despite numerous studies demonstrating ORF74 as a potent oncogene, current clinical trials for KSHV-derived cancers have not investigated ORF74-targeted therapies. This gap is likely due to a limited understanding of the mechanistic regulation of ORF74, particularly its inactive structure.~”

RIII-Q3. Some formatting and grammar errors need to be carefully checked and corrected.

Including but not limited to:

Page 6 Line 127, “represent” should be “represents”.

RIII-Q3-R: As suggested by reviewer #3, we have conducted a thorough language edit of the manuscript through peer review to address grammatical errors and improve overall clarity. We have revised the sentence as follows.

“Results

~This result suggests that the ORF74 structure from the antiparallel dimer likely represents the bona fide apo-state conformation.~”

REVIEWER COMMENTS

Reviewer #1 (Remarks to the Author):

Park et al. revised their manuscript to strengthen their conclusion and add deeper insights into the ORF74 activation. In our opinion, no additional experiment might be strictly required, but we highly recommend re-assessing their cryo-EM maps/models and signaling data and suggest expanding the discussion.

Q1) Map reconstruction

The authors attempted to improve the CXCL1-ORF74-Gi map, but several issues remain. Firstly, the authors created the composite map for visualization and model building, but it is unclear which software was used and even which maps were combined. Local refinements are usually performed on the receptor and G protein parts in the case of GPCR-G protein complexes, but Supplementary Table 1 indicates only the CXCL1-ORF74 part is locally refined, and there is no description if this is one of the constituent maps and if the G protein part is also refined locally. Please clarify this point and provide this information in the table, methods, and figure legends. In addition, a sharpening factor and/or method should be provided for each map.

R1) As suggested, we have updated the Methods section to include details on composite map generation. Additionally, **Supplementary Table 1** now provides information on both the composite map, and the auto-sharpening method applied to each map, as follows.

“Methods

*~In the case of the ORF74 Apo dimer and the ORF74-BRIL-BAK5 Fab-Nb structures, further Non-Uniform refinements led to high quality 2.8 Å, and 3.8 Å resolution cryoEM density maps, respectively. For the ORF74-CXCL1-Gi_{trimer}-scFv16 structure, a 3D-flexible reconstruction and refinement with a static mask based on the NU-refinement 3D volume enabled a similar high-quality 3 Å resolution cryoEM density map. To further improve the resolution of the CXCL1-ORF74 interface, we selectively masked and local refined the CXCL1-ORF74 region, then integrated it with the 3DFlex map using the ‘vop_maximum’ command in ChimeraX, ensuring proper scaling. Separate local refinement of the Gtrimer region was not performed as it was already well-resolved in the 3DFlex map. Interpretability of both 3DFlex map and CXCL1-ORF74 local refined maps were auto-enhanced using DeepEMhancer before combining them to generate the composite map, which was used for model building and visualization purposes. Comprehensive statistical information regarding all the cryoEM datasets and structures are available in the **Supplementary Table 1.**~”*

Supplementary Table 1. CryoEM Data Collection, Refinement and Validation statistics

	(ORF74 Apo Dimer) (EMD-43717) (PDB 8W1A)	(ORF74-BRIL-BAG2-Nb Apo Monomer) (EMD-43720)	KSHV ORF74-CXCL1-Gi-scFv16 Complex			
			Composite map (3DFlex+local refined map) (EMD-48100) (PDB 9EJC)	NU-refined consensus map (EMD-43718)	3DFlex refined map (EMD-48095)	CXCL1-ORF74 local refined map (EMD-48097)
Data collection and processing						
Magnification	130,000×	130,000×	130,000×			
Voltage (kV)	300	300	300			
Electron exposure (e-/Å ²)	50	50	50			
Defocus range (μm)	-0.8 to -1.5	-0.8 to -1.5	-0.8 to -1.5			
Pixel size (Å)	0.66	0.66	0.66			
Symmetry imposed	C1	C1	C1			
Initial particle images (no.)	4,932,212	4,346,475	3,975,174			
Final particle images (no.)	697,456	111,016	190,755			
Map resolution (Å)	2.89	3.73	N/A	3.06	2.98	3.61
FSC threshold	0.143	0.143		0.143	0.143	0.143
Auto sharpening	CryoSPARC	CryoSPARC		DeepEMhancer	DeepEMhancer	DeepEMhancer
Refinement						
Initial model used (PDB code)	AlphaFold2 predicted ORF74		AlphaFold2 predicted ORF74 and CXCL1, Gi-scFv16 from PDB: 7JHJ			
Model resolution (Å) FSC threshold	2.89 0.143		2.98 0.143			
Model composition Non-hydrogen atoms Protein residues Ligands	4664 584 0	N/A	9485 1214 0	N/A		
R.M.S. deviations Bond lengths (Å) Bond angles (°)	0.003 0.573		0.002 0.481			
Validation						
MolProbity score Clash score Rotamer outliers (%)	1.28 4.08 0		1.43 3.16 0			
Ramachandran plot Favored (%) Allowed (%) Disallowed (%)	97.59% 2.41% 0.00%		95.57% 4.43% 0.00%			

[Revised Supplementary Table 1]

Modeling

Q2-1) Inactive ORF74 and CXCL1-ORF74-Gi:

The authors need to consider removing atoms without adequate cryo-EM density, such as ones within the flexible loop.

R2-1) We have confirmed that using a lower cryoEM density threshold enables clearer tracing of nearly all modeled loops. Based on this observation, we believe that further modifications to these regions may not be necessary at this stage.

Q2-2) CXCL1-ORF74-Gi:

The C-terminus of Gai1, F(-1), is not modeled but terminated at L(-2). Phenix or the PDB deposition system must have identified this critical issue; the authors need to carefully check their model using these tools for validation. Furthermore, the N-terminal loops of the CXCL1 and ORF74 do not best describe the cryo-EM density. Although this paper does not focus on the ligand-receptor interaction, this must be improved before the coordinates are publicly

available, as it can significantly impact future studies. Lastly, it is controversial to create a model only for a composite map. The authors may consider if they want to keep this modeling strategy instead of creating models for each constituent map.

For these reasons, we request the below files for the revision of the manuscript.

- 1) unsharpen maps
- 2) sharpen maps used for modeling and refinement
- 3) models
- 4) local maps used to create the composite map
- 5) PDB validation reports

R2-2) Modeling the C-terminus of Gai1 – We acknowledge that the F(-1) residue was not modeled, and this has now been corrected.

N-terminal loops of CXCL1 and ORF74 – We have made slight adjustments to further improve map/model fitting for this region. A comparison with the published ORF74-CXCL1-Gtrimer structure (PDB: 8K4O, PNAS) now shows a high degree of similarity.

Model building from a composite map – The final model is essentially an improved version of the original 3DFlex-based model, with refinements around the CXCL1-ORF74 interface. Therefore, we think it is not necessary to create separate models for each map.

Requested files – All requested files have been provided for the reviewer's reference.

Signaling

Q3-1) Outline:

The authors collected ligand dose response for G protein signaling, which is nice, but the data analysis strategy is confusing, giving the impression that the results are arbitrarily selected. The authors might use EC50 and Emax for a more robust data interpretation. We do not think the signaling strength is proportional to the receptor expression in both the assay methods used in the manuscript, while normalization to the WT value is not a problem. The authors need to compare the signaling activities without normalization to receptor expression and then refer to expression between mutants. If the authors consider surface expression levels comparable, state so; if not, indicate the difference. For this purpose, statistical analysis of receptor expression is also necessary. Also, the description is complicated and confusing due to the mixed claims about basal and ligand-induced activities. The authors need to consider reorganizing the main text to help readers follow the claims.

R3-1) As reviewer pointed out, we compared the efficacy and potency of CXCL1 by analyzing BRET ratio differences among various constructs, focusing on pEC50 and Emin values. We chose Emin instead of Emax because, in some constructs, the BRET signal did not reach saturation at high CXCL1 concentrations, making Emax ambiguous. Additionally, since it is important to assess the basal activity of ORF74, we prioritized Emin, which corresponds to baseline activity (**Figure 6d**).

We confirmed that within the ORF74 group (including wild-type and chimeras) and within the CXCR2 group, expression levels showed no statistically significant differences in the BRET assay (Trupath assay) (**Supplementary Figure 13b**). Therefore, we did not normalize BRET signal to surface expression levels when comparing wild-type and chimeras.

In the β -arrestin recruitment assay (Tango assay), differences in surface expression levels were observed in both the ORF74 and CXCR2 groups (**Supplementary Figure 13c**). Therefore, we present both the data normalized to surface expression levels (**Supplementary Figure 12**) and the non-normalized data (**Supplementary Figure 11**). Additionally, statistical analysis confirmed that normalization to surface expression levels did not affect the results.

Furthermore, we explicitly stated the statistical analyses performed for the BRET assay, surface expression levels, and other relevant data to ensure clarity regarding statistical significance. To enhance clarity and improve reader comprehension, we emphasized on the high basal activity of ORF74 when preparing figures and writing the manuscript, as outline below.

[Revised Figure 6d]

[Revised Supplementary Figure 11]

[Revised Supplementary Figure 12]

[Revised Supplementary Figure 13b,c]

“Result

~We conducted a bioluminescence resonance energy transfer (BRET2) assay to investigate the role of the intracellular loops ICL2 and ICL3 of ORF74 in $G_{\alpha i}$ -mediated signaling. The BRET2 assay is a highly sensitive technique that measures receptor activation by detecting energy transfer between *Renilla luciferase* (*Rluc*)-conjugated G_{α} and GFP2-conjugated G_{γ} in live cells. Upon GPCR activation, a conformational change induces the dissociation of the G_{α} and G_{γ} subunits, increasing the physical distance between *Rluc*-conjugated G_{α} and GFP2-conjugated G_{γ} . This spatial separation reduces energy transfer efficiency, leading to a decrease in the BRET signal. To specifically assess the impact of ICL2 and ICL3 on the basal activity of ORF74, we focused on analyzing the baseline BRET signal.

To determine the functional significance of ICL2 and ICL3, we generated chimeric constructs where ORF74’s ICL2 or ICL3 was replaced with the corresponding loops from CXCR2 (Fig. 6b,c). Both ICL2 and ICL3 alanine mutants exhibited a loss of BRET signal (Supplementary Fig. 10a), indicating that these loops are essential for G protein coupling. Furthermore, the ORF74-ICL2^{CXCR2} and ORF74-ICL3^{CXCR2} chimeras showed reduced baseline activity compared to wild-type ORF74, with the ICL3^{CXCR2} chimera exhibiting a greater reduction than ICL2^{CXCR2} (Fig. 6d). This suggests that while both intracellular loops contribute to ORF74’s constitutive activity, ICL3 may play a more dominant role in G protein-mediated signaling. Taken together, these findings highlight the critical role of both ICL2 and ICL3 in ORF74’s G protein-mediated signaling, with ICL3 serving as a particularly key determinant of basal activity.

We generated reciprocal chimeric constructs where the ICL2 or ICL3 of CXCR2 was replaced with the corresponding loops from ORF74 (CXCR2-ICL2^{ORF74} or CXCR2-ICL3^{ORF74}, **Fig. 6b,c**). Both chimeras exhibited enhanced baseline activity compared to wild-type CXCR2, with CXCR2-ICL3^{ORF74} demonstrating a greater increase than CXCR2-ICL2^{ORF74} (**Fig. 6d**). This reinforces the notion that ICL3 is a stronger determinant of basal activity than ICL2. Interestingly, CXCR2-ICL3^{ORF74} retained CXCL1 potency comparable to wild-type CXCR2 while displaying increased baseline activity, whereas CXCR2-ICL2^{ORF74} exhibited altered ligand responsiveness (**Fig. 6b,c**). This suggests that while both intracellular loops modulate receptor activity, ICL3 plays a distinctive role in stabilizing an active conformation, contributing more significantly to constitutive signaling.~”

“Discussion

~The ICL2 and ICL3 of ORF74 exhibit a characteristic positively charged surface, which we demonstrated to be critical for G protein-mediated signaling through the CXCR2-ORF74 chimeric constructs (**Fig. 6c,d**). Notably, ORF74’s ICL3 contains 11 amino acids, whereas CXCR2 has only 6, suggesting that loop length may contribute to functional divergence. To investigate the specific role of ORF74’s extended ICL3, we generated total five CXCR2_ICL3^{ORF74} variants (including CXCR2_ICL3^{ORF74-A} to ICL3^{ORF74-D}, **Supplementary Fig. 10b**). Among these, the highest baseline activity was observed when only the loop region was replaced with “TKLQAR” amino acids, without modifying the TM5 and TM6 regions of CXCR2. These findings suggest that the loop sequence itself, rather than the adjacent transmembrane domains, plays a dominant role in G protein activation. However, the mechanism underlying the increased baseline activity of the CXCR2 chimera compared to wild-type remains unclear. One possibility is that the positively charged residues in ORF74’s ICL3 facilitate stronger electrostatic interactions with the G_α subunit, thereby promoting high basal activation. Alternatively, the loop’s conformational flexibility may stabilize an active-like state even in the absence of ligand binding. Further structural studies, such as NMR or cryoEM analysis of the chimeric receptors, could provide deeper insights into this mechanism.

ORF74 is also known to recruit β-arrestin and undergo endocytic trafficking upon ligand binding¹⁶. To determine whether ICL2 and ICL3 contribute to this process, we performed a β-arrestin recruitment assay (PRESTO-Tango assay). Unlike the results observed in the BRET assay, none of the chimeric constructs, including CXCR2-ICL2^{ORF74}, CXCR2-ICL3^{ORF74}, and CXCR2-ICL3^{ORF74-A} to ICL3^{ORF74-D}, exhibited enhanced β-arrestin recruitment relative to the wild-type receptor (**Supplementary Fig. 11, 12**). Furthermore, these results remained consistent before and after normalization to surface expression levels (**Supplementary Fig. 11, 12 and 13c**). This suggests that ICL2 and ICL3 play distinct roles in G protein- versus β-arrestin-mediated signaling, and the modifications that increase G protein signaling do not necessarily enhance β-arrestin interaction.

Interestingly, the ORF74-ICL3^{CXCR2} chimera exhibited significantly lower β-arrestin recruitment compared to both wild-type ORF74 and the ORF74-ICL2^{CXCR2} chimera (**Supplementary Fig. 11a, 12a**), indicating that ICL3 is a key determinant of β-arrestin interaction. Taken together, our findings highlight the functional bifurcation of ICL3 in GPCR signaling. While it plays a dominant role in G protein-mediated activation, its influence on β-arrestin recruitment appears to be context-dependent. Future studies should investigate whether specific amino acids

within ICL3 differentially modulate these pathways, and whether this functional divergence could be exploited for biased ligand development targeting ORF74-mediated oncogenic signaling.

*We investigated the distinct functions of ICL2 and ICL3 in ORF74 using CXCR2 chimeras, focusing on differences in activity between CXCR2 wild-type and CXCR2 chimeras, while using ORF74 wild-type and ORF74 chimeras as controls. Comparisons between the CXCR2 and ORF74 groups were not the focus of our analysis, as they fell outside the scope of our study. In the BRET assay, surface expression levels within the ORF74 and CXCR2 groups were statistically comparable (**Supplementary Figure 13b**); therefore, normalization to surface expression was not performed. In contrast, in the β -arrestin recruitment assay, differences in surface expression levels were observed within both the ORF74 and CXCR2 groups (**Supplementary Figure 13c**). As results, normalization was performed separately within each group, using their respective wild-type as a reference. Consequently, direct comparisons between ORF74 and CXCR2, including their chimeras, are not appropriate based on our data.~”*

Q3-2) Supplementary Figure S10a:

It is somewhat concerning that not all the responses are completely saturated. We understand it might be challenging to increase ligand concentration further, but the authors may consider adjusting the concentration range for more reliable fitting.

When comparing the ligand effect, it's better to adjust the origin of the curves.

In addition, the fitting for CXCR2_WT does not appear to be high quality. The authors must indicate fitting errors of the EC50 values.

R3-2) We agree that the BRET signal does not reach saturation at high concentrations of CXCL1. Since we observed a saturation plateau at low CXCL1 concentrations, we used the Emin value to compare the baseline activity across different GPCR constructs.

To address concerns regarding curve fitting, we performed additional replicates of the BRET assay. We have included the resulting graphs with R² values and 95% confidence interval (CI_{95%}) for the pEC50 values to ensure transparency and reliability (**Figure 6d, Supplementary Figure 10a, 11a**).

[Revised Figure 6d]

[Revised Supplementary Figure 10a]

Q3-3) Supplementary Figure S10b:

Please include mock MFI in the graphs and perform statistical analysis to check whether these expression levels differ significantly.

R3-3) We have added the mock MFI values to the graph and statistical analysis results in Supplementary Figure 13b,c as shown below.

Q3-4) Supplementary Figure S10b:

We think both graphs are sensitive to experimental errors.

In the left bar graph, the authors picked the net BRET value around the approximate midpoint of the curves. EC50 should be a more robust indicator of potency. In the right bar graph, the dose-response curves indicate the authors picked the net BRET value without the ligand effect, which is confusing. If the authors want to discuss ORF74's constitutive activity, which is important, a receptor gene dose is necessary without ligand stimulation to check the relationship between receptor expression and signaling for each variant and to obtain unambiguous conclusions. To discuss the ligand effect, use EC50 and/or Emax, or calculate intrinsic relative activity if the authors want.

R3-4) To address concerns regarding experimental variability and to better examine the high basal activity of ORF74 as well as the role of ICL2 and ICL3 on this activity, we focused on Emin and pEC50 values, as reviewer suggested. Specifically, we emphasized baseline activity by using Emin value, which closely reflects basal activity (10.1016/j.jbc.2021.100503).

Q3-5) Figure S6d:

The situation is like the comment for the FigS10b left, and it is unclear if this is a robust analysis. We do not know the reason for making three bar graphs instead of showing and analyzing the ligand dose-response curves.

R3-5) We have removed the previous data according to Figure 6d (It seems Figure S6d from

reviewer's question is a typo, and it should refer to Figure 6d.) We revised the Figure 6d and manuscript to present the BRET assay results as a CXCL1 dose-response curve.

[Revised Figure 6d]

Q3-6) Figure S6d:

We do not think the authors can decide the ligand concentration for the β -arrestin signaling assays solely based on the G protein signaling data, but as the authors might draw a main conclusion from the G protein signaling assays, we do not argue reconsideration, although the data needs to be re-analyzed as outlined.

R3-6) To better understand the dose-dependent β -arrestin signaling activity, we performed the β -arrestin recruitment assay, measuring the CXCL1-dose dependent activity and presented the data graphically in **Supplementary Figure 11, 12**. This approach helps address the limitations of using a single ligand concentration (previously only 5 nM CXCL1).

Since the primary focus of this paper is not on β -arrestin recruitment, all related data have been moved to the Supplementary Figures. We used the β -arrestin recruitment assay data in conjunction with the G protein signaling data to further expand and complement our understanding of ORF74's ICL function.

[Revised Supplementary Figure 11]

[Revised Supplementary Figure 12]

Q3-7) Chimera selection:

The authors generated a series of CXCR2-ICL3ORF74 chimeras (A-E) for the β -arrestin recruitment assay. Did the authors perform the G protein signaling assay using the same constructs? Is there a specific reason to pick CXCR2_ICL3ORF74_C for the G protein assay?

R3-7) We use different vectors for the β -arrestin recruitment assay and the G protein signaling (BRET) assay. For the β -arrestin recruitment assay, we used the PRESTO-Tango assay vector (Addgene kit number: Kit #1000000068, 10.1038/nsmb.3014). This vector contains a V2 tail and rTA attached to the C-terminus of the GPCR, with a TEV protease cleavage site in between. Additionally, a Flag tag is located at the N-terminus. In contrast, for the G protein signaling assay (BRET assay), we only added an N-terminal Flag tag to the GPCR for measuring surface expression levels.

To investigate the functions of ORF74 ICLs, we performed both PRESTO-Tango and BRET

assay on all CXCR2_ICL3^{ORF74} A to E constructs. The results were included in **Supplementary Figure 10b, 11b and 12b**. The original CXCR2_ICL3^{ORF74}_C construct was renamed to CXCR2_ICL3^{ORF74}, and the A, B, D, E constructs were renamed to A, B, C, D for the experiments, respectively.

[Revised Supplementary Figure 10b]

[Revised Supplementary Figure 11b]

[Revised Supplementary Figure 12b]

Q3-8) Discussion

The structures of CXCL1-ORF74-Gi and ORF74-Gi complexes are now published in PNAS (Liu et al. bioRxiv 2023/PNAS 2024). We do not think the authors need to prepare a figure for comparison. Still, a brief mention of their study in the discussion, especially the ORF74-Gi structure, would make this manuscript more comprehensive.

R3-8) As the reviewer pointed out, we have cited that paper and incorporated the following **Supplementary Figure 7b** and corresponding content into the Discussion section, as follows.

[Revised Supplementary Figure 7b]

“Discussion

*~A recent study published the structure of ORF74-G_itrimer (8K4P), and we compared the ECS region between this structure and the CXCL1-G_itrimer bound ORF74. The RMSD between the two structures was 0.6 Å, indicating high similarity. Additionally, the ECS regions exhibited comparable structures, confirming that the shape of the orthosteric binding pocket remains consistent (**Supplementary Fig. 7b**). Notably, the characteristic structural feature of TM2 was also observed. These findings support the notion that ORF74 maintains a similar ECS conformation regardless of ligand binding and/or G_itrimer association.~”*

Other issue

Q4-1) Proofreading:

The authors need to proofread the manuscript more carefully without rushing to submit the manuscript. We don't point out one by one at this stage, but for example, the name of Dr. Brian Kobilka is misspelled in the acknowledgment.

R4-1) We proofread the entire manuscript and made revisions to several sections, including the correction of Dr. Brian Kobilka's name. The major changes have been highlighted in green in the manuscript.

Q4-2) Labeling:

Figure labels and captions should also be improved; for example, it is unclear which map is displayed in each figure without proper labeling and description. In Figure S2c, the GSFSC is for the 3DFlex map based on Table S1, so Figure S2d probably shows the 3DFlex map, but the caption says only "cryoEM density map." Figure S2e is the overlay of the cryo-EM map on the model, so it is probably the composite map or 7TM local refined map, but the figure legend again says "cryoEM density map." Please go through each figure for visibility and interpretability.

R4-2) We have now added appropriate labels to clarify the maps shown in **Figure 1** and **Supplementary Figure 2**. Additionally, we have reviewed all figure captions to ensure clear and accurate descriptions for improved visibility and interpretability.

“Figure legend

~Supplementary Figure 2. CryoEM structural characterization of ORF74-CXCL1-G_itrimer-scFv16. a, Representative cryo-micrograph and b, 2D class averages of the ORF74-CXCL1-G_itrimer-scFv16 sample. c, Top panel - Gold standard Fourier shell correlation (GSFSC) plot for the 3DFlex map and bottom panel - orientation distribution plot for particles (n = 190,755) used in the consensus 3D reconstruction. d, Local resolution of the 3DFlex cryoEM density map. e, High resolution information of the TM regions from overlay of the ORF74-CXCL1-G_itrimer-scFv16 composite cryoEM density map and model.~”

Reviewer #2 (Remarks to the Author):

Reviewer #3 (Remarks to the Author):

The authors have addressed my concerns, and this research is recommended to be accepted.

Reviewer #1:

We received the revision on April 3, 2025. The manuscript is substantially improved by refining the dose-response signaling, data reanalysis, and thorough editing. We acknowledge the authors' effort to enhance the quality of the work. There are mandatory changes and corrections, and we still suggest adjustments to controversial aspects of the data analysis, further manuscript editing, and corrections to minor errors. However, another round of peer review is not necessarily required, and an editorial assessment is sufficient for acceptance to prevent unintended delay in publication. Consider the comments below when finalizing the manuscript.

Mandatory changes

R1Q1) Statistical analysis. "ANOVA" is not enough information to disclose. Please indicate what post hoc methods are used to determine the significance between the values.

- Thank you for your insightful comment. In our study, we performed ordinary one-way ANOVA followed by the default multiple comparisons test implemented in GraphPad Prism (version 10.1.2). Depending on the experimental context, either ORF74 WT or CXCR2 WT was used as the reference group for comparisons. The "Statistical Analysis" section in the Methods has been updated accordingly. Also, the figure legend indicates that detailed statistical methods are described in the Methods section.

*"All data are presented as mean \pm SEM. Statistical significance among multiple groups was assessed using Ordinary one-way ANOVA with GraphPad Prism's built-in multiple comparisons test (version 10.1.2). Comparisons were performed using ORF74 WT or CXCR2 WT as the reference group, depending on the experimental condition. Differences were considered statistically significant at $p < 0.05$ (**** $p < 0.0001$, *** $p < 0.001$, ** $p < 0.01$, * $p < 0.05$, ns > 0.05)."*

R1Q2) Figure S1c. Remove the right panel. We do not think this way of normalization is helpful, and even if it is allowed to normalize signaling to surface expression, the normalized mock signaling (~40% activity to WT) doesn't make sense. The mock values should be subtracted from the test values before normalization.

- Thank you for your suggestion. As reviewer recommended, we deleted the right panel (relative activity), and for both "basal activity" and "surface expression" graphs, we re-calculated the values by subtracting the mock values.

R1Q3) Table S1. Please add sharpening B factors below "CryoSPARC" (or replace CryoSPARC with the B factors) as it is uniformly sharpened.

- As suggested, we have now replaced "CryoSPARC" with corresponding global sharpening B-factor values in Table S1.

R1Q4) Use FSC = 0.5 to determine the model resolution (typo?). The authors can display the value at FSC = 0.143 together as a reference, but FSC = 0.5 must be used to report the model resolution. Another option is to show the map-model FSC curve in Figure S2/4.

- We appreciate the reviewer's careful reading and have now updated the model resolution reporting to reflect the FSC = 0.5 criterion.

R1Q5) Page 3. The authors state ORF74 is activated by "CCL1, 2, and CXCL1-12, " but no publication describes the activity at CXCL9 and CXCL11. Consult the most recent review papers or cite each original research.

- We appreciate the corrections. As reported in the cited references (DOI: 10.1124/mol.109.057091), we have revised the ligand list to "CCL1, CCL5, CXCL1–CXCL8, CXCL10, and CXCL12"

R1Q6) Page 12. "Among human CXCRs, CXCR2 has the highest ligand promiscuity, with 7 distinct known ligands, whereas ORF74 has approximately twice as many known ligands, with 13 known ligands." The numbers of ligands identified don't match the ones shown in the reference.

- The number of CXCR2 ligands has been corrected to eight (CXCL1–CXCL3, CXCL5–CXCL8, and vCXCL1). In line with the reviewer's comment in **R1Q4'** regarding redundancy and the need to streamline the Discussion section, the corresponding sentence has been removed.

Suggested adjustments

R1Q1') Maps and models. In addition to the previous discussion about the use of composite maps for modeling and refinement, it is even more controversial to combine globally and locally refined maps for modeling building and refinement, as well as to refine the structure in the deepEMhancer sharpened map. It is recommended to use these kinds of maps for simple visualization or a modeling guide. However, the methods and descriptions are now detailed, and the peer review discussion will be publicly available for readers. Thus, we do not repeatedly request the change. Please make sure to deposit the half-maps, full maps, and masks used to refine the final maps in EMDB.

- As suggested, all the requested files used in the final refinements, including half-maps, and full maps have been deposited to the EMDB.

R1Q2') To avoid confusion, we do not recommend including hydrogen atoms in the final CXCL1-ORF74-Gi model at a resolution of around 3 Å, although it might be helpful to have these when refining the model.

- We have removed the hydrogen atoms from the final CXCL1–ORF74–Gi model and re-deposited the updated structure to the PDB.

R1Q3') Signaling. We still think there is room to better support the claims, but we acknowledge these adjustments would not change the main conclusion of the study. For example, normalization of signaling to expression does not sound, although they might behave as if they are proportional at a certain range of expression levels. However, we also acknowledge that the authors state normalization didn't change the significance of differences. Gene-dose signaling would be more reliable for assessing

basal activity than Emin when comparing multiple variants. However, the improved dose-response curves are at least more convincing.

- We appreciate the reviewer's thoughtful comments. We agree that normalization to expression level may not always fully reflect the functional output, particularly when the relationship between expression and signaling is nonlinear. However, as noted by the reviewer, we confirmed that normalization did not alter the statistical significance of the observed differences between variants. This supports the interpretation that the functional differences between specific constructs of ORF74 and CXCR2 are not merely due to variations in expression levels.

We also thank the reviewer for suggesting gene-dose signaling analysis. While such an approach would indeed provide additional insights, we believe that the ligand dose-response curves represent a valid and complementary method that supports our hypothesis, as they consistently revealed differential basal and ligand-induced signaling activities across constructs and experimental conditions.

R1Q4') Discussion (around page 14-16). The Discussion is a little too lengthy because of the redundancy with the Result section and between paragraphs in the Discussion. It is helpful for readers to streamline the point already raised in the Result section to focus on more important aspects in Discussion.

- We thank the reviewer for the helpful suggestion. In response, we have carefully revised the Discussion to remove redundancies with the Results section. Specifically, content that directly repeated key findings already described in the Results was removed, and the remaining text was streamlined to focus on broader implications and more interpretative insights. We believe these changes have improved the clarity and conciseness of the Discussion.

R1Q5') Page 22. The cryo-EM data analysis heavily relied on cryoSPARC, but the reference is not cited.

- We have now included the cryoSPARC citation (doi.org/10.1038/nmeth.4169) in the Methods section.

R1Q6') As the authors used deepEMhancer maps for model refinement, more detailed information about deepEMhancer sharpening (i.e., half maps-based or full map-based, model used, and normalization mode) would be required.

- The Methods section has been updated to include details of the DeepEMhancer sharpening procedure used.

Minor points

R1Q1'') Maybe describe the Ballesteros-Weinstein numbering and cite the paper.

- As reviewer pointed out, we described the Ballesteros-Weinstein numbering and cited the paper.

R1Q2'') We recommend CXCL1-ORF74-Gi-scFv16, not ORF74-CXCL1-Gi-scFv16,

to represent the complex as CXCL1 doesn't interact with Gi. These two are currently mixed, but it is better to use one of them to be consistent throughout the manuscript.

- We have now consistently used the term CXCL1-ORF74-G_{itrimer}-scFv16 throughout the manuscript.

R1Q3) Figure 1 legend. In one instance, "trimer" in "Gitrimer" is not subscripted. "Gtrimer" should be "Gitrimer" in the Fig1f caption.

- We have fixed the "G_{itrimer}" typo and resolved the subscription issue.

R1Q4) Figure 5d. The interpretation and discussion would be more straightforward if the SASAs at the extracellular and intracellular sides were quantified separately.

- Thank the R1 for this comment. However, since there is no significant change in the SASA values for the extracellular surface, calculating SASA separately for this region is unlikely to provide additional insights. We believe the overall SASA calculation already offers meaningful information.

R1Q5) Figure 6 title. "ICL2 and ICL3 of ORF74 roles in its activity" should be something like "The role of ORF74's ICL2 and ICL3 in its activity" or insert "play" before "roles".

- In accordance with the reviewer's suggestion, we have revised the title to "Figure 6. The role of ORF74's ICL2 and ICL3 in its activity."

R1Q6) Figure 6a. Add unit next to the red-white-blue bar or describe the unit in the figure legend. kT/e? at what temperature?

- As suggested, we added the unit "kT/e" next to the color bar in Figure 6a, and clarified in the figure legend that the electrostatic potential was calculated at 298.15 K.

R1Q7) The structure of "ORF74-Active" is silhouetted in green. It is probably captured in a selected state in ChimeraX.

- We have changed the silhouette style in Figure 6a to make it consistent.

R1Q8) Figure 6d legend. It is probably nice for broad readers to indicate in the caption that the graphs demonstrate Gi activation.

- We have revised the sentence to "BRET changes in ORF74 and its chimeras (left panel) and CXCR2 and its chimeras (middle panel) in response to dose-dependent CXCL1 stimulation, reflecting activation of G_{ai} protein."

R1Q9) Table S1. Ramachandran plot: The authors do not need to add % in each column as the unit is indicated in the left most column. Please be consistent on whether the authors use the thousand separators or not.

- All the suggested changes have now been incorporated.

R1Q10) Figure S1d. The positions of the asterisks are confusing. They should be on top of the peaks, not on the side.

- We have moved the asterisks to the top of the peaks.

R1Q11) Figure S2/4/5b. Add a scale bar on a 2D class.

- We have now added a scale bar to the 2D class image as requested.

R1Q12) Figure S2/4/5d. Add "Å" next to the number indicating local resolutions.

- We have now added "Å" next to the numbers indicating local resolutions as requested.

R1Q13) Figure S2e/s4e legend. Add the map contour levels.

- Contour levels of 0.4 and 0.5 have been included in the legends for Figures S2e and S4e, respectively.

R1Q14) Figure S9b. In the MSA, the yellow highlighting makes the figure less visible. It would be easier to read by coloring only positively charged amino acids.

- We have removed the yellow highlighting and marked the positively charged residues lysine and arginine in blue, while histidine was marked in green. A new Supplementary Figure has been added, so Figure S9 has been renumbered as Figure S11.

R1Q15) Figure S12. We recommend removing this figure, as Figures S11 and S13 should be sufficient. However, as we indicated earlier, we respect the author's decision to use this normalization for ease of discussion. Just reconsider the treatment of mock values as commented for Figure S1c.

- As suggested by the reviewer, we have deleted Figure S12 and re-calculated Figure S13b and S13c (now revised as Fig. S14b and S14c). As in Fig. S1c, we have subtracted the mock values during normalization.

R1Q16) Page 3. NF-κB, not NF-kB

- We have revised it to NF-κB.

R1Q17) Page 5. "To capture the active conformation" is misleading. This should be "To capture the CXCL1-bound state" or similar.

- We have revised the sentence to "To capture the CXCL1-bound state".

R1Q18) "β-arrestin recruitment assays confirmed that these stabilizing and cysteine mutations did not affect ORF74 signaling" is misleading, as only β-arrestin recruitment is assessed. Either tone down this statement, or specifically state "did not affect ORF74-mediated β-arrestin signaling".

- We have revised the sentence to "β-arrestin recruitment assays suggested that these stabilizing and cysteine mutations do not significantly alter ORF74-mediated β-arrestin

signaling.”.

R1Q19) "HEK293S GnTi- cells" should be "HEK293S GnTI-(superscript) cells"

- We have revised it to “HEK293S GnTI- cells”

R1Q19”) "3.0 Å resolution" and similar: be consistent with the numbers (significant digits) between the main text and Table S1. It is probably better to keep the ones in the main text.

-We have revised the sentence for clarity.

R1Q20) Page 6. "BILF1 GPCR of Epstein-Barr virus (EBV)": " of Epstein-Barr virus (EBV)" should be used when the word BILF1 appears in the manuscript for the first time.

- As suggested, we have specified “Epstein-Barr virus (EBV)” when BILF1 is first mentioned in the manuscript.

R1Q21) "38.06" should be "38.1" to be consistent with other numbers.

- We have corrected “38.06” to “38.1”

R1Q22) Page 9. HCMV, not "HCMV-1" The abbreviation HCMV is not defined.

- We have replaced “HCMV-1” with “HCMV” and added the full term “human cytomegalovirus (HCMV)”.

R1Q23) Page10. "a novel inactive conformation" should be "a unique inactive conformation" or similar.

- We have revised the phrase “a novel inactive conformation” to “a unique inactive conformation”

R1Q24) "TM3(3.50) and TM6(6.30)" should be " R/K(3.50) and R/K(6.30)"

- We have revised “TM3^{3.50} and TM6^{6.30}” to “R/K^{3.50} and R/K^{6.30}”.

R1Q25) Page 11. "electrostatic surface charge" should be "surface" to avoid redundancy

- We have revised “electrostatic surface charge” to “surface”.

R1Q26) "curvature" should be "conformation"

- We have revised “curvature” to “conformation”.

R1Q27) The sentence "However, the ICL2 of ORF74 includes a proline residue, which is absent in CXCR2, resulting in differences in loop curvature and the presence of a bulky tryptophan side chain." is misleading. "the presence of a bulky tryptophan

side chain" is not because of the proline residue.

- We have revised the sentence to avoid the misleading implication.

"The ICL2 of ORF74 contains a proline residue absent in CXCR2, leading to differences in loop curvature. Separately, the presence of a bulky tryptophan side chain in ORF74 also contributes to structural differences."

R1Q28) Page 12. "We identified an antiparallel dimer structure in the ligand-free ORF74, which is consistent with the structure of monomeric ORF74 containing a BRIL insertion in the ICL3, suggesting that both structures likely represent the same conformational state." This sentence is confusing, as it can be read as if the BRIL fusion ORF74 also forms the anti-parallel dimer. Please rephrase.

- We have revised the sentence to avoid the misleading implication.

"We identified an antiparallel dimer in ligand-free ORF74. The structure of each protomer in this dimer closely resembles that of monomeric ORF74 containing a BRIL insertion in ICL3, suggesting that both represent the same conformational state."

R1Q29) "In crystallographic approaches for elucidating GPCR structures, antiparallel orientations are frequently observed⁶². Recent cryoEM studies have shown that Frizzled receptor 7, a class F GPCR, exists as an inactive antiparallel dimer⁶³. However, the functional significance of these orientations under physiological conditions remains unclear. Recent biophysical assays have revealed that GPCRs in antiparallel dimer configurations may possess distinct functional characteristics^{64,65}." It is probably better to introduce the Fzd7 (or single-particle cryo-EM) structure first and emphasize it is in a detergent micelle, then introduce the review paper regarding crystal structures but emphasize they are crystal packing generally considered non-functional, as reference #62 concludes, to be fair. Reference #64 (Rhodopsin) might not be appropriate here, but it might be discussed with #63.

- We have revised the sentence at the reviewer's suggestion.

"Recent single-particle cryoEM analysis has revealed that the class F GPCR Frizzled receptor 7 (Fzd7) adopts an antiparallel dimer conformation in its inactive state within a detergent micelle environment. In contrast, many crystallographic studies have also reported antiparallel orientations of GPCRs, although these are generally considered artifacts of crystal packing and not physiologically relevant, as noted in the original review. While the functional significance of such dimer orientations under physiological conditions remains uncertain, recent biophysical studies suggest that antiparallel GPCR dimers may exhibit distinct functional properties."

R1Q30) Page 14. The BW numberings are not superscripted.

- We have corrected the formatting and superscripted all BW numbering.

R1Q31) Page 16 "such a molecular insight" should probably be "molecular insights"

- We have revised "such a molecular insight" to "molecular insights".

R1Q32) Page 17. "P2 virus was added to cells": "cells" should be HEK293S GnTI⁻ (superscript) cells?

- We have revised "cells" to "HEK293S GnTI⁻ cells".

Reviewer #2:

Reviewer #4:

In this work, the authors elucidate structural and functional details of ORF74, a viral GPCR that promotes cancer development. The authors perform a cryoEM analysis of ORF74 in various bound and apo forms in order to explain ORF74's high basal and agonist-induced activity. A detailed structural analysis is performed, as well as some metadynamics simulations, and the authors use the results to assert mechanisms involved in activation, including explanation for ligand promiscuity and a "highly dynamic equilibrium" between the active and inactive states.

This work is highly significant and interesting for the explanation of high basal and agonist-induced activity of this vGPCR. Plus, any novel structural explanation for activation/inactivation of membrane receptors represents interesting insight in my opinion. However, I have some critiques, and believe that the assertion by the authors that they have demonstrated that the "metadynamics simulations reveal a highly dynamic equilibrium between inactive and active states" is an overstatement given the amount of simulation evidence that they provide. Additional work will need to be performed before the authors will have adequate support for this.

R2Q1) On the topic of the metadynamics simulations, a review of the methods employed seem reasonable and valid, although the authors should specify what type of metadynamics they employed, and cite any relevant literature on the type, as well as, how they chose the settings, and what citations these came from, if any. From the use of a bias factor, I assume that they are using Well-Tempered Metadynamics. Also, the authors should not use statements like "standard cut-offs", because the standard or optimal cutoffs are debatable. They should report the exact values.

- We agree with the reviewer on the importance of providing specific details about the metadynamics method used. We employed Well-Tempered Metadynamics simulations for enhanced sampling and have now explicitly stated this in the Methods section. The choice of collective variables (CVs) is indeed crucial, as it significantly influences the sampling of conformational transitions. We selected two CVs: (1) the distance between TM3b and TM6b, which corresponds to the ionic lock distance-an established molecular switch for GPCR activation-and (2) the backbone RMSD of the receptor to monitor overall structural transitions from the initial conformation. We have now added appropriate citations to foundational and recent literature on Well-Tempered Metadynamics, including Barducci et al., 2008 (Phys. Rev. Lett), Wang et

al., 2022 (Comput. Struct. Biotechnol. J) and D'Amore et al., 2024, (Chem), to provide context for our choice of method and settings. Regarding the simulation parameters, we have replaced all vague terms such as “standard cut-offs” with exact values. For example, the cut-off distance for non-bonded interactions was set to 12 Å, and the neighbor list was updated every 20 steps. We have included these details explicitly in the Methods section to ensure full transparency and reproducibility.

However, the authors' interpretations of the metadynamics simulations are where the problems start. For one, they only display the free energy landscape starting from the inactive structure (Figure 5a), and going to the transition state. They do not report or display an equivalent landscape for the active structure(s), which is a key missing piece for asserting a dynamic equilibrium. For instance, what if the active structure is even higher in energy than the transition state, making a dynamic equilibrium impossible? Also, according to that 2D energy landscape, with a transition height of approximately 30 kJ/mol, that is not a low barrier in relation to kT , so it feels like an overstatement to say “highly dynamic”.

- We thank the reviewer for raising this important point about the interpretation of the metadynamics simulations. We acknowledge that our earlier description lacked sufficient clarity. Our data indicate substantial overlap in the local conformations of key microswitches, supporting the existence of a broader ensemble of active-like states. We also agree that showing only the energy landscape starting from the inactive structure was not sufficient to capture the full picture of the transition dynamics.

To address this, we have now included additional simulations and analyses. Specifically, we analyzed the free-energy landscape of active ORF74 (Supplementary Fig. 9a), which follows a distinct path toward lower-energy (fully active) states, likely due to stabilization by both ligand and G protein binding. To ensure a fair comparison with the ligand-free inactive structure, we additionally simulated the active structure of ORF74 in the absence of the ligand and G protein, and computed its corresponding free energy landscape (Supplementary Fig. 9b). Together, we compared the free energy landscapes of the inactive, active-unbound, and active conformations within a composite depiction (Supplementary Fig. 9c). The active-unbound structure, like the inactive form, progresses toward a higher-energy intermediate-referred to as transition state 2, whereas the inactive starting structure progresses towards transition state 1. This triad of simulations provides a clearer picture of the energy landscape and supports the presence of a wider ensemble of structures in both the inactive and active-like states. As noted above, the rotameric conformations of the micro-switches show substantially overlap between both the inactive and active states.

To further support this interpretation, we performed a comparable analysis for CXCR2 and BILF1 (Supplementary Fig. 9d). The active structures of CXCR2 and BILF1 exhibit trajectories similar to that of active ORF74, progressing toward fully active, lower-energy conformations. In contrast, the inactive CXCR2 transitions toward lower-energy states that likely represent its fully inactive conformation, highlighting its reliance on ligand-induced activation, as previously proposed. However, this contrast highlights the distinct energetic behavior of ORF74, whose inactive form appears capable of exploring higher-energy intermediates also accessible from the active-unbound form. These results collectively suggest an ensemble of dynamic structures unique to ORF74.

We have modified the manuscript to reflect these additional findings. We also changed the previous phrasing of “highly dynamic equilibrium” to “a dynamic ensemble between local switch structures corresponding to the inactive and active states, supporting spontaneous activation” to better align with our results. We hope these additions provide a more comprehensive and accurate interpretation of the metadynamics simulations.

Furthermore, an examination of the rest of figure 5 does not necessarily lend enough support to the idea of an equilibrium that is significantly more dynamic than CXCR or BILF1. For one, the methods state that the authors ran simulations in four separate replicas. If so, then which of these are plotted in figures 5c and 5d? And just because a metadynamics includes a change more quickly in one system simulation is no proof that the equilibrium between the active and inactive forms are more dynamic. Unfortunately, since the existence of a “highly dynamic equilibrium” is one of the main concluding points of this manuscript, the authors have not produced nearly enough convincing evidence to make this assertion.

- We appreciate the reviewer’s concern and have revised the manuscript to provide a more comprehensive and transparent analysis. In the original version of manuscript we showed the results from a single run. In the updated version, we now show the TM3-TM6 C α distance and the solvent-accessible surface area (SASA) profiles for all four independent replicas of the inactive and active ORF74 simulations (Supplementary Fig. 10). Across all replicas of the inactive ORF74 simulations, we observed a consistent increase in the TM3-TM6 distance, a hallmark of activation-associated conformational change in GPCRs. Similarly, the SASA values increased across all replicas, further supporting the notion that the receptor transitions away from the inactive state toward more solvent-exposed, active-like conformations. These observations strengthen the conclusion that the transitions are not isolated to a single trajectory but are reproducible features across multiple independent simulations.

Data from the active-unbound ORF74 simulations were used solely for comparison of the overall free-energy landscape. As our primary focus is on ORF74, we note that similar trends were observed in the replica simulations of inactive-active CXCR2 and active BILF1, consistent with the results shown in Fig. 5c and 5d. However, to maintain clarity and focus on ORF74, these data are not included.

Minor Issues:

R2Q1’) In page 6 of the PDF provided to reviewers, the authors assert that the apo and active conformations of ORF74 were superimposed and the RMSD was computed at 2.7 Angstroms. First of all, which subset of atoms were overlaid? Alpha carbons? Backbone atoms?

- The RMSD value was calculated using C α atoms, and we have revised the sentence accordingly.

“When the apo and active conformations of ORF74 were superimposed to analyze their structural differences (Fig. 1f), the root-mean-square deviation (RMSD) value for this overlay was 2.7 Å based on the C α atoms.”

R2Q2') Secondly, they say that this represents “significant structural divergence”. I beg to differ - an RMSD of only 2.7 Angstroms does not seem to me to represent a significant structural divergence, in fact, if two structures were this close, I would consider them to be very similar. By what standard are these authors asserting that the divergence is “significant”? The authors should consider using a different, more compelling, metric to make this assertion.

- We have removed the term “significant” from the sentence.

“When the apo and active conformations of ORF74 were superimposed to analyze their structural differences (Fig. 1f), the root-mean-square deviation (RMSD) value for this overlay was 2.7 Å based on the C α atoms.”

R2Q3') In the section “System setup for metadynamics simulation”, the authors mention that they modeled two distinct systems. They say “In the later system...”, but they probably meant to say “In the latter system...”

- We have revised the phrase “In the later system” to “In the latter system” in the manuscript.